# Distributional Diffusion Models with Scoring Rules

**Valentin De Bortoli** [* 1]  **Alexandre Galashov** [* 1 2]  **J. Swaroop Guntupalli** [1]  **Guangyao Zhou** [1]  **Kevin Murphy** [1]
**Arthur Gretton** [1 2]  **Arnaud Doucet** [1]

## Abstract

Diffusion models generate high-quality synthetic data. They operate by defining a continuous-time forward process which gradually adds Gaussian noise to data until fully corrupted. The corresponding reverse process progressively "denoises" a Gaussian sample into a sample from the data distribution. However, generating high-quality outputs requires many discretization steps to obtain a faithful approximation of the reverse process. This is expensive and has motivated the development of many acceleration methods. We propose to speed up sample generation by learning the posterior *distribution* of clean data samples given their noisy versions, instead of only the mean of this distribution. This allows us to sample from the probability transitions of the reverse process on a coarse time scale, significantly accelerating inference with minimal degradation of the quality of the output. This is accomplished by replacing the standard regression loss used to estimate conditional means with a scoring rule. We validate our method on image and robot trajectory generation, where we consistently outperform standard diffusion models at few discretization steps.

## 1. Introduction

Diffusion models (Ho et al., 2020; Song & Ermon, 2019; Sohl-Dickstein et al., 2015) have demonstrated remarkable success in synthesizing high-quality data across various domains, including images (Saharia et al., 2022b), videos (Ho et al., 2022), and 3D (Poole et al., 2023). These models proceed as follows: First, a forward diffusion process is defined where Gaussian noise is progressively introduced to corrupt the data. This allows us to learn a denoiser at a continuum of noise levels. Next, to generate new sam-

---
[*]Equal contribution [1]Google DeepMind [2]Gatsby UCL. Correspondence to: Valentin De Bortoli <vdebortoli@google.com>, Alexandre Galashov <agalashov@google.com>.

*Proceedings of the $42^{nd}$ International Conference on Machine Learning*, Vancouver, Canada. PMLR 267, 2025. Copyright 2025 by the author(s).

ples, we sample from the time-reversed diffusion process, leveraging the learned denoiser. Despite their impressive capabilities, diffusion models often require a large number of steps to faithfully approximate the time-reversal so as to obtain high-fidelity samples. To overcome this limitation, different solutions have been explored. One approach focuses on the development of improved numerical integrators (e.g. (Karras et al., 2022; Lu et al., 2022; Zheng et al., 2023)). Another common strategy involves the use of distillation techniques (e.g. (Luhman & Luhman, 2021; Song et al., 2023; Luo, 2023; Salimans et al., 2024)), which requires training a smaller, more computationally efficient model to emulate the behavior of a larger, pre-trained diffusion model. Finally, parallel simulation methods have also been explored (e.g. (Shih et al., 2023; Chen et al., 2024a)) and use more memory to avoid slow sequential processing.

We depart from these earlier approaches in simply proposing to sample from the generative process on a coarser time scale. However, naively using denoisers obtained from the training of diffusion models significantly degrades the quality of the outputs, as these denoisers do not capture the full posterior distribution of the clean data given its noisy version, but rather its conditional mean. We review the relevant background on diffusions and their limitations in Section 2.1. In earlier works, Xiao et al. (2022); Xu et al. (2024) proposed to use Generative Adversarial Networks (GANs) (Goodfellow et al., 2014) to learn such conditional distributions. We show here how to bypass adversarial training by relying on generalized scoring rules (Gneiting & Raftery, 2007), which we describe in Section 2.2. This yields a simple loss which interpolates between the classical regression loss used in diffusion models and *distributional* losses. We present our new class of **Distributional Diffusion Models** in Section 3, and provide theoretical grounding for our choice of distributional loss, as well as details of the interpolation to the regression loss, in Section 4. We review related work in Section 5.

Experiments in Section 6 demonstrate that our approach produces high-quality samples with significantly fewer denoising steps. We observe substantial benefits across a range of tasks, including image-generation tasks in both pixel and latent spaces, and in robotics applications. All proofs are in the supplementary and a table of notation is in Appendix A.

## 2. Background & Motivation

### 2.1. Diffusion models

We follow the Denoising Diffusion Implicit Models (DDIM) framework of Song et al. (2021a). Let $p_0$ be a target data distribution on $\mathbb{R}^d$. Consider $X_{t_0} \sim p_0$ and the process $X_{t_1:t_N} := (X_{t_1}, ..., X_{t_N})$ distributed according to

$$p(x_{t_1:t_N}|x_{t_0}) = p(x_{t_1}|x_{t_0}) \prod_{k=1}^{N-1} p(x_{t_k}|x_{t_0}, x_{t_{k+1}}), \quad (1)$$

with $0 = t_0 < \cdots < t_N = 1$. For $0 \leq s < t \leq 1$, we define $p(x_s|x_0, x_t) = \mathcal{N}(x_s; \mu_{s,t}(x_0, x_t), \Sigma_{s,t})$ to ensure that for any $t \in [0, 1]$,

$$p(x_t|x_0) = \mathcal{N}(x_t; \alpha_t x_0, \sigma_t^2 \mathrm{Id}), \quad (2)$$

for some schedule $\alpha_t, \sigma_t$ chosen such that $\alpha_0 = \sigma_1 = 1$ and $\alpha_1 = \sigma_0 = 0$. This ensures in particular that $p(x_1|x_0) = \mathcal{N}(x_1; 0, \mathrm{Id})$. One possible popular schedule is given by the *flow matching* noise schedule (Lipman et al., 2023; Albergo et al., 2023; Gao et al., 2024), i.e.,

$$\alpha_t = 1 - t, \qquad \sigma_t = t. \quad (3)$$

The mean and covariance of $p(x_s|x_0, x_t)$ are given by

$$\begin{aligned}\mu_{s,t}(x_0, x_t) &= (\varepsilon^2 r_{1,2}(s,t) + (1-\varepsilon^2)r_{0,1})x_t \\ &+ \alpha_s(1 - \varepsilon^2 r_{2,2}(s,t) - (1-\varepsilon^2)r_{1,1}(s,t))x_0, \\ \Sigma_{s,t} &= \sigma_s^2(1 - (\varepsilon^2 r_{1,1}(s,t) + (1-\varepsilon^2))^2)\mathrm{Id}, \quad (4)\end{aligned}$$

with $r_{i,j}(s,t) = (\alpha_t/\alpha_s)^i (\sigma_s^2/\sigma_t^2)^j$; see (Song et al., 2021a) and Appendix F for a discussion. In (4), $\varepsilon \in [0, 1]$ is a *churn* parameter which interpolates between a *deterministic* process ($\varepsilon = 0$) and a *stochastic* one ($\varepsilon = 1$).

We generate data $X_{t_0} \sim p_0$ by sampling $X_{t_N} \sim \mathcal{N}(0, \mathrm{Id})$ and $X_{t_k} \sim p(\cdot|X_{t_{k+1}})$ for $k = N-1, ..., 0$, where for $0 \leq s < t \leq 1$

$$p(x_s|x_t) = \int_{\mathbb{R}^d} p(x_s|x_0, x_t)p(x_0|x_t)\mathrm{d}x_0. \quad (5)$$

It can be easily checked that this process has the same marginal distributions, denoted $p_{t_k}(x_{t_k})$, as the process defined by $p_0(x_{t_0})p(x_{t_1:t_N}|x_{t_0})$ (see (1)) so that in particular $X_{t_0} \sim p_0$; see e.g. (Shi et al., 2024, Appendix E).

Subsequently, to avoid ambiguity, we write $p_{s|t}(x_s|x_t)$ for $p(x_s|x_t)$ and $p_{s,t}(x_s, x_t)$ for $p(x_s, x_t)$ for any $0 \leq s < t \leq 1$. Usually $p_{0|t}(x_0|x_t)$ in (5) is approximated for any $t$ by $\delta_{\hat{x}_\theta(t,x_t)}$, where $\hat{x}_\theta(t, x_t) \approx \mathbb{E}[X_0|X_t = x_t]$ is a neural network denoiser trained using a regression loss; i.e.

$$\mathcal{L}_{\mathrm{diff}}(\theta) = \int_0^1 w_t \mathbb{E}[||X_0 - \hat{x}_\theta(t, X_t)||^2]\mathrm{d}t, \quad (6)$$

for a weighting function $w_t$ (Kingma et al., 2021). Approximating $p_{0|t}(x_0|x_t)$ with a Dirac mass located at $\hat{x}_\theta(t, x_t)$ seems crude at first. However, as $N \to \infty$ with $t_{k+1} \to t_k$, the resulting *discrete* time process converges to a *continuous* time process which recovers the data distribution if $\hat{x}_\theta(t, X_t) = \mathbb{E}[X_0|X_t]$; see e.g. Song et al. (2021b).

When performing a coarse time discretization, however, this result no longer holds, and the Dirac approximation is poor. We propose here to learn a generative network to sample approximately from $p_{0|t}(x_0|x_t)$.

### 2.2. Scoring rules

We next recall the framework of scoring rules, as reviewed by Gneiting & Raftery (2007), and then describe the specific scoring rules used in this work. Consider two probability distributions $p, q$. A *scoring rule* $S(p, x)$ indicates the quality of a prediction $p$ when the event $Y = y$ is observed. The *expected score* is its expectation under $q$,

$$S(p, q) := \mathbb{E}_q[S(p, Y)]. \quad (7)$$

A scoring rule is *proper* when $S(q, q) \geq S(p, q)$. It is called *strictly proper* with respect to a class of distributions $\mathcal{P}$ when the equality holds if and only if $p = q$, for all $p, q \in \mathcal{P}$. In this work we focus on the class of proper scoring rules called *kernel scores* introduced by Gneiting & Raftery (2007, Section 5.1) which take the form

$$S_\rho(p, y) = \frac{1}{2}\mathbb{E}_{p \otimes p}[\rho(X, X')] - \mathbb{E}_p[\rho(X, y)],$$

where $\rho$ is a continuous negative definite kernel. Following Bouchacourt et al. (2016, Section 3.2), we will also consider *generalized kernel scores* of the form

$$S_{\lambda,\rho}(p, y) = \frac{\lambda}{2}\mathbb{E}_{p \otimes p}[\rho(X, X')] - \mathbb{E}_p[\rho(X, y)], \quad (8)$$

with $\lambda \in [0, 1]$. One score of particular interest is the *energy score* (Gneiting & Raftery, 2007, eq. 22) denoted $S_\beta$, which uses $\rho(x, x') = \|x - x'\|^\beta$ with $\beta \in (0, 2)$. We denote $S_{\lambda,\beta}$, its generalized version, called the *generalized energy score*. We will also employ $\rho = -k$ with $k$ a continuous positive definite kernel, see e.g. (Berlinet & Thomas-Agnan, 2004) and Appendix A for a definition. Similarly to (7), we define the *(generalized) expected energy score* and *(generalized) expected kernel score*.

Positive definite kernels are said to be characteristic to $\mathcal{P}$ when $S_\rho$ yields a strictly proper scoring rule on $\mathcal{P}$ (Sriperumbudur et al., 2010; 2011). While the exponentiated quadratic (rbf) kernel satisfies this property, we will also investigate other kernels such as the inverse multiquadratic

(imq) kernel and the exponential (exp) kernel,

$$k_{\texttt{imq}}(x, x') = (\|x - x'\|^2 + c)^{-1/2}, \quad (9a)$$

$$k_{\texttt{rbf}}(x, x') = \exp\left[-\|x - x'\|^2/2\sigma^2\right], \quad (9b)$$

$$k_{\texttt{exp}}(x, x') = \exp\left[-\|x - x'\|/\sigma\right]. \quad (9c)$$

Any proper scoring rule $S_\rho$ defines a divergence

$$D_\rho(p, q) := S(q, q) - S(p, q). \quad (10)$$

When $S = S_\rho$ and $\rho(x, x') = -k(x, x')$ with $k$ a positive definite kernel, we recover the squared Maximum Mean Discrepancy ($\mathrm{MMD}^2$) (Gretton et al., 2012). For $\rho(x, x') = \|x - x'\|^\beta$, we obtain the *energy distance* (Rizzo & Székely, 2016). The relation between these discrepancies was established by Sejdinovic et al. (2013) where the specific positive definite kernel family for which the MMD and energy distance are equivalent is described.

Letting $\beta \to 2$ for the energy distance, we obtain $D_\rho(p, q) = \|\mathbb{E}_p[X] - \mathbb{E}_q[X]\|^2$, resembling the integrand of (6). Indeed, up to a constant $C$ independent of $\theta$, letting $p = \delta_{\hat{x}_\theta(t, X_t)}$ and $q = p_{0|t}(\cdot|X_t)$, we get

$$\mathbb{E}_{p_t}[D_\rho(p, q)] = \mathbb{E}[\|X_0 - \hat{x}_\theta(t, X_t)\|^2] + C, \quad (11)$$

see Appendix B for a proof. In the case of $\rho = -k$ with $k$ given by (9a) or (9b), the relation between $D_\rho$ and the diffusion loss (6) is less immediate than for the energy distance. However the connection becomes apparent by keeping the leading terms in the Taylor expansions of the kernels, as shown in Proposition 4.3.

# 3. Distributional Diffusion Models

We now introduce Distributional Diffusion Models. At a high level, we replace the *regression* loss (6), used to learn approximation of the conditional mean $\mathbb{E}[X_0|X_t]$ in diffusion models, with a loss based on *scoring rules*, to learn an approximation $p^\theta_{0|t}(x_0|x_t)$ of $p_{0|t}(x_0|x_t)$. This is achieved by learning a generative network $\hat{x}_\theta(t, x_t, \xi)$ which aims to produce samples $X_0 \sim p^\theta_{0|t}(\cdot|x_t)$. This model takes as input a time $t \in [0, 1]$, a noisy sample $x_t$ and a Gaussian noise sample $\xi \sim \mathcal{N}(0, \mathrm{Id})$. Sampling from the noise $\xi$ allows us to *sample* from this model (as in a GAN). The model's objective is to approximate the full distribution $p_{0|t}(x_0|x_t)$, i.e., $\hat{x}_\theta(t, X_t, \xi) \stackrel{d}{=} X_0|X_t, t$ and $\xi \sim \mathcal{N}(0, \mathrm{Id})$. Whenever the model $\hat{x}_\theta(t, X_t, \xi)$ is used, we refer to such a method as *distributional*. Using this model allows us to sample from diffusion models with fewer steps than classical diffusion models, leveraging (5).

**Conditional Generalized Energy Score.** We base our training loss on *generalized energy scores* as introduced in Section 2.2. For each $t, x_0, x_t$, we consider a *conditional*

generalized energy score with $\beta \in (0, 2]$ and $\lambda \in [0, 1]$

$$S_{\lambda,\beta}(p^\theta_{0|t}(\cdot|x_t), x_0) = \frac{\lambda}{2}\mathbb{E}_{p^\theta_{0|t}(\cdot|x_t) \otimes p^\theta_{0|t}(\cdot|x_t)}[\|X - X'\|^\beta]$$
$$- \mathbb{E}_{p^\theta_{0|t}(\cdot|x_t)}[\|X - x_0\|^\beta]. \quad (12)$$

The conditional energy score was first proposed for training conditional generative models by Bouchacourt et al. (2016), and its use in extrapolation and model estimation was established by Shen & Meinshausen (2024) in the case $\lambda = \beta = 1$. We emphasize that (12) is *conditional*, since it compares the conditional distribution $p^\theta_{0|t}(\cdot|x_t)$ to $x_0$. We recall that this scoring rule is strictly proper when $\lambda = 1$ and $\beta \in (0, 2)$. The hyperparameter $\lambda$ allows us to control the trade-off between diversity (first "interaction" term) and accuracy (second "confinement" term).

**Energy diffusion loss.** Our final expected loss integrates the *conditional generalized* energy score over both the noise level $t \in [0, 1]$, *and* the samples from the dataset, $X_0 \sim p_0$, together with noisy samples $X_t \sim p_{t|0}(\cdot|X_0)$ so that $(X_0, X_t) \sim p_{0,t}$. This gives the *energy diffusion loss*

$$\mathcal{L}(\theta) = -\int_0^1 w_t \mathbb{E}_{p_{0,t}}\left[S_{\lambda,\beta}(p^\theta_{0|t}(\cdot|X_t), X_0)\right] \mathrm{d}t, \quad (13)$$

where $w_t$ is a user-defined weighting function. The loss (13) has the following remarkable property: when we set $\beta \to 2$ and $\lambda \to 0$, we recover the classical regression diffusion loss (6). When $\beta \in (0, 2)$ and $\lambda = 1$ we obtain an (integrated) strictly proper expected *conditional* energy score. By selecting $\lambda \in (0, 1)$ and $\beta \in (0, 2]$, we can interpolate between these two cases. Our approach shares with (Bouchacourt et al., 2016; Gritsenko et al., 2020; Chen et al., 2024b; Pacchiardi et al., 2024; Shen & Meinshausen, 2024) the conditional energy score (12), but differs in that these earlier works learned only a *single* conditional distribution, whereas we integrate over many different noise levels.

**Empirical energy diffusion loss.** For samples $\{X_0^i\}_{i=1}^n \stackrel{\text{i.i.d.}}{\sim} p_0$ from the training set and corresponding noise levels $\{t_i\}_{i=1}^n \stackrel{\text{i.i.d.}}{\sim} \mathcal{U}[0, 1]$, we sample noisy data points $X_{t_i}^i|X_0^i, t_i \sim p_{t_i|0}(\cdot|X_0^i)$ from the forward diffusion process (2) with noise schedule (3). Then, for each pair $(X_0^i, X_{t_i}^i)$, we sample $m$ Gaussian noise samples $\xi_i^j \sim \mathcal{N}(0, \mathrm{Id})$, $j \in [m]$ which allows us to compute $\hat{x}_\theta(t_i, X_{t_i}, \xi_i^j)$. We then obtain the following empirical energy diffusion loss

$$\mathcal{L}_{n,m}(\theta) = \frac{1}{nm}\sum_{i,j=1}^{n,m} w_{t_i}\left[\left\|X_0^i - \hat{x}_\theta(t_i, X_{t_i}, \xi_i^j)\right\|^\beta \quad (14)\right.$$

$$\left. \frac{-\lambda}{2(m-1)}\sum_{j' \neq j}^m \left\|\hat{x}_\theta(t_i, X_{t_i}, \xi_i^j) - \hat{x}_\theta(t_i, X_{t_i}, \xi_i^{j'})\right\|^\beta\right].$$

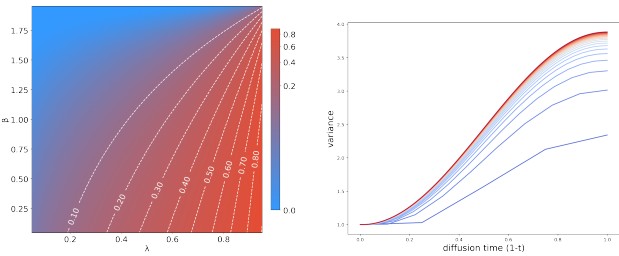

*Figure 1.* Left: reduction factor $f(\lambda, \beta)$ as a function of $\beta \in [0, 2]$ and $\lambda \in [0, 1]$. Right: evolution of the covariance during the sampling of distributional diffusion model for a Gaussian target $p_0 = \mathcal{N}(0, 4\mathrm{Id})$ for 20 of $N \in \{5, 10, \ldots, 95, 100\}$, $\lambda = 0.5$. As $N$ increases, the output variance gets closer to 4.

In Section 4, we discuss an alternative training loss using the *joint generalized* energy score which would focus on $p_{0,t}^\theta$ and show that its empirical version suffers from much higher variance. See also Appendix D for a longer discussion.

**Kernel diffusion loss.** Similarly, we can define a *kernel diffusion loss* by replacing $\|x - x'\|^\beta$ with a characteristic kernel $-k(x, x')$, such as (9a) or (9b), in (12) and (13). As for the energy score, we can recover the diffusion loss (6) for $k_{\mathtt{imq}}$ and $k_{\mathtt{rbf}}$ by identifying the leading terms in the Taylor expansion of these kernels. We refer the reader to Section 4.3 for more details.

**Architecture and training.** Unlike standard diffusion models, the neural network $\hat{x}_\theta(t, x_t, \xi)$ takes not only $t$ and $x_t$ as input, but also $\xi \sim \mathcal{N}(0, \mathrm{Id})$ to obtain an approximate sample from $p_{0|t}(x_0|x_t)$. For simplicity, in image generation, we concatenate $[x_t, \xi]$ along the channel dimension without modifying the rest of the architecture, see Appendix I for more details, and Algorithm 1 for the full training algorithm.

---

**Algorithm 1** Distributional Diffusion Model (training)

**Require:** $M$ training steps, schedule $(\alpha_t, \sigma_t)$, distribution $p_0$, weights $\theta_0$, batch size $n$, population size $m$
  **for** $k = 1 : M$ **do**
    Sample $t_i \sim \mathrm{Unif}([0, 1])$ for $i \in [n]$
    Sample $X_0^i \overset{\text{i.i.d.}}{\sim} p_0$ for $i \in [n]$
    Sample $X_{t_i}^i \sim p_{t_i|0}(\cdot|X_0^i)$ for $i \in [n]$ using (2)
    Sample $\xi_{i,j} \sim \mathcal{N}(0, \mathrm{Id})$ for $i \in [n], j \in [m]$
    Set $\theta_k = \theta_{k-1} - \delta \nabla \mathcal{L}_{n,m}(\theta)|_{\theta_{k-1}}$ using (14)
  **end for**
**Return:** $\theta_M$

---

Once again, we emphasize that setting $\lambda = 0$ and $\beta = 2$ in Algorithm 1 recovers the classical diffusion model loss.

**Sampling.** Once we have trained our model $\hat{x}_\theta$, we generate samples using Algorithm 2. This procedure is very similar to DDIM (Song et al., 2021a). The only difference is that

$\hat{x}_\theta$ now outputs an approximate *sample* from $p_{0|t}(x_0|x_t)$ instead of an approximation of $\mathbb{E}[X_0|X_t = x_t]$.

---

**Algorithm 2** Distributional Diffusion Model (sampling)

**Require:** $\{t_k\}_{k=0}^N$ with $t_0 = 0 < \cdots < t_N = 1$, churn parameter $\varepsilon$
  Sample $X_{t_N} \sim \mathcal{N}(0, \mathrm{Id})$
  **for** $k \in \{N - 1, \ldots, 0\}$ **do**
    Sample $\xi \sim \mathcal{N}(0, \mathrm{Id})$
    Sample $Z \sim \mathcal{N}(0, \mathrm{Id})$
    Set $\hat{X}_0 = \hat{x}_\theta(t_{k+1}, X_{t_{k+1}}, \xi)$
    Compute $\mu_{t_k, t_{k+1}}(\hat{X}_0, X_{t_{k+1}}), \Sigma_{t_k, t_{k+1}}^{1/2}$ using (4)
    Set $X_{t_k} = \mu_{t_k, t_{k+1}}(\hat{X}_0, X_{t_{k+1}}) + \Sigma_{t_k, t_{k+1}}^{1/2} Z$
  **end for**
**Return:** $X_0$

---

## 4. Theoretical analysis

In this section, we provide some theoretical understanding of the generalized scoring rules introduced in Section 3.

### 4.1. Gaussian analysis

We shed more light on the two main hyperparameters of the generalized kernel score introduced in Section 2.2: (i) the kernel parameter ($\beta$ in the case of the energy score); (ii) the trade-off term $\lambda$. We focus here on the case of *generalized energy score* defined in (8).

**Proposition 4.1:** *Assume that $p = \mathcal{N}(\mu, \sigma^2 \mathrm{Id})$ for $\mu \in \mathbb{R}^d$ and $\sigma > 0$. Then, for any $\lambda \in [0, 1]$ and $\beta \in [0, 2]$, we have that $q_{\lambda,\beta}^\star = \mathcal{N}(\mu_\star, \sigma_\star^2 \mathrm{Id})$ maximizes $S_{\lambda,\beta}(p, q)$ defined by (8) among all Gaussian distributions $q$, with parameters*

$$\mu_\star = \mu, \qquad \sigma_\star^2 = (2\lambda^{-2/(2-\beta)} - 1)^{-1}\sigma^2.$$

A notable outcome of maximizing the generalized energy score is that $\mu^\star = \mu$ no matter the value of $\lambda$ and $\beta$. With $\lambda = 1$ and $\beta < 2$, we recover as expected the correct variance, since $S_{\lambda,\beta}$ is a strictly proper energy score. As soon as $\lambda < 1$, the variance $\sigma^2$ is underestimated by a factor $f(\lambda, \beta) = (2\lambda^{-2/(2-\beta)} - 1)^{-1}$; see Figure 1.

We now consider the effect of these hyperparameters when sampling from the corresponding generative model using Algorithm 2. For simplicity, we let the churn parameter $\varepsilon = 1$ but our analysis also applies to $\varepsilon \in [0, 1)$. We consider a Gaussian target $p_0 = \mathcal{N}(\mu, \sigma^2 \mathrm{Id})$. In that case, $p_{0|t}(x_0|x_t)$ is also Gaussian for any $t \in [0, 1]$. Therefore if this density were estimated using the generalized energy score, then the results of Proposition 4.1 apply. In particular, this implies that the induced estimate of the Gaussian distribution $p_{s|t}$ obtained by plugging our approximation of $p_{0|t}$ into (5)

would have the correct mean but an incorrect variance for $\lambda < 1$.

As expected, differences between classical diffusion models and distributional diffusion models arise when considering larger discretization stepsizes, where the Dirac approximation of $p_{0|t}(x_0|x_t)$ is no longer valid, see Figure 1.

## 4.2. Joint or conditional scoring rules

Given two unbiased estimators $\hat{A}$ and $\hat{B}$ of $A$ and $B$, we say that $A$ is easier to approximate than $B$ if $\mathrm{SNR}(\hat{A}) \geq \mathrm{SNR}(\hat{B})$ where, for an unbiased estimate $\hat{C}$, $\mathrm{SNR}(\hat{C}) := \mathbb{E}[\hat{C}]^2/\mathrm{Var}(\hat{C})$; i.e. the inverse relative variance of $\hat{C}$. We investigate here one alternative to the loss (13) and show that its empirical version exhibits lower SNR than the empirical version of (13) under the simplifying assumption that $\theta$ is chosen so that $p_{0|t}^\theta = p_{0|t}$. While the loss (13) leverages generalized energy score on the *conditional* distribution $p_{0|t}$, we can instead leverage the generalized energy score on the *joint* distribution $p_{0,t}$ to define a new loss

$$\mathcal{L}_{\text{joint}}(\theta) = -\int_0^1 w_t \mathbb{E}_{p_{0,t}} \left[ S_{\lambda,\beta}(p_{0|t}^\theta p_t, (X_0, X_t)) \right] dt, \tag{15}$$

where $p_{0|t}^\theta p_t$ denotes $p_{0|t}^\theta(x_0|x_t)p_t(x_t)$. In the case of the *conditional* loss (13), an empirical interaction estimate is

$$\mathcal{I}_{n,m} = \sum_{i=1}^n \sum_{j \neq j'}^m \frac{\lambda}{2n(m-1)} \Delta_{i,j}$$

where $\Delta_{i,j} = \|\hat{x}_\theta(t_i, X_{t_i}, \xi_i^j) - \hat{x}_\theta(t_i, X_{t_i}, \xi_i^{j'})\|^\beta$ where $X_{t_i} \sim p_{t_i}, \xi_i^j \sim \mathcal{N}(0, \text{Id})$ for all $i, j$. In the case of the *joint* loss (15), an empirical interaction estimate is

$$\mathcal{I}_{n,m,\text{joint}} = \sum_{i=1}^n \sum_{j \neq j'}^m \frac{\lambda}{2n(m-1)} \left[ \|X_{t_i}^j - X_{t_i}^{j'}\| + \Delta'_{i,j} \right],$$

where $\Delta'_{i,j} = \|\hat{x}_\theta(t_i, X_{t_i}^j, \xi_i^j) - \hat{x}_\theta(t_i, X_{t_i}^{j'}, \xi_i^{j'})\|^\beta$. Here we have $X_{t_i}^j \sim p_{t_i}, \xi_i^j \sim \mathcal{N}(0, \text{Id})$ for all $i, j$. We then ask whether $\mathrm{SNR}(\mathcal{I}_{n,m}) \geq \mathrm{SNR}(\mathcal{I}_{n,m,\text{joint}})$. Unfortunately, we cannot answer this in the general case. However, if $p_0 = \mathcal{N}(0, \sigma^2 \text{Id})$, and leveraging results from U-statistics, we have the following result.

**Proposition 4.2:** *Let $U \sim p_U$ and $V \sim p_V$ where $p_U \propto w_t/(1 + \frac{\alpha_t^2 \sigma^2}{\sigma_t^2})^{1/2}$ and $p_V \propto w_t(\sigma + (\alpha_t^2 \sigma^2 + \sigma_t^2)^{1/2})$ are two distributions on $[0, 1]$. Then, we have that*

$$\mathrm{SNR}(\mathcal{I}_n) := \lim_{m \to +\infty} \mathrm{SNR}(\mathcal{I}_{n,m}) = n\mathrm{SNR}(U),$$

$$\mathrm{SNR}(\mathcal{I}_{n,\text{joint}}) := \lim_{m \to +\infty} \mathrm{SNR}(\mathcal{I}_{n,m,\text{joint}}) = n\mathrm{SNR}(V).$$

In practice, we consider the sigmoid weighting scheme $w_t = (1 + \exp[b - \log(\alpha_t^2/\sigma_t^2)])^{-1}$ (Kingma et al., 2021;

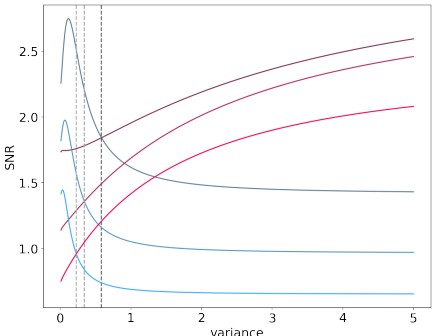

*Figure 2.* $\mathrm{SNR}(\mathcal{I}_n)$ (red) and $\mathrm{SNR}(\mathcal{I}_{n,\text{joint}})$ (blue) w.r.t. $\sigma^2$ ($x$-axis) for 3 possible bias levels $b \in \{0, 1, 2\}$ (from dark to light). Vertically, we plot $(\sigma_b^\star)^2$ such that for the bias $b$, we have $\mathrm{SNR}(\mathcal{I}_n) \geq \mathrm{SNR}(\mathcal{I}_{n,\text{joint}})$ for $\sigma \geq \sigma_b^\star$.

Hoogeboom et al., 2023), where $b \in \mathbb{R}$ is some bias. The SNR of $U, V$ can easily be computed and we observe that, across a large range of values of $\sigma$ and bias values $b$, we indeed have $\mathrm{SNR}(\mathcal{I}_{n,m}) \geq \mathrm{SNR}(\mathcal{I}_{n,m,\text{joint}})$ in the large $m$ limit, see Figure 2.

## 4.3. From kernel scores to diffusion losses

We have shown in eq. (11) of Section 2.2 that the energy diffusion loss (14) recovers the diffusion loss (6) when $\beta \to 2$. In that case, we say that the scoring rule is *diffusion compatible*. A natural question to ask is whether other scoring rules are diffusion compatible: i.e., given $\rho_c$ with $c \in \mathbb{R}$ a hyperparameter of the kernel, that there exists $c^\star \in [-\infty, +\infty]$ and $f : \mathbb{R} \to \mathbb{R}$ such that

$$\lim_{c \to c^\star} f(c) D_{\rho_c}(p, q) = \|\mathbb{E}_p[X] - \mathbb{E}_q[X]\|^2.$$

The following result shows that $k_{\text{imq}}$ and $k_{\text{rbf}}$ can also recover the diffusion loss.

**Proposition 4.3:** *Assume that $\rho = -k$ with $k$ given by (9a) or (9b). Then the scoring rule is diffusion compatible.*

This proposition justifies using scoring rules other than the energy one, which also allow recovering diffusion models in the limit. We compare the performance of these different kernels in Section 6.2. Our experiments suggest that diffusion compatibility, as demonstrated by the energy score, $k_{\text{imq}}$, and $k_{\text{rbf}}$, is sufficient for defining a loss function that leads to high-quality sample generation when used to train a model. However, we have identified other kernels, such as $k_{\text{exp}}$, that do not satisfy the diffusion compatibility requirement but still result in models with desirable properties.

## 5. Related Work

**Accelerated diffusion models.** Many strategies accelerate diffusion models, broadly classifiable as distillation-

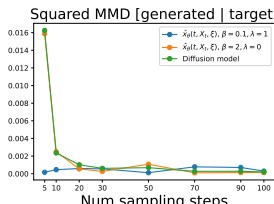
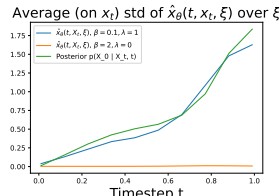

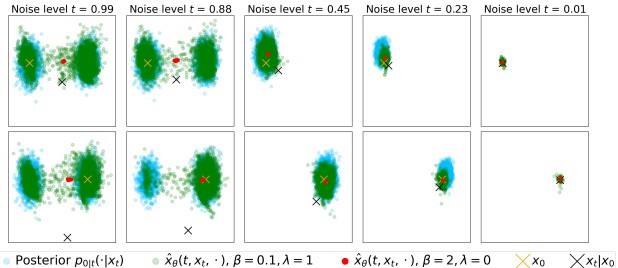

*Figure 3.* Left: Squared MMD between target distribution and sampled data according to different models, $x$-axis denotes the number of sampling steps. Right: average standard deviation of samples $X_0|X_t$ produced by either true posterior distribution $p_{0|t}$ or the models $\hat{x}_\theta(t, X_t, \xi)$, $x$-axis is the timestep $t$.

*Figure 4.* Samples from true posterior $p_{0|t}(\cdot|x_t)$ (light blue) for a sample $x_t$ (black cross) from $p_{t|0}(x_t|x_0)$ for a specific $x_0$ (orange cross) and samples $\hat{x}_\theta(t, x_t, \xi) \sim p_{0|t}^\theta(\cdot|x_t)$ (green dots) for $\beta = 0.1, \lambda = 1$. Top/bottom row: $x_0$ is the mean of the left/right Gaussian. Samples from $\hat{x}_\theta(t, x_t, \xi)$ (red) with $\beta = 2, \lambda = 0$ concentrate around $\mathbb{E}[X_0|x_t]$ as expected.

or sampling-based. Distillation methods (Luhman & Luhman, 2021; Salimans & Ho, 2022; Luo, 2023; Salimans et al., 2024; Dieleman, 2024; Liu et al., 2023; Meng et al., 2023; Song et al., 2023; Franceschi et al., 2024; Huang et al., 2024; Sauer et al., 2025) train a student model to mimic a teacher diffusion model, leveraging consistency losses (Song et al., 2023), adversarial losses (Franceschi et al., 2024; Xu et al., 2024; Sauer et al., 2025), or noise coupling (Huang et al., 2024). See (Dieleman, 2024) for a detailed discussion. While distillation is prevalent, other work focuses on improved samplers for larger step sizes (Jolicoeur-Martineau et al., 2021; Lu et al., 2022; Zheng et al., 2023). Our work differs from both, neither training a student nor proposing new samplers.

**Discrepancy and diffusion.** We modify the diffusion model training loss to learn the conditional distribution $p_{0|t}(x_0|x_t)$. The importance of approximating the covariance of this distribution was noted by Nichol & Dhariwal (2021) and exploited in (Ho et al., 2020; Nichol & Dhariwal, 2021; Bao et al., 2022a;b; Ou et al., 2024). Closest to our approach is (Xiao et al., 2022), which uses a GAN to learn $p_{0|t}(x_0|x_t)$ for fewer sampling steps. While sharing the same motivation, our method avoids adversarial training and discriminator training, offering a simple loss modification that encompasses standard diffusion models. Other loss modifications, like using the $\ell_1$ loss (Chen et al., 2021; Saharia et al., 2022a), aim to improve output quality, not reduce sampling steps. Galashov et al. (2025) can be seen as dual to our work: they learn discriminative kernel features with a flow at different noise levels, while we focus solely on learning a generator.

**Energy distances, MMD, and generative modeling.** The MMD (Gretton et al., 2012), of which energy distances (Székely & Rizzo, 2013; Rizzo & Székely, 2016) are a special case (Sejdinovic et al., 2013), has been widely used as a distributional loss in generative modeling. GANs have used MMDs with fixed kernels as critics to distinguish generated from reference samples (Li et al., 2015; Dziugaite et al., 2015; Unterthiner et al., 2018). MMDs (Li et al., 2017;

Bińkowski et al., 2018) and energy distances (Liu, 2017; Bellemare et al., 2017; Salimans et al., 2018) have also been defined on adversarially trained discriminative neural net features. The conditional generalized energy score of (12) has been used in learning conditional distributions by the DISCONet approach of Bouchacourt et al. (2016), and by the engression approach of Shen & Meinshausen (2024) (which corresponds to the special case of $\lambda, \beta = 1$). Recently, energy distances have been applied to speech synthesis (Gritsenko et al., 2020), normalizing flows (Si et al., 2023), neural SDEs (Issa et al., 2024), and other generative models (Chen et al., 2024b; Pacchiardi et al., 2024).

To our knowledge, such losses have not previously been used to train diffusion models. It is possible, however, to use an adaptation of (Galashov et al., 2025) in combination with the approach of Bouchacourt et al. (2016); Shen & Meinshausen (2024), in order to incorporate conditional GAN-style generation into the reverse process of a diffusion model. This idea was proposed by X. Shen (personal communication, 6th November 2024) in an open discussion following a presentation at Google Deepmind, as a future research direction of interest. The main idea of Galashov et al. (2025) is retained, namely to use a standard forward diffusion process, and a distributional loss for the reverse process: however the adaptive-kernel MMD loss for the reverse process is replaced with a fixed-kernel energy score; and this score is used to train a sequence of conditional GAN generators to approximate $p(x_{t_{k-1}}|x_{t_k})$, rather than using particle diffusion directly. The approach indeed represents a promising line of work (since developed in Shen et al., 2025) for discretizing the reverse diffusion process, as distinct from the DDIM-style approach adopted in Section 3. The approach can further be understood as a non-adversarial formulation of Cheng et al. (2024), which uses an adaptive conditional GAN critic in place of the fixed-kernel energy score.

# 6. Experiments

In this section, we validate the performance of our approach in 2D, image generation, and robotics settings. The main experiments are presented here, while additional experiments and ablations are given in Appendix K. All the experimental details are described in Appendix I, while Appendix J analyzes computational complexity.

## 6.1. 2D experiments

Consider a target given by a mixture of two Gaussians, i.e. $p_0 = 0.5\mathcal{N}(\mu_1, \sigma^2 \mathrm{Id}) + 0.5\mathcal{N}(\mu_2, \sigma^2 \mathrm{Id})$, where $\mu_1 = (3, 3), \mu_2 = (-3, 3)$ and $\sigma = 0.5$. We train an unconditional *distributional* model $\hat{x}_\theta$ with $\beta = 0.1$ and $\lambda = 1$ using Algorithm 1. Additionally, we train a baseline unconditional diffusion model by optimizing (6) and the model $\hat{x}_\theta$ with $\beta = 2$ and $\lambda = 0$ using Algorithm 1, i.e. a classical diffusion model using the same architecture as the *distributional* models. More details are provided in Appendix I.1.

To measure the quality of the samples produced, we use the MMD squared given by $D_\rho$ (10) with $\rho(x, x') = -k_{\mathrm{rbf}}(x, x')$ (9b) for $\sigma = 1$. In Figure 3 (left), we observe that with few denoising steps, the *distributional* variant has a smaller $D_\rho$ compared to other models, and achieves similar performance as the diffusion model using a large number of steps. In Figure 3 (right), for each $X_t^i$, we produce 8 samples $\xi \sim \mathcal{N}(0, \mathrm{Id})$ and we compute the standard deviation (std) of $\hat{x}_\theta(t, x_t^i, \xi)$ over $\xi$. We then average the std over all the $x_t^i$. While classical diffusion models cannot model the variance of the posterior, *distributional* diffusion models have std close to that of the true posterior.

To further our analysis, we visualize the samples from the *distributional* model in Figure 4. For given $x_t$ values, we sample from the true posterior $p_{0|t}(x_0|x_t)$ which is available in closed form. We also sample $\hat{x}_\theta(t, x_t, \xi) \sim p_{0|t}^\theta(\cdot|X_t^i)$ where $\xi \sim \mathcal{N}(0, \mathrm{Id})$ for a model trained using $\beta = 0.1, \lambda = 1$, showing that our generating network is able to learn a good approximation of the posterior. We also display samples from $\hat{x}_\theta(t, x_t, \xi)$ trained with $\beta = 2, \lambda = 0$ which are concentrated around $\mathbb{E}[X_0|x_t]$, as expected.

## 6.2. Image experiments

**Main results.** We train conditional pixel-space models on CIFAR-10 (32x32x3) and on CelebA (64x64x3), as well as unconditional pixel-space models on LSUN Bedrooms (64x64x3). We further use an autoencoder trained on CelebA-HQ (256x256x3) producing latent codes of shape 64x64x3, which are then used to build latent unconditional models. For each dataset, we train a diffusion model by optimizing (6) as well as a *distributional* model (Algorithm 1) for different $\lambda \in [0, 1]$ and $\beta \in [0.001, 2]$. Models are evaluated using the FID score (Heusel et al., 2017). See Ap-

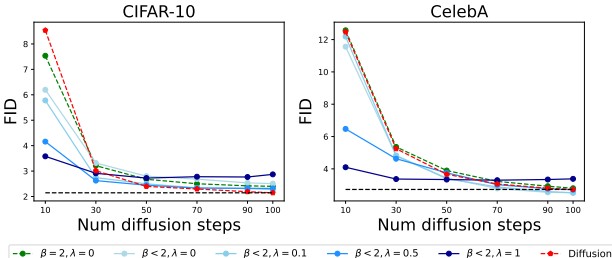

*Figure 5.* Conditional image generation. $x$-axis denotes number of denoising steps, $y$-axis represents FID. Black dashed line denotes the performance of diffusion model at 100 steps.

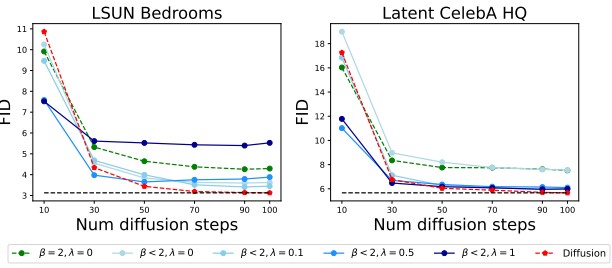

*Figure 6.* Unconditional image generation. $x$-axis denotes number of denoising steps, $y$-axis represents FID. Black dashed line denotes the performance of the diffusion model at 100 steps.

pendix I.2 for details.

In Figure 5, we show results of conditional image generation on CIFAR-10 and CelebA, and in Figure 6 results of unconditional image generation on LSUN Bedrooms and latent CelebA-HQ. Whenever we report performance of *distributional* model and write $\beta < 2$, we select a parameter $\beta$ given one fixed $\lambda$ which minimizes the FID. More detailed results are given in Appendix K.2, which include FIDs for different numbers of steps for every combination of $(\beta, \lambda)$.

**Main takeaways.** The results suggest that whenever the number of diffusion steps is low, then *distributional* models with $\lambda = 1, \beta < 2$ achieve better performance than classical diffusion models. This confirms that the diffusion model approximation is poor in the "few steps" regime, and can be improved with better modeling of $p_{0|t}$ as explained in Section 2.1. As the number of diffusion steps increases, the Dirac approximation becomes more accurate and diffusion models achieve the best performance. In that scenario, we observe that *distributional* models with $\lambda = 1, \beta < 2$ yield worse performance. Our hypothesis is that since *distributional* models $\hat{x}_\theta$ only approximate the posterior distribution $p_{0|t}$, this could lead to an increased amount of noise at sampling time and consequently to error accumulation. However, we observe that *distributional* model variants with

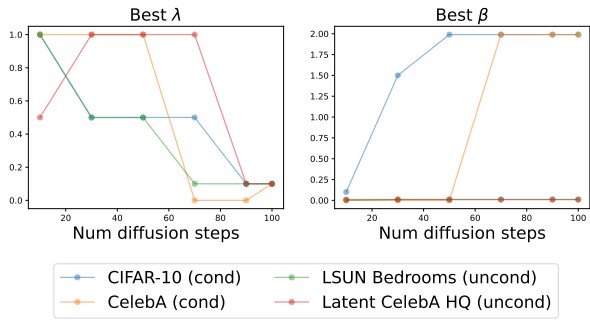

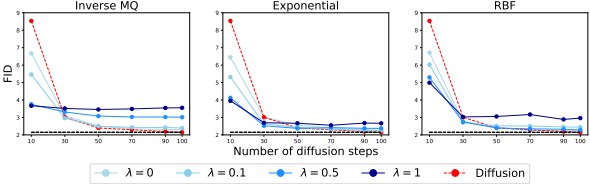

Figure 8. Conditional generation on CIFAR-10 with different kernels. $x$-axis is the number of diffusion steps. $y$-axis is the FID.

Figure 7. Parameters $\lambda$ and $\beta$ minimizing the FID for a given number of diffusion steps. $x$-axis indicates the number of diffusion steps and legend color indicates the dataset.

$\lambda \in (0, 1)$ still yield good performance as the number of diffusion steps increases. This highlights the trade-off between *distributional* and diffusion models controlled by the parameter $\lambda$. As argued in Section 4.1, $\lambda < 1$ leads to underestimation of the target variance which, for denoising diffusions, implies that the variance of $\hat{x}_\theta$ is more concentrated around $p_{0|t}$ than would occur for posterior samples. Remarkably, we also found that using $\beta < 2, \lambda = 0$ led to comparable or even slightly better performance compared to classical diffusion models. This suggests that it could be valuable to train diffusion models using a different loss than Mean Square Error (MSE) regression.

**Conditional vs unconditional generation.** We observe slightly different behavior of *distributional* models in the conditional image generation case (Figure 5), and in the unconditional case (Figure 6). For a small number of diffusion steps, we notice that the gap in performance between the model trained with $\lambda = 1, \beta < 2$ and a diffusion model, is much smaller in the unconditional case (1.5x) compared to the conditional one (3x). In Figure 7 we visualize the best parameters $\lambda, \beta$ selected to achieve the lowest FID, as a function of diffusion steps. More detailed results are presented in Appendix K.2. We observe that overall $\lambda$ follows a downward trend, decreasing as the number of steps increases. However, $\beta$ behaves differently in each setting, increasing monotonically as a function of diffusion steps in the conditional case, while remaining at a minimal value in the unconditional case. We hypothesise that this is because learning unconditional distributions is a much harder task since the data is spread out more, leading to a much higher variance in the distribution of $\hat{x}_\theta$.

**Using different kernels.** As discussed in Section 2.2 and in Section 3, we can also use the kernel diffusion loss. We train $\hat{x}_\theta$ with kernels (9a), (9b), (9c), varying kernel parameters and $\lambda \in [0, 1]$; see Figure 8 for results on conditional image generation for CIFAR-10. We observe similar behavior to the energy diffusion loss, achieving the best results when

$\lambda = 1$ for few diffusion steps and better performance with $\lambda < 1$ when number of steps increases; see Appendix K.2 for detailed results.

**Comparisons to distillation and numerical solvers.** In Appendix K.3, we compare our approach with the *DPM-solver++* method Lu et al. (2022) and the state-of-the-art, multi-step distillation method, moment-matching distillation (Salimans et al., 2024). Overall, our approach achieves results competitive with multistep distillation (where we compared different numbers of steps) and outperforms *DPM-solver++* for more than 8 NFEs. The performance of *DPM-solver++* degrades as the number of steps increases, which is expected due to its numerical instability.

Our approach offers two compelling benefits over multistep distillation. First, it does not rely on distillation, eliminating the need to train a large (and slow) teacher model. Second, it does not require specifying additional sampler hyperparameters during training. Combining our distributional approach with a modern distillation method is an interesting direction for future research.

### 6.3. Robotics experiments

**Experimental setup.** We demonstrate here the effectiveness of our *distributional* diffusion models for robotics applications. We experiment with diffusion policies on the Libero (Liu et al., 2024) benchmark, a life-long learning benchmark that consists of 130 language-conditioned robotic manipulation tasks. In our experiments, we used 4 Libero suites with 10 tasks in each: Libero-Long, Libero-Goal, Libero-Object, and Libero-Spatial. Our multi-task diffusion policy is conditioned on the encoded visual and proprioceptive observations, and the task descriptions and generates a chunk of 8 7-dimensional actions to execute; see Appendix I.3 for additional details on the architecture.

**Distributional diffusion model.** We train the *distributional* diffusion model with Algorithm 1 using the energy diffusion loss with population size $m = 16$ and varying $\beta$ and $\lambda$. We use a diffusion model to model a sequence of 8 actions $a = (a_1, \ldots, a_8)$, where $a_i \in \mathbb{R}^7$. We use the noise $\xi \in \mathbb{R}^{8,2}$ which we concatenate with $a$ on the last dimension. At sampling time, we follow Algorithm 2 to sample a sequence

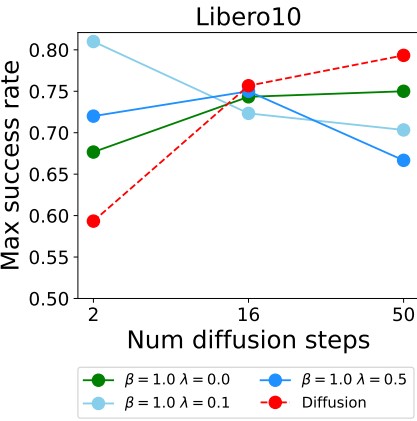

*Figure 9.* Performance in Libero10 as function of diffusion steps and $\lambda$. We plot the maximum success rate from best checkpoint during training. $\lambda = 0$ works best when more diffusion steps are used, and $\lambda > 0$ gives best results when fewest diffusion steps are used. $\lambda > 0$ also reduces the performance by a little for 50 diffusion steps (see 0.5 vs 0.0).

of actions from a diffusion policy and we execute it. We sweep over a number $\{2, 16, 50\}$ of diffusion steps.

**Results.** For the main result, we focus on Libero-Long, the most challenging suite with 10 tasks that features long-horizon tasks with diverse object interactions. Similar to the results on image experiments, $\lambda = 0$ works best when more diffusion steps (50) are used during evaluation. Using $\lambda > 0$ gives best results when fewer diffusion steps are used $(2, 16)$. We note that the performance slightly deteriorates with $\lambda > 0$ when the number of diffusion steps is large, see Figure 9. See Figure 19 for similar results on other Libero suites.

## 7. Discussion

We present a novel approach for training diffusion models, by learning the full conditional distributions $p_{0|t}$ using scoring rules. This approach achieves good performance in image generation and robotics tasks, outperforming standard diffusion models in the "few-step" regime with minimal degradation of output quality. Our experimental results further suggest that standard diffusion models can also be trained with objectives other than classical MSE regression.

So far, our generative networks approximating $p_{0|t}$ incorporate the noise $\xi$ only by concatenating it to the $x_t$ along the channel dimension. We believe that other architectures could significantly improve sample quality, and will explore this in future work. Another promising direction is in learning the kernel $\rho$ and the parameter $\lambda$ to maximize generative performance in the few-step regime. Combining our approach with powerful distillation techniques such as

(Salimans et al., 2024) is a further avenue for future work.

Finally kernel scores allow for more general domains than $\mathbb{R}^d$, by way of characteristic kernels on groups (e.g. the group SO(3) of 3d rotations) and semigroups ($d$-dimensional histograms) (Fukumizu et al., 2008); and graphs (Vishwanathan et al., 2010). It is thus possible to extend our methodology to these scenarios.

## Impact Statement

This paper presents work whose goal is to advance the field of Machine Learning. There are many potential societal consequences of our work, none which we feel must be specifically highlighted here.

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

## Organization of the Appendix

The appendix is organized as follows. In Appendix A, we present a notation table, to help the reader. In Appendix B, we prove that diffusion compatible scoring rules can be linked to diffusion model losses as claimed in Section 2.2. In Appendix C, we prove the results regarding the minimization of generalized expected energy score in a Gaussian setting presented in Section 4.1. In Appendix D, we expand on the discussion of Section 4.2 regarding joint or conditional scoring rules. In Appendix E, we present the results of Section 4.3 regarding diffusion-compatible kernels. In Appendix F, we present a new perspective on the DDIM updates derived in (Song et al., 2021a). More precisely, we show that these updates can be recovered using a Stochastic Differential Equation perspective. This justifies the form of the DDIM mean and covariance updates presented in (4). We present some pseudocode to compute our loss function in Appendix G. Experimental details are provided in Appendix I. We present a computational complexity analysis in Appendix J. We present an extended related work in Appendix H. In Appendix K, we present additional experimental results including ablation studies.

## A. Notation

We recall that a kernel defined on a space $\mathsf{X}$ is a symmetric function $k : \mathsf{X} \times \mathsf{X} \to \mathbb{R}$, i.e., for any $x, y \in \mathsf{X}$, $k(x,y) = k(y,x)$. A kernel $k$ is said to be *positive semi-definite* if for any $n \in \mathbb{N}$, $\{x_1, \ldots, x_n\} \in \mathsf{X}^n$ and $\{c_1, \ldots, c_n\} \in \mathbb{R}^n$ we have

$$\sum_{i=1}^{n} \sum_{j=1}^{n} c_i c_j k(x_i, x_j) \geq 0. \tag{16}$$

The kernel $k$ is *positive definite* if equality in (16) occurs if and only $c_i = 0$ for any $i \in \{1, \ldots, n\}$. We refer the reader to (Berlinet & Thomas-Agnan, 2004; Steinwart & Christmann, 2008; Gretton, 2013) for an overview on kernel theory.

| Name | Notation | Definition | Appeared 1st |
|------|----------|------------|--------------|
| · | $\rho$ | Continuous negative definite kernel | Sec 2.2 |
| · | $k$ | Continuous positive definite kernel | Sec 2.2 |
| Scoring rule | $S(p, y)$ | · | Sec 2.2 |
| Expected score | $S(p, q)$ | $S(p, q) = \mathbb{E}_q[S(p, Y)]$ | Sec 2.2 |
| Kernel score | $S_\rho(p, y)$ | $S_\rho(p, y) = \frac{1}{2}\mathbb{E}[\rho(X, X')] - \mathbb{E}[\rho(X, y)]$ | Sec 2.2 |
| Energy score | $S_\beta(p, y)$ | $S_\beta(p, y) = S_\rho(p, y)$ with $\rho(x, x') = \|x - x'\|^\beta$ | Sec 2.2 |
| Expected Kernel score | $S_\rho(p, q)$ | $S_\rho(p, q) = \mathbb{E}_q[S_\rho(p, Y)]$ | Sec 2.2 |
| Expected Energy score | $S_\beta(p, q)$ | $S_\beta(p, q) = S_\rho(p, q)$ with $\rho(x, x') = \|x - x'\|^\beta$ | Sec 2.2 |
| Generalized (Gen.) Kernel score | $S_{\lambda,\rho}(p, y)$ | $S_{\lambda,\rho}(p, y) = \frac{\lambda}{2}\mathbb{E}_{p \otimes p}[\rho(X, X')] - \mathbb{E}[\rho(X, y)]$ | Sec 2.2 |
| Generalized Energy score | $S_{\lambda,\beta}(p, y)$ | $S_{\lambda,\beta}(p, y) = S_{\lambda,\rho}(p, y)$ with $\rho(x, x') = \|x - x'\|^\beta$ | Sec 2.2 |
| Expected Gen. Kernel score | $S_{\lambda,\rho}(p, q)$ | $S_{\lambda,\rho}(p, q) = \mathbb{E}_q[S_{\lambda,\rho}(p, Y)]$ | Sec 2.2 |
| Expected Gen. Energy score | $S_{\lambda,\beta}(p, q)$ | $S_{\lambda,\beta}(p, q) = S_{\lambda,\rho}(p, q)$ with $\rho(x, x') = \|x - x'\|^\beta$ | Sec 2.2 |
| Squared MMD | $D_\rho(p, q)$ | $D_\rho(p, q) = S_\rho(q, q) - S_\rho(p, q)$ | Sec 2.2 |
| Energy Distance | $D_\beta(p, q)$ | $D_\beta(p, q) = D_\rho(p, q)$ with $\rho(x, x') = \|x - x'\|^\beta$ | Sec 2.2 |
| Diffusion Loss | $\mathcal{L}_{\text{diff}}(\theta)$ | $\mathcal{L}_{\text{diff}}(\theta) = \int_0^1 w_t \mathbb{E}_{p_{0,t}}\left[\|X_0 - \hat{x}_\theta(t, X_t)\|^2\right] \mathrm{d}t$ | Sec 2.1 |
| Energy Diffusion Loss | $\mathcal{L}(\theta)$ | $\mathcal{L}(\theta) = -\int_0^1 w_t \mathbb{E}_{p_{0,t}}\left[S_{\lambda,\beta}(p_{0|t}^\theta(\cdot|X_t), X_0)\right] \mathrm{d}t$ | Sec 3 |
| Kernel Diffusion Loss | $\mathcal{L}(\theta)$ | $\mathcal{L}(\theta) = -\int_0^1 w_t \mathbb{E}_{p_{0,t}}\left[S_{\lambda,\rho}(p_{0|t}^\theta(\cdot|X_t), X_0)\right] \mathrm{d}t$ | Sec 3 |
| Joint Diffusion Energy Loss | $\mathcal{L}_{\text{joint}}(\theta)$ | $\mathcal{L}_{\text{joint}}(\theta) = -\int_0^1 w_t \mathbb{E}_{p_{0,t}}\left[S_{\lambda,\beta}(p_{0|t}^\theta p_t, (X_0, X_t))\right] \mathrm{d}t$ | Sec 4.2 |

## B. Correspondence between Discrepancy and Diffusion Loss

In this section, we demonstrate the connection between diffusion-compatible scoring rules and diffusion model losses, as stated in Section 2.2. More precisely, we prove the following result.

**Proposition B.1:** *Assume that $D_\rho(p, q) = \|\mathbb{E}_p[X] - \mathbb{E}_q[X]\|^2$ and that $p = \delta_{\hat{x}_\theta(t, X_t)}$ and $q = p_{0|t}(\cdot|X_t)$. Then, we have*

$$\mathbb{E}_{p_t}[D_\rho(p, q)] = \mathbb{E}_{p_{0,t}}[\|X_0 - \hat{x}_\theta(t, X_t)\|^2] - \mathbb{E}_{p_{0,t}}[\|X_0 - \mathbb{E}[X_0|X_t]\|^2],$$

*where the second term on the r.h.s. is independent of $\theta$.*

*Proof.* First we have

$$\begin{aligned}
\mathbb{E}_{p_t}[D_\rho(p, q)] &= \mathbb{E}_{p_t}[\|\hat{x}_\theta(t, X_t) - \mathbb{E}[X_0|X_t]\|^2] \\
&= \mathbb{E}_{p_{0,t}}[\|\hat{x}_\theta(t, X_t) - X_0 + X_0 - \mathbb{E}[X_0|X_t]\|^2] \\
&= \mathbb{E}_{p_{0,t}}[\|\hat{x}_\theta(t, X_t) - X_0\|^2] + \mathbb{E}_{p_{0,t}}[\|X_0 - \mathbb{E}[X_0|X_t]\|^2] + 2\mathbb{E}_{p_{0,t}}[\langle \hat{x}_\theta(t, X_t) - X_0, X_0 - \mathbb{E}[X_0|X_t]\rangle] \\
&= \mathbb{E}_{p_{0,t}}[\|\hat{x}_\theta(t, X_t) - X_0\|^2] + \mathbb{E}_{p_{0,t}}[\|X_0 - \mathbb{E}[X_0|X_t]\|^2] \\
&\quad + 2\mathbb{E}_{p_{0,t}}[\langle \hat{x}_\theta(t, X_t), X_0 - \mathbb{E}[X_0|X_t]\rangle] - 2\mathbb{E}_{p_{0,t}}[\langle X_0, X_0 - \mathbb{E}[X_0|X_t]\rangle] \\
&= \mathbb{E}_{p_{0,t}}[\|\hat{x}_\theta(t, X_t) - X_0\|^2] + \mathbb{E}_{p_{0,t}}[\|X_0 - \mathbb{E}[X_0|X_t]\|^2] \\
&\quad - 2\mathbb{E}_{p_{0,t}}[\langle X_0, X_0 - \mathbb{E}[X_0|X_t]\rangle].
\end{aligned} \tag{17}$$

In addition, we have

$$\mathbb{E}_{p_{0,t}}[\|X_0 - \mathbb{E}[X_0|X_t]\|^2] = \mathbb{E}_{p_0}[\|X_0\|^2] - \mathbb{E}_{p_0}[\|\mathbb{E}[X_0|X_t]\|^2]. \tag{18}$$

Similarly, we have

$$\mathbb{E}_{p_{0,t}}[\langle X_0, X_0 - \mathbb{E}[X_0|X_t]\rangle] = \mathbb{E}_{p_0}[\|X_0\|^2] - \mathbb{E}_{p_0}[\|\mathbb{E}[X_0|X_t]\|^2]. \tag{19}$$

Combining (17), (18) and (19), we get

$$\mathbb{E}_{p_t}[D_\rho(p, q)] = \mathbb{E}_{p_{0,t}}[\|X_0 - \hat{x}_\theta(t, X_t)\|^2] - \mathbb{E}_{p_{0,t}}[\|X_0 - \mathbb{E}[X_0|X_t]\|^2],$$

which concludes the proof. $\square$

## C. Generalized Energy Score for Gaussian Targets

We first investigate the properties of the generalized energy score for a single Gaussian target in Appendix C.1 (Proposition 4.1) and then apply these results to diffusion models in Appendix C.2

### C.1. Learning a Gaussian distribution

For any $\beta \in (0, 2]$ and $\lambda \in [0, 1]$ we recall that $S_{\lambda, \beta}$ is the generalized expected energy score given for any distributions $p, q$ on $\mathbb{R}^d$ with $\beta$ moments by

$$S_{\beta, \lambda}(p, q) = -\mathbb{E}_{(X,Y) \sim p \otimes q}[\|X - Y\|^\beta] + \frac{\lambda}{2}\mathbb{E}_{(X, X') \sim p \otimes p}[\|X - X'\|^\beta].$$

We refer the reader to Appendix A for a remainder on the notation used throughout the paper. For any $\mu \in \mathbb{R}^d$ and $\sigma > 0$, let $p_{\mu, \sigma} = \mathcal{N}(\mu, \sigma^2\text{Id})$ and denote by $\mathfrak{G}$ the space of Gaussian distributions with scalar covariance, i.e.,

$$\mathfrak{G} = \{\mathcal{N}(\nu, \gamma^2\text{Id}) , \nu \in \mathbb{R}^d, \gamma \geq 0\}.$$

We consider the following minimization problem: for any $\mu \in \mathbb{R}^d$ and $\sigma > 0$ solve

$$p_{\mu_\star, \sigma_\star} = \text{argmax}_{q \in \mathfrak{G}} S_{\beta, \lambda}(q, p_{\mu, \sigma}).$$

When $\lambda = 1$ and $\beta \in (0, 2)$, $S_{\lambda, \beta}(p, q)$ is a proper scoring rule (Rizzo & Székely, 2016; Gneiting & Raftery, 2007) so we have $p_{\mu_\star, \sigma_\star} = p_{\mu, \sigma}$. For $\lambda \in [0, 1]$, we have the following result. This is a simple restatement of Proposition 4.1.

**Proposition C.1:** *For any $\beta \in (0, 2)$ and $\lambda \in (0, 1)$, we have that $p_{\mu_\star, \sigma_\star} \in \mathrm{argmax}_{q \in \mathfrak{G}} S_{\lambda, \beta}(q, p_{\mu, \sigma})$ with*

$$\mu_\star = \mu, \qquad \sigma_\star = (2\lambda^{-2/(2-\beta)} - 1)^{-1}\sigma^2.$$

We start with the following lemma.

**Lemma C.2:** *Assume that $X \sim \mathcal{N}(0, \sigma^2 \mathrm{Id})$ for $\sigma > 0$. Then for any $\beta > 0$, we have $0 \in \mathrm{argmin}_{c \in \mathbb{R}^d} \mathbb{E}[\|X - c\|^\beta]$.*

*Proof.* For any $\beta > 0$, let $f_\beta(c) = \mathbb{E}[\|X - c\|^\beta]$. For any $\beta > 0$, we have that $f_\beta$ is continuous, using the dominated convergence theorem, and coercive, using Fatou's lemma, hence admits at least one minimizer. If $\beta \geq 1$, then $f_\beta$ is convex. Assume that $c^\star$ is a minimizer. Then $-c^\star$ is also a minimizer and by convexity, we have that $0$ is also a minimizer. If $\beta \in (0, 1)$ then $f_\beta$ is strictly concave. Assume that $c^\star$ is a minimizer. Then $-c^\star$ is also a minimizer. If $0$ is not a minimizer of $f_\beta$, then $f_\beta(0) > f_\beta(c^\star)$. Hence, by strict concavity, we have that $\lim_{t \to +\infty} f_\beta(tc^\star) = -\infty$ which is absurd. Hence, $0$ is also a minimizer. □

*Proof.* First, we have that for any $q \in \mathfrak{G}$ with mean $\tilde{\mu}$ and covariance $\tilde{\sigma}^2 \mathrm{Id}$

$$S_{\lambda, \beta}(q, p_{\mu, \sigma}) = -\mathbb{E}[\|\mu - \tilde{\mu} + (\sigma^2 + \tilde{\sigma}^2)^{1/2} Z\|^\beta] + \frac{\lambda}{2}\mathbb{E}[\|\sqrt{2}\tilde{\sigma} Z\|^\beta],$$

where $Z \sim \mathcal{N}(0, \mathrm{Id})$. Maximizing with respect to $\tilde{\mu}$ first, we get that $\mu^\star = \mu$ thanks to Lemma C.2. In addition, if $\tilde{\mu} = \mu$, we have

$$
\begin{aligned}
S_{\lambda, \beta}(q, p_{\mu, \sigma}) &= -\mathbb{E}[\|\mu - \tilde{\mu} + (\sigma^2 + \tilde{\sigma}^2)^{1/2} Z\|^\beta] + \frac{\lambda}{2}\mathbb{E}[\|\sqrt{2}\tilde{\sigma} Z\|^\beta] \\
&= -(\sigma^2 + \tilde{\sigma}^2)^{\beta/2}\mathbb{E}[\|Z\|^\beta] + \frac{\lambda}{2}2^{\beta/2}\tilde{\sigma}^\beta \mathbb{E}[\|Z\|^\beta] \\
&= -\mathbb{E}[\|Z\|^\beta]((\sigma^2 + \tilde{\sigma}^2)^{\beta/2} + \frac{\lambda}{2}2^{\beta/2}\tilde{\sigma}^\beta).
\end{aligned}
$$

Maximizing with respect to $\tilde{\sigma}$, we get that

$$\sigma_\star^2 = (2\lambda^{-2/(2-\beta)} - 1)^{-1}\sigma^2.$$

□

So the effect of the introduction of $\lambda < 1$ is that $\sigma_\star$ underestimates $\sigma$, i.e. the model is too concentrated around its mean. The effect is stronger as $\beta \to 2$. In what follows, we denote

$$f(\lambda, \beta) = (2\lambda^{-2/(2-\beta)} - 1)^{-1}. \tag{20}$$

### C.2. Application to diffusion sampling

Consider a Gaussian target distribution $p_0$. We investigate the effect of using the generalized energy score to learn the Gaussian conditionals $p_{0|t}$ on the sampling updates used at inference time, i.e. when substituting our approximation of $p_{0|t}$ within (5) to obtain an approximation of $p_{s|t}$.

We first recall a few useful lemmas. Using (2), the forward model is given by

$$\mathbf{X}_t = \alpha_t \mathbf{X}_0 + \sigma_t \mathbf{Z}, \tag{21}$$

with $\mathbf{Z} \sim \mathcal{N}(0, \mathrm{Id})$ and $\mathbf{X}_0 \sim p_{\mu, \sigma}$, the target density. The diffusion $(\mathbf{X}_t)_{t \in [0, 1]}$ given by

$$\mathrm{d}\mathbf{X}_t = f_t \mathbf{X}_t \mathrm{d}t + g_t \mathrm{d}\mathbf{B}_t, \tag{22}$$

where $(\mathbf{B}_t)_{t \in [0, 1]}$ is a $d$-dimensional Brownian motion has the same marginal distributions as (21) for

$$f_t = \partial_t \log(\alpha_t), \qquad g_t^2 = 2\alpha_t \sigma_t \partial_t(\sigma_t/\alpha_t).$$

We refer to Appendix F for a derivation of this fact, see also (Song et al., 2021b).

**Lemma C.3:** *For any $t \in [0,1]$ we have that*

$$(\mathbf{X}_0, \mathbf{X}_t) \sim \mathcal{N}\left(\begin{pmatrix} \mu \\ \alpha_t \mu \end{pmatrix}, \begin{pmatrix} \sigma^2 \mathrm{Id} & \alpha_t \sigma^2 \mathrm{Id} \\ \alpha_t \sigma^2 \mathrm{Id} & (\alpha_t^2 \sigma^2 + \sigma_t^2)\mathrm{Id} \end{pmatrix}\right).$$

*Proof.* This is a direct consequence of (21). □

In particular, we also have the following lemma.

**Lemma C.4:** *For any $t \in [0,1]$ we have that*

$$\mathbf{X}_0 | \mathbf{X}_t \sim \mathcal{N}(r(t)\mathbf{X}_t + (1 - r(t))\mu, \sigma^2(1 - r_{2,2}(t))\mathrm{Id}),$$

*where*

$$r(t) = \frac{\alpha_t \sigma^2}{\alpha_t^2 \sigma^2 + \sigma_t^2}, \qquad r_{2,2}(t) = \frac{\alpha_t^2 \sigma^2}{\alpha_t^2 \sigma^2 + \sigma_t^2}.$$

*Proof.* This is a combination of Lemma C.3 and the formula for computing Gaussian posteriors. □

**Lemma C.5:** *For $s, t \in [0,1]$ with $t \geq s$ we have that*

$$\mathbf{X}_t | \mathbf{X}_s \sim \mathcal{N}\left(\frac{\alpha_t}{\alpha_s}\mathbf{X}_s, \left(\sigma_t^2 - \left(\frac{\sigma_s \alpha_t}{\alpha_s}\right)^2\right)\mathrm{Id}\right).$$

*Proof.* This is a direct consequence of (22). □

**Lemma C.6:** *For $s, t \in [0,1]$ with $t \geq s$ we have that*

$$\mathbf{X}_s | \mathbf{X}_0, \mathbf{X}_t \sim \mathcal{N}\left(r(s,t)\mathbf{X}_t + \alpha_s(1 - r_{2,2}(s,t))\mathbf{X}_0, \sigma_s^2(1 - r_{2,2}(s,t))\mathrm{Id}\right),$$

*where*

$$r(s,t) = \frac{\alpha_t \sigma_s^2}{\alpha_s \sigma_t^2}, \qquad r_{2,2}(s,t) = \frac{\alpha_t^2 \sigma_s^2}{\alpha_s^2 \sigma_t^2}.$$

*Proof.* Using that $\mathbf{X}_t | \mathbf{X}_s$ is the same as $\mathbf{X}_t | \mathbf{X}_s, \mathbf{X}_0$ by the Markov property, we get the distribution of $(\mathbf{X}_s, \mathbf{X}_t) | \mathbf{X}_0$. We conclude upon using the formula for computing Gaussian posteriors. □

Note that in Appendix F, we will establish Lemma C.6 in a more general setting, i.e., we will introduce a churn parameter $\varepsilon$ and draw further connection with DDIM (Song et al., 2021a). At this stage, Lemma C.6 recovers (Ho et al., 2020) which corresponds to setting the churn parameter to $\varepsilon = 1$. Now, combining (20), Lemma C.4 and Lemma C.6 we get the following proposition.

**Proposition C.7:** *For $s, t \in [0, 1]$ with $t \geq s$ we have that*

$$\mathbf{X}_s | \mathbf{X}_t = \mathcal{N}(\hat{\mu}_{s|t}, \hat{\sigma}_{s|t}^2 \mathrm{Id}),$$

*with*

$$\hat{\mu}_{s|t} = [r(s,t) + \alpha_s(1 - r_{2,2}(s,t))r(t)]\mathbf{X}_t + \alpha_s(1 - r_{2,2}(s,t))(1 - r(t))\mu,$$
$$\hat{\sigma}_{s|t}^2 = \sigma_s^2(1 - r_{2,2}(s,t)) + f(\lambda, \beta)\sigma^2 \alpha_s^2 (1 - r_{2,2}(s,t))^2 (1 - r_{2,2}(t)).$$

*In particular, denoting $\mathbf{X}_t \sim \mathcal{N}(\hat{\mu}_t, \hat{\sigma}_t^2 \mathrm{Id})$ we get that*

$$\hat{\mu}_s = [r(s,t) + \alpha_s(1 - r_{2,2}(s,t))r(t)]\hat{\mu}_t + \alpha_s(1 - r_{2,2}(s,t))(1 - \alpha_t r(t))\mu, \tag{23}$$
$$\hat{\sigma}_s^2 = \sigma_s^2(1 - r_{2,2}(s,t)) + f(\lambda, \beta)\sigma^2 \alpha_s^2 (1 - r_{2,2}(s,t))^2 (1 - r_{2,2}(t)) + \hat{\sigma}_t^2 [r(s,t) + \alpha_s(1 - r_{2,2}(s,t))r(t)]^2.$$

In particular, the update on the mean in (23) is not dependent on $\lambda$ and $\beta$. Therefore we estimate correctly the mean of $p_{s|t}$ but incorrectly the variance for $\lambda < 1$.

The curves presented in Figure 1 are obtained using Proposition C.7.

# D. Different learning objectives

First in Appendix D.1 we compare the energy diffusion loss (13) with a similar loss based on Maximum Mean Discrepancy and show that they only differ by a term which does not depend on $\theta$. In Appendix D.2, we compare the *conditional* loss and the *joint* loss, as discussed in Section 4.2. In Appendix D.3, we consider a third possible option, i.e., we compare the *conditional* loss with a *marginal* loss. We highlight the limitations of the *marginal* loss in that case. We derive the empirical versions of the conditional and joint losses in Appendix D.4. Finally in Appendix D.5, we prove the main result of Section 4.2, i.e., we prove Proposition 4.2 and compare the SNR of the interaction terms of the joint and conditional losses.

## D.1. Energy diffusion loss and Maximum Mean Discrepancy diffusion loss

In this section, we first compare the energy diffusion loss (13) and another loss based on Maximum Mean Discrepancy (MMD), denoted MMD diffusion loss. We show that these two losses are equal up to a constant which does not depend on the network parameters $\theta$. However, this additional term appearing in the MMD diffusion loss is intractable, even though it could be potentially estimated using important sampling techniques. We consider the case where the kernel $k$ is given by $k(x, x') = -\|x - x'\|$. While our discussion can be extended to other settings, we also focus on the case where $\lambda = 1, \beta = 1$ and $w_t = 1$ for simplicity.

First, we consider the following MMD diffusion loss

$$\mathcal{L}_{\mathrm{MMD}}(\theta) = \int_0^1 \int_{\mathbb{R}^d} p_t(x_t) D_\rho(p_{0|t}^\theta, p_{0|t}) \mathrm{d}x_t \mathrm{d}t. \tag{24}$$

We recall that the squared MMD is given by $D_\rho(p, q) = S_\rho(q, q) - S_\rho(p, q)$ with $\rho = -k$ and

$$\mathcal{L}(\theta) = -\int_0^1 \int_{\mathbb{R}^d \times \mathbb{R}^d} p_{0,t}(x_0, x_t) S_\rho(p_{0|t}^\theta, x_0) \mathrm{d}x_0 \mathrm{d}x_t \mathrm{d}t. \tag{25}$$

First, we highlight the following result which draws a connection between kernel scores and squared MMD.

**Proposition D.1:** *Let $x \in \mathbb{R}^d$ be such that $k(x, x) = 0$, then we have $D_\rho(p, \delta_x) = -S_\rho(p, x)$.*

*Proof.* For any distribution $q$, we have $S_\rho(q, q) = \frac{1}{2}\mathbb{E}_{q \otimes q}[k(X, X')]$ and therefore $S_\rho(\delta_x, \delta_x) = \frac{1}{2}k(x, x) = 0$. Similarly, we have that

$$S_\rho(p, \delta_x) = -\frac{1}{2}\mathbb{E}_{p \otimes p}[k(X, X')] + \mathbb{E}_{p \otimes \delta_x}[k(X, Y)] = -\frac{1}{2}\mathbb{E}_{p \otimes p}[k(X, X')] + \mathbb{E}_p[k(X, x)] = S_\rho(p, x).$$

Therefore, we have that

$$D_\rho(p, \delta_x) = S_\rho(\delta_x, \delta_x) - S_\rho(p, \delta_x) = -S_\rho(p, x).$$

□

Using Proposition D.1, (25) can be rewritten as

$$\mathcal{L}(\theta) = \int_0^1 \int_{\mathbb{R}^d \times \mathbb{R}^d} p_{0,t}(x_0, x_t) D_\rho(p_{0|t}^\theta, \delta_{x_0}) \mathrm{d}x_0 \mathrm{d}x_t \mathrm{d}t. \tag{26}$$

Therefore, our energy diffusion loss can be expressed in terms of squared MMD. We observe that both (24) and (26) involve an integrated version of the squared MMD. However, in (26) we target $\delta_{x_0}$ while in (24) we target $p_{0|t}$. In the rest of this section, we show that these two losses actually only differ by a term which does not depend on the network parameters $\theta$.

Let us start with the *energy diffusion loss* $\mathcal{L}$ given in (25) . We have

$$\begin{aligned}
\mathcal{L}(\theta) = &-\frac{1}{2} \int_0^1 \int_{(\mathbb{R}^d)^4} p_{0,t}(x_0, x_t) p_{0|t}^\theta(x_0^1|x_t) p_{0|t}(x_0^2|x_t) \|x_0^1 - x_0^2\| \mathrm{d}x_0 \mathrm{d}x_0^1 \mathrm{d}x_0^2 \mathrm{d}x_t \mathrm{d}t \\
&+ \int_0^1 \int_{(\mathbb{R}^d)^3} p_{0,t}(x_0, x_t) p_{0|t}^\theta(x_0^1|x_t) \|x_0 - x_0^1\| \mathrm{d}x_0 \mathrm{d}x_0^1 \mathrm{d}x_t \mathrm{d}t \\
= &-\frac{1}{2} \int_0^1 \int_{(\mathbb{R}^d)^3} p_t(x_t) p_{0|t}^\theta(x_0^1|x_t) p_{0|t}^\theta(x_0^2|x_t) \|x_0^1 - x_0^2\| \mathrm{d}x_0^1 \mathrm{d}x_0^2 \mathrm{d}x_t \mathrm{d}t \\
&+ \int_0^1 \int_{(\mathbb{R}^d)^3} p_t(x_t) p_{0|t}(x_0|x_t) p_{0|t}^\theta(x_0^1|x_t) \|x_0 - x_0^1\| \mathrm{d}x_0 \mathrm{d}x_0^1 \mathrm{d}x_t \mathrm{d}t.
\end{aligned}$$

For the MMD diffusion loss (24), we have

$$\begin{aligned}
\mathcal{L}_{\mathrm{MMD}}(\theta) = &-\frac{1}{2} \int_0^1 \int_{(\mathbb{R}^d)^3} p_t(x_t) p_{0|t}(x_0^1|x_t) p_{0|t}(x_0^2|x_t) \|x_0^1 - x_0^2\| \mathrm{d}x_0^1 \mathrm{d}x_0^2 \mathrm{d}x_t \mathrm{d}t \\
&-\frac{1}{2} \int_0^1 \int_{(\mathbb{R}^d)^3} p_t(x_t) p_{0|t}^\theta(x_0^1|x_t) p_{0|t}^\theta(x_0^2|x_t) \|x_0^1 - x_0^2\| \mathrm{d}x_0^1 \mathrm{d}x_0^2 \mathrm{d}x_t \mathrm{d}t \\
&+ \int_0^1 \int_{(\mathbb{R}^d)^3} p_t(x_t) p_{0|t}(x_0|x_t) p_{0|t}^\theta(x_0^1|x_t) \|x_0 - x_0^1\| \mathrm{d}x_0 \mathrm{d}x_0^1 \mathrm{d}x_t \mathrm{d}t.
\end{aligned}$$

To summarize, in this section we have shown that the kernel score losses that we have defined in Section 3 can be expressed in terms of squared MMD. More precisely, we have shown the following result in the specific case where $k(x, x') = -\|x - x'\|$ (but the result remains true for every positive definite kernel).

**Proposition D.2:** *We have that*

$$\mathcal{L}(\theta) = \mathcal{L}_{\mathrm{MMD}}(\theta) + C,$$

*with $C \in \mathbb{R}$ a constant independent of $\theta$.*

This means that $p_{0|t}^\theta$ also minimizes $\mathcal{L}(\theta)$. Note that we could also have derived this result from the properties of the *strictly proper* scoring rules.

When optimizing with respect to the parameters of the kernel, here the kernel $\rho(x, x') = \|x - x'\|$ does not depend on any parameter, the losses $\mathcal{L}(\theta)$ and $\mathcal{L}_{\mathrm{MMD}}(\theta)$ differ since now we cannot neglect the additional term appearing in $\mathcal{L}_{\mathrm{MMD}}(\theta)$. Unfortunately, this term recalled below is computationally problematic

$$\int_0^1 \int_{(\mathbb{R}^d)^3} p_t(x_t) p_{0|t}(x_0|x_t) p_{0|t}(x_0^1|x_t) \|x_0 - x_0^1\| \mathrm{d}x_0 \mathrm{d}x_0^1 \mathrm{d}x_t \mathrm{d}t.$$

Indeed an estimate of this integral requires sampling $x_0^1$ from the conditional distribution $p_{0|t}(x_0|x_t)$. However this can be approximated by using the following self-normalized importance sampling approximation of this distribution

$$p_{0|t}(x_0|x_t) = \frac{p_{t|0}(x_t|x_0)p_0(x_0)}{p_t(x_t)} \approx \sum_{i=1}^{n} w^i \delta_{X_0^i}(x_0), \quad w^i \propto p_{t|0}(x_t|X_0^i), \; \sum_{i=1}^{n} w^i = 1, \qquad (27)$$

where $X_0^i \overset{\text{i.i.d.}}{\sim} p_0$. Such approximation was considered by Xu et al. (2023) to derive low variance regression targets in diffusion models. However, for $t$ small, the distribution of the weights $(w_i)_{i=1}^{N}$ is very degenerate and the approximation of $p_{0|t}(x_0|x_t)$ will be poor.

### D.2. Joint Diffusion Energy Loss

Now that we have established the connection between MMD diffusion losses and energy diffusion losses in Proposition D.2, we are going to discuss other possible learning objectives. We recall the *joint diffusion energy loss* introduced in (15)

$$\mathcal{L}_{\text{joint}}(\theta) = -\int_0^1 w_t \mathbb{E}_{p_{0,t}}\left[S_{\lambda,\beta}(p_{0|t}^{\theta} p_t, (X_0, X_t))\right] \mathrm{d}t.$$

and the $\mathcal{L}_{\text{MMD,joint}}$ loss which is defined using *joint* distributions as follows

$$\mathcal{L}_{\text{MMD,joint}}(\theta) = \int_0^1 D_\rho(p_t p_{0|t}^{\theta}, p_{0,t})\mathrm{d}t.$$

We emphasize again that $D_\rho$ appearing in $\mathcal{L}_{\text{MMD,joint}}$ is defined over $\mathbb{R}^d \times \mathbb{R}^d$ and not $\mathbb{R}^d$ as in $\mathcal{L}_{\text{MMD}}$.

Similarly to Proposition D.2, we can show the following result. Recall that we consider $w_t = 1$ to simplify presentation.

> **Proposition D.3:** *Let $\rho((x_0, x_t), (x_0', x_t')) = \rho(x_0, x_0') + \rho(x_t, x_t')$. We have that*
>
> $$\mathcal{L}_{\text{joint}}(\theta) = \mathcal{L}_{\text{MMD,joint}}(\theta) + C,$$
>
> *with $C \in \mathbb{R}$ a constant independent of $\theta$.*

Developing $\mathcal{L}_{\text{MMD,joint}}(\theta)$, we obtain

$$\mathcal{L}_{\text{MMD,joint}}(\theta) = -\frac{1}{2}\int_0^1\int_{(\mathbb{R}^d)^4} p_t(x_t^1)p_{0|t}(x_0^1|x_t^1)p_t(x_t^2)p_{0|t}(x_0^2|x_t^2)[\|x_0^1 - x_0^2\| + \|x_t^1 - x_t^2\|]\mathrm{d}x_0^1\mathrm{d}x_0^2\mathrm{d}x_t^1\mathrm{d}x_t^2\mathrm{d}t$$

$$-\frac{1}{2}\int_0^1\int_{(\mathbb{R}^d)^4} p_t(x_t^1)p_{0|t}^{\theta}(x_0^1|x_t^1)p_t(x_t^2)p_{0|t}^{\theta}(x_0^2|x_t^2)[\|x_0^1 - x_0^2\| + \|x_t^1 - x_t^2\|]\mathrm{d}x_0^1\mathrm{d}x_0^2\mathrm{d}x_t^1\mathrm{d}x_t^2\mathrm{d}t$$

$$+\int_0^1\int_{(\mathbb{R}^d)^4} p_t(x_t^1)p_{0|t}^{\theta}(x_0^1|x_t^1)p_t(x_t^2)p_{0|t}(x_0^2|x_t^2)[\|x_0^1 - x_0^2\| + \|x_t^1 - x_t^2\|]\mathrm{d}x_0^1\mathrm{d}x_0^2\mathrm{d}x_t^1\mathrm{d}x_t^2\mathrm{d}t.$$

We recall that the term which does not depend on $\theta$ in $\mathcal{L}_{\text{MMD}}$ is difficult to approximate, the proposed approximation using importance sampling (27) is indeed expected to perform poorly. On the contrary, the term which does not depend on $\theta$ in $\mathcal{L}_{\text{MMD,joint}}$ is very easy to approximate. This suggests that if one would optimize the loss with respect to the parameters of the kernel, then the choice $\mathcal{L}_{\text{MMD,joint}}$ is more suitable. However, in the case where the parameters are fixed, as it is the case in our framework, then we will show in Appendix D.5 that $\mathcal{L}_{\text{MMD}}$ is more sample efficient.

### D.3. Marginal Diffusion Energy Loss

Finally, we introduce a last discrepancy $\mathcal{L}_{\text{MMD,marginal}}$ given by

$$\mathcal{L}_{\text{MMD,marginal}}(\theta) = \int_0^1 D_\rho(p_0^{\theta,t}, p_0)\mathrm{d}t,$$

where

$$p_0^{\theta,t}(x_0) = \int_{\mathbb{R}^d} p_t(x_t) p_{0|t}^\theta(x_0|x_t) \mathrm{d}x_t.$$

After a few simplifications, we obtain

$$\mathcal{L}_{\mathrm{MMD,marginal}}(\theta) = -\frac{1}{2} \int_0^1 \int_{(\mathbb{R}^d)^4} p_t(x_t^1) p_{0|t}(x_0^1|x_t^1) p_t(x_t^2) p_{0|t}(x_0^2|x_t^2) \|x_0^1 - x_0^2\| \mathrm{d}x_0^1 \mathrm{d}x_0^2 \mathrm{d}x_t^1 \mathrm{d}x_t^2 \mathrm{d}t$$

$$-\frac{1}{2} \int_0^1 \int_{(\mathbb{R}^d)^4} p_t(x_t^1) p_{0|t}^\theta(x_0^1|x_t^1) p_t(x_t^2) p_{0|t}^\theta(x_0^2|x_t^2) \|x_0^1 - x_0^2\| \mathrm{d}x_0^1 \mathrm{d}x_0^2 \mathrm{d}x_t^1 \mathrm{d}x_t^2 \mathrm{d}t$$

$$+ \int_0^1 \int_{(\mathbb{R}^d)^4} p_t(x_t^1) p_{0|t}^\theta(x_0^1|x_t^1) p_t(x_t^2) p_{0|t}(x_0^2|x_t^2) \|x_0^1 - x_0^2\| \mathrm{d}x_0^1 \mathrm{d}x_0^2 \mathrm{d}x_t^1 \mathrm{d}x_t^2 \mathrm{d}t.$$

Note that while the minimizers of the conditional and joint losses are unique and given by $p_{0|t}$, this is not the case for the marginal loss. Indeed, we simply have that a minimizer of the marginal loss should satisfy

$$p_0(x_0) = \int_{\mathbb{R}^d} p_{0|t}^\theta(x_0|x_t) p_t(x_t) \mathrm{d}x_t.$$

This simply says that we should transport $p_t$ to $p_0$ and there is an infinite number of such transports.

### D.4. Empirical versions of MMD and MMD Joint diffusion losses

In this section we justify the form of (14), i.e., the empirical version of $\mathcal{L}$ given by (13). For completeness, here we present the empirical version of $\mathcal{L}_{\mathrm{MMD}}$ which can be expressed as $\mathcal{L}_{\mathrm{MMD}}(\theta) = \mathcal{L}(\theta) + C$, where $C$ is a term which does not depend on $\theta$, according to Proposition D.2

We recall that we have

$$\mathcal{L}_{\mathrm{MMD}}(\theta) = -\frac{1}{2} \int_0^1 \int_{(\mathbb{R}^d)^3} p_t(x_t) p_{0|t}(x_0^1|x_t) p_{0|t}(x_0^2|x_t) \|x_0^1 - x_0^2\| \mathrm{d}x_0^1 \mathrm{d}x_0^2 \mathrm{d}x_t \mathrm{d}t$$

$$-\frac{1}{2} \int_0^1 \int_{(\mathbb{R}^d)^3} p_t(x_t) p_{0|t}^\theta(x_0^1|x_t) p_{0|t}^\theta(x_0^2|x_t) \|x_0^1 - x_0^2\| \mathrm{d}x_0^1 \mathrm{d}x_0^2 \mathrm{d}x_t \mathrm{d}t$$

$$+ \int_0^1 \int_{(\mathbb{R}^d)^3} p_t(x_t) p_{0|t}(x_0|x_t) p_{0|t}^\theta(x_0^1|x_t) \|x_0 - x_0^1\| \mathrm{d}x_0 \mathrm{d}x_0^1 \mathrm{d}x_t \mathrm{d}t$$

**Confinement term.** We first focus on the last term of the loss, known as *confinement term*

$$\mathcal{C}(\theta) = \int_0^1 \int_{(\mathbb{R}^d)^3} p_{0,t}(x_0^1, x_t^1) p_{0|t}^\theta(x_0^2|x_t^1) \|x_0^1 - x_0^2\| \mathrm{d}x_0^1 \mathrm{d}x_0^2 \mathrm{d}x_t^1 \mathrm{d}t.$$

This can be estimated unbiasedly by

$$\mathcal{C}_{n,m} = \frac{1}{n} \sum_{i=1}^n \frac{1}{m} \sum_{j=1}^m \|X_0^{1,i} - X_0^{2,i,j}\|,$$

where $t_i \sim \mathrm{Unif}([0,1])$, $(X_0^{1,i}, X_{t_i}^i) \sim p_{0,t_i}$ for $i \in [n]$ and $X_0^{2,i,j} \sim p_{0|t}^\theta(\cdot|X_{t_i}^i)$ for $j \in [m]$.

**Prediction Interaction term.** We now focus on the *prediction interaction term*

$$\mathcal{I} = \int_0^1 \int_{(\mathbb{R}^d)^3} p_t(x_t) p_{0|t}^\theta(x_0^1|x_t) p_{0|t}^\theta(x_0^2|x_t) \|x_0^1 - x_0^2\| \mathrm{d}x_0^1 \mathrm{d}x_0^2 \mathrm{d}x_t \mathrm{d}t.$$

This can be approximated unbiasedly by

$$\mathcal{I}_{n,m} = \frac{1}{n} \sum_{i=1}^n \frac{1}{m(m-1)} \sum_{j,j'=1}^m \|X_0^{i,j} - X_0^{i,j'}\|.$$

where $t_i \sim \mathrm{Unif}([0,1])$, $X_{t_i}^i \sim p_{t_i}$ for $i \in [n]$ and $X_0^{i,j} \sim p_{0|t}^\theta(\cdot|X_{t_i}^i)$ for $j \in [m]$.

**Target interaction term.** We finally focus on the *target interaction term*. This term is harder to describe and implement. However, we emphasize that it is not needed in order to optimize $\mathcal{L}$ given by (13). It is only needed when computing (24). We have

$$\mathcal{I}^t = \int_0^1 \int_{(\mathbb{R}^d)^3} p_t(x_t) p_{0|t}(x_0^1|x_t) p_{0|t}(x_0^2|x_t) \|x_0^1 - x_0^2\| \mathrm{d}x_0^1 \mathrm{d}x_0^2 \mathrm{d}x_t \mathrm{d}t.$$

As explained before it can also be written as

$$\mathcal{I}^t = \int_0^1 \int_{(\mathbb{R}^d)^3} p_{0,t}(x_0^1, x_t^1) \frac{p_{t|0}(x_t^1|x_0^2) p_0(x_0^2)}{\int_{\mathbb{R}^d} p_{t|0}(x_t^1|x_0^3) p_0(x_0^3) \mathrm{d}x_0^3} \|x_0^1 - x_0^2\| \mathrm{d}x_0^1 \mathrm{d}x_0^2 \mathrm{d}x_t^1 \mathrm{d}t.$$

One estimate of this objective is given by

$$\mathcal{I}_{n,m}^t = \frac{1}{n} \sum_{i=1}^n \sum_{j=1, j \neq i}^n \frac{p_{t_i|0}(X_{t_i}^i|X_0^j)}{\sum_{k=1, k \neq i}^n p_{t_i|0}(X_{t_i}^i|X_0^k)} \|X_0^i - X_0^j\|,$$

where $t_i \sim \mathrm{Unif}([0,1])$, $X_0^i, X_{t_i}^i \sim p_{0,t_i}$.

## D.5. Sample efficiency

Previous subsections assume $w_t = 1$ for ease of presentation. We consider a general weighting function $w_t$ here. Finally, in this section, we discuss the sample efficiency of the empirical version of the *energy diffusion loss* (13) and the *joint energy diffusion loss* (15). In particular, we prove Proposition 4.2.

**U-statistics.** We need first to recall a few basic results about $U$-statistics of order 2. Consider a symmetric function $h : (\mathbb{R}^d)^2 \to \mathbb{R}$ and let

$$U_n = \binom{n}{2}^{-1} \sum_{1 \leq i_1 < i_2 \leq n} h(X_{i_1}, X_{i_2}), \qquad X_i \overset{\text{i.i.d.}}{\sim} p.$$

We recall Hoeffding's theorem on the variance of $U$-statistics, see (Serfling, 2009) for instance.

**Proposition D.4 (Hoeffding's theorem):** *We have*

$$\mathrm{Var}(U_n) = \frac{2}{n(n-1)} [2(n-2) \mathrm{Var}(\mathbb{E}[h(X_1, X_2) \mid X_1]) + \mathrm{Var}(h(X_1, X_2))].$$

This implies that

$$\lim_{n \to +\infty} n \mathrm{Var}(U_n) = 4 \mathrm{Var}(\mathbb{E}[h(X_1, X_2) \mid X_1]).$$

Another classical result is the law of total variance.

**Lemma D.5:** *We have*
$$\mathrm{Var}(Y) = \mathrm{Var}(\mathbb{E}[Y \mid X]) + \mathbb{E}[\mathrm{Var}(Y \mid X)].$$

Combining those two results, we obtain the following proposition which is central to the rest of the study.

**Proposition D.6:** *Let $X_i \overset{\text{i.i.d.}}{\sim} p$ for $i \in \{1, ..., n\}$ and $n \in \mathbb{N}$. Define also $\{U_i\}_{i=1}^n$ such that for each $i \in \{1, \ldots, n\}$, $U_i$ is independent from $X_j$ for $j \neq i$, $\mathbb{E}[U_i|X_i]$ are i.i.d. and $U_i$ is a $U$-statistics of order $2$ with $m$ samples. We define*

$$U = \frac{1}{n} \sum_{i=1}^n U_i.$$

*Then, we have that*

$$\text{Var}(U) = \frac{1}{n}\text{Var}(\mathbb{E}[U_1 \mid X_1]) + \frac{1}{n}\mathbb{E}[\text{Var}(U_1 \mid X_1)].$$

*In particular, we have that*

$$\lim_{m \to +\infty} \text{Var}(U) = \frac{1}{n}\text{Var}(\mathbb{E}[U_1 \mid X_1]).$$

**Conditional and joint interaction terms.** We consider two interaction terms. Recall that we assume that we are working here under the strong assumption that we are at parameter $\theta$ such that $p_{0|t}^\theta = p_{0|t}$. The first one is given by the empirical interaction term in the *joint energy diffusion loss*

$$\mathcal{I}_{n,m,\text{joint}} = \frac{1}{n} \sum_{i=1}^n \frac{1}{m(m-1)} \sum_{j,j'=1}^m w_{t_i}\{\|X_0^{i,j} - X_0^{i,j'}\| + \|X_{t_i}^{i,j} - X_{t_i}^{i,j'}\|\},$$

with $t_i \overset{\text{i.i.d.}}{\sim} \text{Unif}([0,1])$, $\{X_0^{i,j}\}_{i,j=1}^{n,m} \overset{\text{i.i.d.}}{\sim} p_0$ and $X_{t_i}^{i,j} \sim p_{t_i|0}(\cdot|X_0^{i,j})$. The second one is given by the empirical interaction term in the *conditional* loss:

$$\mathcal{I}_{n,m} = \frac{1}{n} \sum_{i=1}^n \frac{1}{m(m-1)} \sum_{j,j'=1}^m w_{t_i}\{\|X_0^{i,j} - X_0^{i,j'}\|\},$$

where $t_i \overset{\text{i.i.d.}}{\sim} \text{Unif}([0,1])$ and for any $i \in [n]$, $X_{t_i} \sim p_{t_i}$. In addition, $\{X_0^{i,j}\}_{i,j=1}^{n,m}$ are conditionally independent, i.e. $X_0^{i,j} \sim p_{0|t_i}(\cdot|X_{t_i})$.

We start with the following proposition giving the mean of $\mathcal{I}_{n,m}$ and $\mathcal{I}_{n,m,\text{joint}}$.

**Proposition D.7:** *We have that*

$$\mathbb{E}[\mathcal{I}_{n,m,\text{joint}}] = \int_0^1 \int_{(\mathbb{R}^d)^4} w_t\{\|x_0 - x_0'\| + \|x_t - x_t'\|\}p_{0,t}(x_0, x_t)p_{0,t}(x_0', x_t')\mathrm{d}x_0\mathrm{d}x_t\mathrm{d}x_0'\mathrm{d}x_t'\mathrm{d}t.$$

*Similarly, we have that*

$$\mathbb{E}[\mathcal{I}_{n,m}] = \int_0^1 \int_{(\mathbb{R}^d)^3} w_t\|x_0 - x_0'\|p_t(x_t)p_{0|t}(x_0|x_t)p_{0|t}(x_0'|x_t)\mathrm{d}x_0\mathrm{d}x_0'\mathrm{d}x_t\mathrm{d}t.$$

Next, we can obtain the asymptotic variance of those random variables.

**Proposition D.8:** *We have that*

$$\lim_{m \to +\infty} \text{Var}(\mathcal{I}_{n,m,\text{joint}})$$

$$= \frac{1}{n}\text{Var}_{t \sim \text{Unif}([0,1])}\left(\int_{(\mathbb{R}^d)^4} w_t\{\|x_0 - x_0'\| + \|x_t - x_t'\|\}p_{0,t}(x_0, x_t)p_{0,t}(x_0'x_t')\mathrm{d}x_0\mathrm{d}x_t\mathrm{d}x_0'\mathrm{d}x_t'\right).$$

*Similarly, we have that*

$$\lim_{m \to +\infty} \text{Var}(\mathcal{I}_{n,m}) = \frac{1}{n}\text{Var}_{t \sim \text{Unif}([0,1]), X_t \sim p_t}\left(\int_{(\mathbb{R}^d)^2} w_t\|x_0 - x_0'\|p_{0|t}(x_0|x_t)p_{0|t}(x_0'|x_t)\mathrm{d}x_0\mathrm{d}x_0'\right).$$

*Proof.* This is an application of Proposition D.6, with $X = t \sim \mathrm{Unif}([0,1])$ in the case of $\mathcal{I}_{n,m,\mathrm{joint}}$ and $X = (t, X_t) \sim \mathrm{Unif}([0,1]) \otimes p_t$ in the case of $\mathcal{I}_{n,m}$. $\qquad\square$

In order to compare the sample efficiency of these different terms, we compute their Signal-to-Noise Ratio (SNR). In particular, we compare

$$\mathrm{SNR}(\mathcal{I}_{n,\mathrm{joint}}) := \lim_{m \to +\infty} \mathrm{SNR}(\mathcal{I}_{n,m,\mathrm{joint}}) = \lim_{m \to +\infty} \frac{\mathbb{E}[\mathcal{I}_{n,m,\mathrm{joint}}]^2}{\mathrm{Var}(\mathcal{I}_{n,m,\mathrm{joint}})},$$

$$\mathrm{SNR}(\mathcal{I}_n) := \lim_{m \to +\infty} \mathrm{SNR}(\mathcal{I}_{n,m}) = \lim_{m \to +\infty} \frac{\mathbb{E}[\mathcal{I}_{n,m}]^2}{\mathrm{Var}(\mathcal{I}_{n,m})}.$$

The larger the SNR the better. In particular, if $\mathrm{SNR}(\mathcal{I}_n) \geq \mathrm{SNR}(\mathcal{I}_{n,\mathrm{joint}})$ then we claim that the conditional version is more sample efficient than the joint version. Note that both $\mathrm{SNR}(\mathcal{I}_n)$ and $\mathrm{SNR}(\mathcal{I}_{n,\mathrm{joint}})$ are linear functions of $N$. Therefore if $\mathrm{SNR}(\mathcal{I}_n) \geq \mathrm{SNR}(\mathcal{I}_{n,\mathrm{joint}})$ then we can interpret this as one sample used in the conditional version is worth $\mathrm{SNR}(\mathcal{I}_n)/\mathrm{SNR}(\mathcal{I}_{n,\mathrm{joint}})$ used in the joint version.

While the general expression for these SNR can be hard to obtain, we are going to analyze the specific case where the target is Gaussian. In that case the posterior is known and explicit expressions can be derived.

**Gaussian case.** We assume that $p_0 \sim \mathcal{N}(0, \sigma^2 \mathrm{Id})$ and consider $p_{t|0}(x_t|x_0) = \mathcal{N}(x_t; \alpha_t x_0, \sigma_t^2 \mathrm{Id})$. In that case it can be shown that

$$p_{0|t}(x_0|x_t) = \mathcal{N}(x_0; \alpha_t \sigma^2/(\alpha_t^2 \sigma^2 + \sigma_t^2) x_t, \sigma^2(1 - \alpha_t^2 \sigma^2/(\alpha_t^2 \sigma^2 + \sigma_t^2))\mathrm{Id}).$$

This can be rewritten as

$$p_{0|t}(x_0|x_t) = \mathcal{N}\left(x_0; \frac{x_t}{1+u_t}, \frac{\sigma^2 u_t}{1+u_t} \mathrm{Id}\right),$$

where $u_t = \sigma_t^2/\alpha_t^2 \sigma^2$. Note that $\lim_{t \to 0} u_t = 0$ and $\lim_{t \to 1} u_t = +\infty$. First, we compute the mean in both cases.

**Proposition D.9:** *We have that*

$$\mathbb{E}[\mathcal{I}_{n,m,\mathrm{joint}}] = \sqrt{2}\mathbb{E}[\|Z\|] \int_0^1 w_t(\sigma + (\alpha_t^2 \sigma^2 + \sigma_t^2)^{1/2})\mathrm{d}t.$$

*In addition, we have that*

$$\mathbb{E}[\mathcal{I}_{n,m}] = \sqrt{2}\sigma\mathbb{E}[\|Z\|] \int_0^1 w_t \left(\frac{u_t}{1+u_t}\right)^{1/2} \mathrm{d}t.$$

Then, we compute the variance terms.

**Proposition D.10:** *We have that*

$$\lim_{m \to +\infty} \mathrm{Var}(\mathcal{I}_{n,m,\mathrm{joint}}) = \frac{2\mathbb{E}[\|Z\|]^2}{n} \left[ \int_0^1 w_t^2(\sigma + (\alpha_t^2 \sigma^2 + \sigma_t^2)^{1/2})^2 \mathrm{d}t - \left( \int_0^1 w_t(\sigma + (\alpha_t^2 \sigma^2 + \sigma_t^2)^{1/2})\mathrm{d}t \right)^2 \right].$$

*Similarly, we have that*

$$\lim_{m \to \infty} \mathrm{Var}(\mathcal{I}_{n,m}) = \frac{2\mathbb{E}[\|Z\|]^2 \sigma^2}{n} \left[ \int_0^1 w_t^2 \frac{u_t}{1+u_t} \mathrm{d}t - \left( \int_0^1 w_t \left(\frac{u_t}{1+u_t}\right)^{1/2} \mathrm{d}t \right)^2 \right].$$

Combining Proposition D.9 and Proposition D.10, we get the following result.

**Proposition D.11:** *We have that*

$$\mathrm{SNR}(\mathcal{I}_{n,\mathrm{joint}}) = n \frac{\left(\int_0^1 w_t(\sigma + (\alpha_t^2\sigma^2 + \sigma_t^2)^{1/2})\mathrm{d}t\right)^2}{\int_0^1 w_t^2(\sigma + (\alpha_t^2\sigma^2 + \sigma_t^2)^{1/2})^2\mathrm{d}t - \left(\int_0^1 w_t(\sigma + (\alpha_t^2\sigma^2 + \sigma_t^2)^{1/2})\mathrm{d}t\right)^2}.$$

*Similarly, we have*

$$\mathrm{SNR}(\mathcal{I}_n) = n \frac{\left(\int_0^1 w_t \left(\frac{u_t}{1+u_t}\right)^{1/2} \mathrm{d}t\right)^2}{\int_0^1 w_t^2 \frac{u_t}{1+u_t}\mathrm{d}t - \left(\int_0^1 w_t \left(\frac{u_t}{1+u_t}\right)^{1/2} \mathrm{d}t\right)^2}.$$

This concludes the proof of Proposition 4.2. In Figure 2, we show that in a setting where the weighting is chosen to follow the sigmoid one of (Kingma et al., 2021; Hoogeboom et al., 2023), we obtain that $\mathrm{SNR}(\mathcal{I}_{n,\mathrm{joint}}) \leq \mathrm{SNR}(\mathcal{I}_n)$ for a larger range of $\sigma$.

# E. Diffusion-compatible kernels

In this section, we prove the results of Section 4.3. In particular, we prove Proposition 4.3. We recall that for a continuous negative kernel $\rho_c$ with $c \in \mathbb{R}$ a hyperparameter of the scoring rule, we say that $\rho_c$ is diffusion compatible, if there exist $c^\star \in [-\infty, +\infty]$ and $f : \mathbb{R} \to \mathbb{R}$ such that

$$\lim_{c \to c^\star} f(c) D_{\rho_c}(p, q) = \|\mathbb{E}_p[X] - \mathbb{E}_q[X]\|^2.$$

We also recall some of the kernels we use.

$$k_{\mathrm{imq}}(x, x') = (\|x - x'\|^2 + c)^{-1/2}, \tag{28a}$$
$$k_{\mathrm{rbf}}(x, x') = \exp\left[-\|x - x'\|^2/2\sigma^2\right], \tag{28b}$$
$$k_{\mathrm{exp}}(x, x') = \exp\left[-\|x - x'\|/\sigma\right]. \tag{28c}$$

In particular, we prove the following proposition.

**Proposition E.1:** *Assume that $\rho = -k$ with $k$ given by (28a) or (28b). Then the scoring rule is diffusion compatible. In particular, if $k$ is given by (28a), we have that $c^\star = +\infty$ and $f(c) = 2c$. If $k$ is given by (28b), we have that $c^\star = +\infty$ and $f(c) = 2c$.*

*Proof.* First, we have that for any continuous positive kernel $k$, with $\rho = -k$, we get

$$\begin{aligned}
D_\rho(p, q) &= S(q, q) - S(p, q) \\
&= \mathbb{E}_{q \otimes q}[-k(Y, Y')] - \frac{1}{2}\mathbb{E}_{q \otimes q}[-k(Y, Y')] - \mathbb{E}_{p \otimes q}[-k(X, Y)] + \frac{1}{2}\mathbb{E}_{p \otimes p}[-k(X, X')] \\
&= \mathbb{E}_{p \otimes q}[k(X, Y)] - \frac{1}{2}\mathbb{E}_{p \otimes p}[k(X, X')] - \frac{1}{2}\mathbb{E}_{q \otimes q}[k(Y, Y')].
\end{aligned}$$

In addition, we have that for any $u, v \in \mathbb{R}$

$$uD_\rho(p, q) = \mathbb{E}_{p \otimes q}[u(k(X, Y) - v)] - \frac{1}{2}\mathbb{E}_{p \otimes p}[u(k(X, X') - v)] - \frac{1}{2}\mathbb{E}_{q \otimes q}[u(k(Y, Y') - v)].$$

Let $k$ be given by (28a), i.e., for any $x, x' \in \mathbb{R}^d$, we have that $k_{\mathrm{imq}}(x, x') = (\|x - x'\|^2 + c)^{-1/2}$. Let $c > 0$, $u = 2c$ and

$v = c^{-1/2}$ and consider $(c_n)_{n \in \mathbb{N}}$ such that $c_n \to +\infty$. We have that for any $x, x'$ and $c > 0$

$$
\begin{aligned}
|u(k(x, x') - v)| &= 2c^{1/2} - 2c(\|x - x'\|^2 + c)^{-1/2} \\
&= \frac{2c^{1/2}(\|x - x'\|^2 + c)^{1/2} - 2c}{(\|x - x'\|^2 + c)^{1/2}} \\
&\leq \frac{2c^{1/2}\|x - x'\|}{(\|x - x'\|^2 + c)^{1/2}} \\
&\leq \frac{2\|x - x'\|}{(\|x - x'\|^2/c + 1)^{1/2}} \leq 2\|x - x'\|.
\end{aligned}
$$

In addition, we have that for any $x, x'$ and $n \in \mathbb{N}$

$$
2c_n^{1/2} - 2c_n(\|x - x'\|^2 + c_n)^{-1/2} = \frac{2 - 2(\|x - x'\|^2/c_n + 1)^{1/2}}{(\|x - x'\|^2/c_n + 1)^{1/2}}.
$$

Therefore, letting $n \to +\infty$ we have that

$$
\lim_{n \to +\infty} 2c_n^{1/2} - 2c_n(\|x - x'\|^2 + c_n)^{-1/2} = \|x - x'\|^2.
$$

Hence, setting $f(c) = 2c$, and using the dominated convergence theorem, we get that

$$
\begin{aligned}
\lim_{c \to +\infty} f(c)D_\rho(p, q) &= \mathbb{E}_{p \otimes q}[\|X - Y\|^2] - \frac{1}{2}\mathbb{E}_{p \otimes p}[\|X - X'\|^2] - \frac{1}{2}\mathbb{E}_{q \otimes q}[\|Y - Y'\|^2] \\
&= \|\mathbb{E}_p[X] - \mathbb{E}_q[X]\|^2,
\end{aligned}
$$

which concludes the first part of the proof. Next, we consider $k$ be given by (28b), i.e., for any $x, x' \in \mathbb{R}^d$, we have that $k_{\exp}(x, x') = \exp[-\|x - x'\|^2/\sigma^2]$. Let $u = 2\sigma^2$ and $v = 1$. Then, we have that

$$
|u(k(x, x') - v)| = 2\sigma^2(1 - \exp[-\|x - x'\|^2/2\sigma^2]) \leq \|x - x'\|^2.
$$

Now consider $(c_n)_{n \in \mathbb{N}}$ such that $c_n \to +\infty$. We have that

$$
\lim_{n \to +\infty} 2c_n^2(1 - \exp[-\|x - x'\|^2/2c_n^2]) = \|x - x'\|^2.
$$

Hence, setting $f(c) = 2c$, and using the dominated convergence theorem, we get that

$$
\begin{aligned}
\lim_{c \to +\infty} f(c)D_\rho(p, q) &= \mathbb{E}_{p \otimes q}[\|X - Y\|^2] - \frac{1}{2}\mathbb{E}_{p \otimes p}[\|X - X'\|^2] - \frac{1}{2}\mathbb{E}_{q \otimes q}[\|Y - Y'\|^2] \\
&= \|\mathbb{E}_p[X] - \mathbb{E}_q[X]\|^2,
\end{aligned}
$$

which concludes the second part of the proof. $\qquad \square$

Finally, we highlight that while the exponential kernel (28c) is not diffusion compatible it satisfies another compatibility rule. More precisely, we can show the following result. The proof is similar to the proof of Proposition E.1, in the case of $k_{\mathrm{rbf}}$.

> **Proposition E.2:** *Assume that $\rho = -k$ with $k$ given by (28c), i.e., $k(x, x') = \exp[-\|x - x'\|/\sigma]$. In that case, upon setting $f(c) = c$ and $c^\star = +\infty$, we get that*
>
> $$
> \lim_{c \to +\infty} f(c)D_{\rho_c}(p, q) = D_{\rho^\star}(p, q),
> $$
>
> *where $\rho^\star(x, x') = \|x - x'\|$, i.e. we recover the energy distance with $\beta = 1$.*

# F. DDIM updates from a Stochastic Differential Equation perspective

In this section, we show that one can *exactly* recover the DDIM updates from the stochastic process point of view with a careful choice of forward process.

We first show that different choices of forward process yields different interpolation densities $p_{s|0,t}$ in Appendix F.1. With a careful choice of the forward process and appropriate parameters, we show that we can recover DDIM updates (Song et al., 2021a) in Appendix F.2.

## F.1. General forward processes

In this section, we consider general forward processes. The main property of all those forward processes $(\mathbf{X}_t)_{t\in[0,1]}$ is that they will satisfy

$$\mathbf{X}_t = \alpha_t \mathbf{X}_0 + \sigma_t \mathbf{Z}', \qquad \mathbf{X}_0 \sim p_0, \tag{29}$$

with $\mathbf{Z}' \sim \mathcal{N}(0, \mathrm{Id})$. First, we define

$$\varepsilon_t^2 = 2\varepsilon^2 \alpha_t \sigma_t \partial_t(\sigma_t/\alpha_t) = \varepsilon^2 g_t^2.$$

We also consider $\mathbf{Z} \sim \mathcal{N}(0, \mathrm{Id})$ and introduce the dynamics $(\mathbf{X}_t)_{t\in[0,1]}$

$$\mathrm{d}\mathbf{X}_t = \left[\partial_t \log(\alpha_t)\mathbf{X}_t + \frac{g_t^2}{2\sigma_t}(1-\varepsilon^2)\mathbf{Z}\right]\mathrm{d}t + \varepsilon g_t \mathrm{d}\mathbf{B}_t, \tag{30}$$

where $(\mathbf{B}_t)_{t\in[0,1]}$ is a $d$-dimensional Brownian motion. In (30), we identify different forward processes which are all non-Markov except in the case where $\varepsilon = 1$. The remarkable property of these forward processes is that they all recover the interpolation (29). This means that the forward trajectories might be very different but they all admit the same *marginal* distributions. More formally, we get the following result.

> **Proposition F.1:** *For any $t \in [0,1]$, let $\varepsilon_t \in [0,1]$. Let $\mathbf{Z} \sim \mathcal{N}(0, \mathrm{Id})$ and $\mathbf{X}_0 \sim p_0$. Additionally, assume that the following SDE admits a solution*
>
> $$\mathrm{d}\mathbf{X}_t = \left[\partial_t \log(\alpha_t)\mathbf{X}_t + \frac{g_t^2}{2\sigma_t}(1-\varepsilon^2)^{1/2}\mathbf{Z}\right]\mathrm{d}t + \varepsilon g_t \mathrm{d}\mathbf{B}_t,$$
>
> *where $(\mathbf{B}_t)_{t\in[0,1]}$ is a $d$-dimensional Brownian motion, $p(z) = \mathcal{N}(z; 0, \mathrm{Id})$ and $p_t(\cdot|z)$ is the density of $\mathbf{X}_t$ conditionally to $\mathbf{Z} = z$. Denote $p_t(x_t) = \int_{\mathbb{R}^d} p_t(x_t|z)p(z)\mathrm{d}z$. We have that for any $t \in [0,1]$, $\alpha_t \mathbf{X}_0 + \sigma_t \mathbf{Z} \sim p_t$.*

**Explicit integration.** We can solve exactly (30) and its solution is given for any $t \geq s$ by

$$\mathbf{X}_t = \frac{\alpha_t}{\alpha_s}\mathbf{X}_s + (1-\varepsilon^2)^{1/2}\left(\sigma_t - \frac{\alpha_t}{\alpha_s}\sigma_s\right)\mathbf{Z} + \varepsilon\left(\sigma_t^2 - \frac{\alpha_t^2}{\alpha_s^2}\sigma_s^2\right)^{1/2}\tilde{\mathbf{Z}},$$

with $\tilde{\mathbf{Z}} \sim \mathcal{N}(0, \mathrm{Id})$. This can also be rewritten as

$$\begin{aligned}
\mathbf{X}_t - \alpha_t\mathbf{X}_0 &= \frac{\alpha_t}{\alpha_s}(\mathbf{X}_s - \alpha_s\mathbf{X}_0) + (1-\varepsilon^2)^{1/2}\left(\sigma_t - \frac{\alpha_t}{\alpha_s}\sigma_s\right)\mathbf{Z} + \varepsilon\left(\sigma_t^2 - \frac{\alpha_t^2}{\alpha_s^2}\sigma_s^2\right)^{1/2}\tilde{\mathbf{Z}} \\
&= \frac{\alpha_t}{\alpha_s}[(1-\varepsilon^2)^{1/2}\sigma_s\mathbf{Z} + \varepsilon\sigma_s\hat{\mathbf{Z}}] + (1-\varepsilon^2)^{1/2}\left(\sigma_t - \frac{\alpha_t}{\alpha_s}\sigma_s\right)\mathbf{Z} + \varepsilon\left(\sigma_t^2 - \frac{\alpha_t^2}{\alpha_s^2}\sigma_s^2\right)^{1/2}\tilde{\mathbf{Z}} \\
&= (1-\varepsilon^2)^{1/2}\sigma_t\mathbf{Z} + \varepsilon\sigma_s\frac{\alpha_t}{\alpha_s}\hat{\mathbf{Z}} + \varepsilon\left(\sigma_t^2 - \frac{\alpha_t^2}{\alpha_s^2}\sigma_s^2\right)^{1/2}\tilde{\mathbf{Z}},
\end{aligned} \tag{31}$$

with $\mathbf{X}_s = \alpha_s\mathbf{X}_0 + \sigma_s(1-\varepsilon)^{1/2}\mathbf{Z} + \sigma_s\varepsilon\hat{\mathbf{Z}}$, with $\mathbf{Z}, \tilde{\mathbf{Z}}$ and $\hat{\mathbf{Z}}$ independent Gaussian random variables with zero mean and identity covariance. We can now compute the probability distribution $p_{s,t|0}$. This will allow us to compute $p_{s|0,t}$ using Gaussian posteriors. Using (31), we have

$$(\mathbf{X}_s, \mathbf{X}_t)|\mathbf{X}_0 \sim \mathcal{N}\left(\begin{pmatrix}\alpha_s\mathbf{X}_0 \\ \alpha_t\mathbf{X}_0\end{pmatrix}, \begin{pmatrix}\sigma_s^2\mathrm{Id} & C_t \\ C_t & \sigma_t^2\mathrm{Id}\end{pmatrix}\right),$$

with

$$C_t = \left[\varepsilon^2\frac{\alpha_t}{\alpha_s} + (1-\varepsilon^2)\frac{\sigma_t}{\sigma_s}\right]\sigma_s^2\mathrm{Id} = \left[\varepsilon^2\frac{\alpha_t}{\alpha_s}\sigma_s^2 + (1-\varepsilon^2)\sigma_t\sigma_s\right]\mathrm{Id}.$$

Now, let us compute the obtained posterior.

$$\mathbb{E}[\mathbf{X}_s|\mathbf{X}_0, \mathbf{X}_t] = \alpha_s \mathbf{X}_0 + \left[\varepsilon^2 \frac{\alpha_t}{\alpha_s} \frac{\sigma_s^2}{\sigma_t^2} + (1 - \varepsilon^2) \frac{\sigma_s}{\sigma_t}\right](\mathbf{X}_t - \alpha_t \mathbf{X}_0)$$

$$= \left[\varepsilon^2 \frac{\alpha_t}{\alpha_s} \frac{\sigma_s^2}{\sigma_t^2} + (1 - \varepsilon^2) \frac{\sigma_s}{\sigma_t}\right] \mathbf{X}_t + \alpha_s \left(1 - \left[\varepsilon^2 \frac{\alpha_t^2}{\alpha_s^2} \frac{\sigma_s^2}{\sigma_t^2} + (1 - \varepsilon^2) \frac{\sigma_s}{\sigma_t} \frac{\alpha_t}{\alpha_s}\right]\right) \mathbf{X}_0.$$

Finally, we have that

$$\mathrm{Cov}(\mathbf{X}_s|\mathbf{X}_t, \mathbf{X}_0) = \sigma_s^2 \mathrm{Id} - \left[\varepsilon^2 \frac{\alpha_t}{\alpha_s} \sigma_s^2 + (1 - \varepsilon^2) \sigma_t \sigma_s\right]^2 \frac{1}{\sigma_t^2} \mathrm{Id}$$

$$= \sigma_s^2 \left(1 - \left[\varepsilon^2 \frac{\alpha_t}{\alpha_s} \frac{\sigma_s}{\sigma_t} + (1 - \varepsilon^2)\right]^2\right).$$

Note that in the special case where $\varepsilon = 0$, we recover a deterministic sampler. To summarize, we have the following result which justifies (4).

---

**Proposition F.2:** *Let $\varepsilon \in [0, 1]$ and $(\mathbf{X}_t)_{t\in[0,1]}$ given by Proposition F.1. Then, we have that for any $s, t \in [0, 1]$ with $s \le t$,*

$$\mathbf{X}_s = (\varepsilon^2 r_{1,2}(s, t) + (1 - \varepsilon^2) r_{0,1}) \mathbf{X}_t$$
$$+ \alpha_s (1 - \varepsilon^2 r_{2,2}(s, t) - (1 - \varepsilon^2) r_{1,1}(s, t)) \mathbf{X}_0$$
$$+ \sigma_s (1 - (\varepsilon^2 r_{1,1}(s, t) + (1 - \varepsilon^2))^2)^{1/2} \tilde{\mathbf{Z}}, \tag{32}$$

*with $\tilde{\mathbf{Z}} \sim \mathrm{N}(0, \mathrm{Id})$ and where*

$$r_{i,j}(s, t) = \frac{\alpha_t^i}{\alpha_s^i} \frac{\sigma_s^j}{\sigma_t^j}.$$

---

**Limiting behavior.** Looking at (32) one can wonder if we recover the SDE framework when we let $s \to t$. This is indeed the case as established below. We will make use of the following result

$$\lim_{s \to t} \frac{r_{i,j}(s, t) - 1}{t - s} = i \partial_t \log(\alpha_t) - j \partial_t \log(\sigma_t).$$

Therefore, we get that taking the limit in (32) we have

$$\mathrm{d}\mathbf{X}_t = \left[(\varepsilon^2(\partial_t \log(\alpha_t) - 2\partial_t \log(\sigma_t)) - (1 - \varepsilon^2)\partial_t \log(\sigma_t))\mathbf{X}_t\right.$$
$$\left. - \alpha_t(\varepsilon^2(2\partial_t \log(\alpha_t) - 2\partial_t \log(\sigma_t)) + (1 - \varepsilon^2)(\partial_t \log(\alpha_t) - \partial_t \log(\sigma_t)))\mathbf{X}_0\right] \mathrm{d}t$$
$$+ \sigma_t[-2\varepsilon^2(\partial_t \log(\alpha_t) - \partial_t \log(\sigma_t))]^{1/2} \mathrm{d}\mathbf{B}_t.$$

It can easily be shown that

$$-2\sigma_t^2(\partial_t \log(\alpha_t) - \partial_t \log(\sigma_t)) = 2\alpha_t \sigma_t \partial_t(\sigma_t/\alpha_t) = g_t^2. \tag{33}$$

In addition, we have

$$\mathrm{d}\mathbf{X}_t = \left[-\partial_t \log(\alpha_t)\mathbf{X}_t + (\varepsilon^2(2\partial_t \log(\alpha_t) - 2\partial_t \log(\sigma_t)) + (1 - \varepsilon^2)(\partial_t \log(\alpha_t) - \partial_t \log(\sigma_t)))\mathbf{X}_t\right.$$
$$\left. - \alpha_t(\varepsilon^2(2\partial_t \log(\alpha_t) - 2\partial_t \log(\sigma_t)) + (1 - \varepsilon^2)(\partial_t \log(\alpha_t) - \partial_t \log(\sigma_t)))\mathbf{X}_0\right] \mathrm{d}t$$
$$+ \sigma_t[-2\varepsilon^2(\partial_t \log(\alpha_t) - \partial_t \log(\sigma_t))]^{1/2} \mathrm{d}\mathbf{B}_t$$
$$= \left[-\partial_t \log(\alpha_t)\mathbf{X}_t + (\varepsilon^2(2\partial_t \log(\alpha_t) - 2\partial_t \log(\sigma_t))\right.$$
$$\left. + (1 - \varepsilon^2)(\partial_t \log(\alpha_t) - \partial_t \log(\sigma_t)))(\mathbf{X}_t - \alpha_t \mathbf{X}_0)\right] \mathrm{d}t + g_t \mathrm{d}\mathbf{B}_t$$
$$= \left[-\partial_t \log(\alpha_t)\mathbf{X}_t + (1 + \varepsilon^2)(\partial_t \log(\alpha_t) - \partial_t \log(\sigma_t)))(\mathbf{X}_t - \alpha_t \mathbf{X}_0)\right] \mathrm{d}t + \varepsilon g_t \mathrm{d}\mathbf{B}_t.$$

Now, combining this result and (33) we get that

$$d\mathbf{X}_t = \left[ -\partial_t \log(\alpha_t) + g_t^2 \frac{1+\varepsilon^2}{2} \frac{\alpha_t \mathbf{X}_0 - \mathbf{X}_t}{\sigma_t^2} \right] dt + \varepsilon g_t d\mathbf{B}_t.$$

Taking the conditional expectation in this expression, we recover the score $\nabla \log p_t$ and therefore

$$d\mathbf{X}_t = \left[ -\partial_t \log(\alpha_t) + g_t^2 \frac{1+\varepsilon^2}{2} \nabla \log p_t(\mathbf{X}_t) \right] dt + \varepsilon g_t d\mathbf{B}_t.$$

Therefore, we recover the usual sampler derived from the stochastic process point of view by taking the limit $s \to t$ in (32).

## F.2. Connection with DDIM

We recall that in DDIM (Song et al., 2021a), the authors assume that the schedule $(\alpha_t, \sigma_t)$ satisfies $\alpha_t^2 + \sigma_t^2 = 1$. In this case, they consider

$$p(x_s | x_0, x_t) = \mathcal{N}(x_s; \hat{\mu}_{s,t}(x_0, x_t), \hat{\Sigma}_{s,t}),$$

with

$$\hat{\mu}_{s,t}(x_0, x_t) = \sqrt{\frac{1 - \alpha_s^2 - \eta_{s,t}^2}{1 - \alpha_t^2}} x_t + \left[ \alpha_s - \alpha_t \sqrt{\frac{1 - \alpha_s^2 - \eta_{s,t}^2}{1 - \alpha_t^2}} \right] x_0,$$

$$\hat{\Sigma}_{s,t} = \eta_{s,t}^2 \mathrm{Id}. \tag{34}$$

Note that in (Song et al., 2021a), $\alpha_t^2$ is replaced with $\alpha_t$ and $\eta_{s,t}$ is denoted $\sigma_t$. Recall that in our introduction of diffusion models in Section 2.1, see (4) we have

$$\mu_{s,t}(x_0, x_t) = (\varepsilon^2 r_{1,2}(s,t) + (1 - \varepsilon^2) r_{0,1}) x_t$$
$$+ \alpha_s (1 - \varepsilon^2 r_{2,2}(s,t) - (1 - \varepsilon^2) r_{1,1}(s,t)) x_0,$$
$$\Sigma_{s,t} = \sigma_s^2 (1 - (\varepsilon^2 r_{1,1}(s,t) + (1 - \varepsilon^2))^2) \mathrm{Id}. \tag{35}$$

The following result proves that the updates of DDIM (34) and the ones we present (35) are identical upon identification of one parameter.

**Proposition F.3:** *Assume that $\eta_{s,t}^2 = \sigma_s^2 (1 - (\varepsilon^2 r_{1,1}(s,t) + (1-\varepsilon^2))^2)$. Then, we have that $\Sigma_{s,t} = \hat{\Sigma}_{s,t}$ and $\mu_{s,t}(x_0, x_t) = \hat{\mu}_{s,t}(x_0, x_t)$.*

*Proof.* First, we have that

$$\sqrt{\frac{1 - \alpha_s^2 - \eta_{s,t}^2}{1 - \alpha_t^2}} = \sqrt{\frac{1 - \alpha_s^2 - \sigma_s^2(1 - (\varepsilon^2 r_{1,1}(s,t) + (1-\varepsilon^2))^2)}{1 - \alpha_t^2}}$$

$$= \sqrt{\frac{1 - \alpha_s^2 - (1 - \alpha_s^2)(1 - (\varepsilon^2 r_{1,1}(s,t) + (1-\varepsilon^2))^2)}{1 - \alpha_t^2}}$$

$$= \sqrt{r_{0,2}(s,t)(1 - (1 - (\varepsilon^2 r_{1,1}(s,t) + (1-\varepsilon^2))^2))}$$

$$= r_{0,1}(s,t)(\varepsilon^2 r_{1,1}(s,t) + (1-\varepsilon^2))$$

$$= \varepsilon^2 r_{1,2}(s,t) + (1-\varepsilon^2) r_{0,1}(s,t).$$

Hence, we get that

$$
\begin{aligned}
\hat{\mu}_{s,t}(x_0, x_t) &= \sqrt{\frac{1 - \alpha_s^2 - \eta_{s,t}^2}{1 - \alpha_t^2}}\, x_t + \left[\alpha_s - \alpha_t \sqrt{\frac{1 - \alpha_s^2 - \eta_{s,t}^2}{1 - \alpha_t^2}}\right] x_0 \\
&= \sqrt{\frac{1 - \alpha_s^2 - \eta_{s,t}^2}{1 - \alpha_t^2}}\, x_t + \alpha_s \left[1 - r_{1,0}(s,t)\sqrt{\frac{1 - \alpha_s^2 - \eta_{s,t}^2}{1 - \alpha_t^2}}\right] x_0 \\
&= (\varepsilon^2 r_{1,2}(s,t) + (1 - \varepsilon^2) r_{0,1}(s,t)) x_t + \alpha_s \left[1 - \varepsilon^2 r_{2,2}(s,t) - (1 - \varepsilon^2) r_{1,1}(s,t)\right] x_0 = \mu_{s,t}(x_0, x_t),
\end{aligned}
$$

which concludes the proof. $\qquad \square$

In our work, we call $\varepsilon$ the *churn* parameter. In (Song et al., 2021a, Equation (16)), we have that

$$
\eta_{s,t}^2 = \eta^2 \frac{1 - \alpha_s^2}{1 - \alpha_t^2}\left(1 - \frac{\alpha_t^2}{\alpha_s^2}\right) = \eta^2(r_{0,2}(s,t) - r_{2,2}(s,t)).
$$

In that case $\eta$ is another churn parameter. However, we can write the following relation between those two parameters. In particular, in both cases we have that we recover if DDPM if $\varepsilon = 1$ (or equivalently if $\eta = 1$) and DDIM if $\varepsilon = 0$ (or equivalently if $\eta = 0$).

**Proposition F.4:** *We have that*

$$
\eta^2 = \frac{\sigma_s^2(1 - (\varepsilon^2 r_{1,1}(s,t) + (1 - \varepsilon^2))^2)}{r_{0,2}(s,t) - r_{2,2}(s,t)}.
$$

*Similarly, we have that*

$$
\varepsilon^2 = \frac{1 - \left[1 - \eta^2 \frac{r_{0,2}(s,t) - r_{2,2}(s,t)}{\sigma_s^2}\right]^{1/2}}{1 - r_{1,1}(s,t)}.
$$

*Remarkably, we have that $\varepsilon = 0$ if and only if $\eta = 0$ and $\varepsilon = 1$ if and only if $\eta = 1$.*

*Proof.* Proving that $\varepsilon = 0$ if and only if $\eta = 0$ is straightforward and left to the reader. Now assume that $\varepsilon = 1$, we are going to prove that $\eta = 1$.

$$
\begin{aligned}
\eta^2 &= \frac{\sigma_s^2(1 - (\varepsilon^2 r_{1,1}(s,t) + (1 - \varepsilon^2))^2)}{r_{0,2}(s,t) - r_{2,2}(s,t)} \\
&= \frac{\sigma_s^2 - \sigma_s^2 r_{2,2}(s,t)}{r_{0,2}(s,t) - r_{2,2}(s,t)} \\
&= \frac{\frac{(1 - \alpha_s^2)(1 - \alpha_t^2)\alpha_s^2 - (1 - \alpha_s^2)^2 \alpha_t^2}{\alpha_s^2(1 - \alpha_t^2)}}{r_{0,2}(s,t) - r_{2,2}(s,t)} \\
&= \frac{\frac{(1 - \alpha_s^2)(\alpha_s^2 - \alpha_t^2)}{\alpha_s^2(1 - \alpha_t^2)}}{r_{0,2}(s,t) - r_{2,2}(s,t)} \\
&= \frac{r_{0,2}(s,t)(1 - r_{2,0}(s,t))}{r_{0,2}(s,t) - r_{2,2}(s,t)} = 1.
\end{aligned}
$$

Similarly, we get that $\eta = 1$ implies that $\varepsilon = 1$. $\qquad \square$

# G. Pseudo-code for loss function

## G.1. Some replication utils

Given a batch of $[x^1, x^2, x^4, x^5]$, assuming that $n = 4$ and $m = 2$, this function will output (before the reshape operation)

$$\hat{x} = \begin{pmatrix} x^1 & x^1 \\ x^2 & x^2 \\ x^3 & x^3 \\ x^4 & x^4 \end{pmatrix}.$$

```
1  def replicate_fn(n: int, m: int, x: chex.Array) -> chex.Array:
2    batch_size, data_shape = x.shape[0], x.shape[1:]
3    x = x[:n]
4    x = jnp.reshape(x, (n, 1, *data_shape))
5    x = jnp.tile(x, (1, m) + (1,) * len(data_shape))
6    x = jnp.reshape(x, (n * m, *data_shape))
7    return x
```

We also assume that we are provided a function `split_fn` such that given $x$ with shape $(nm, ...)$ the output of `split_fn` has shape $(n, m, ...)$.

### G.2. Loss function

We assume that we are given two functions `compute_rho_fn` and `compute_rho_diagonal_fn` such that given $x$ and $y$ with shape $(n, m, ...)$ the output of `compute_rho_diagonal_fn` is $\frac{1}{n} \sum_{i=1}^{n} \frac{1}{m} \sum_{j=1}^{m} \rho(x_{i,j}, y_{i,j})$ and the output of `compute_rho_fn` is $\frac{1}{n} \sum_{i=1}^{n} \frac{1}{m(m-1)} \sum_{j,j'=1, j\neq j'}^{m} \rho(x_{i,j}, y_{i,j'})$. Function $\rho$ is consistent with the notation defined in Section 2.2.

In addition, we assume that we have access to a function `add_noise_fn` such that given $t$ of shape $(n,)$ and $x_0$ of shape $(n, ...)$ it outputs $x_t = \alpha_t x_0 + \sigma_t z$ where $z$ has the same shape as $x$ and has independent Gaussian entries $\mathcal{N}(0, 1)$.

Finally, we assume that we have access to a function `apply_fn` such that given $t$ of shape $(n,)$ and $x_t$ of shape $(n, ...)$ it outputs $\hat{x}_\theta(t, x_t)$.

```
1  def loss_fn(t: jnp.Array, x0: jnp.Array) -> jnp.Array:
2
3    # add noise and replicate
4    xt = add_noise_fn(key=key, t=t, x0=x0)
5    key, _ = jax.random.split(key)
6    x0_population = replicate_fn(n=n, m=m, x=x0)
7    t_population = replicate_fn(n=n, m=m, x=t)
8    xt_population = replicate_fn(n=n, m=m, x=xt)
9
10   # compute prediction
11   eps_population = jax.random.normal(key=key, shape=xt_population.shape)
12   key, _ = jax.random.split(key)
13   output_population = apply_fn(t=t_population, xt=xt_population, eps=eps_population)
14
15   # split the populations
16   x0_population = split_fn(n=n, m=m, x=x0_population)
17   output_population = split_fn(n=n, m=m, x=output_population)
18
19   # compute confinement term
20   confinement = compute_rho_diagonal_fn(x=x0_population, y=output_population)
21
22   # compute interaction term for predictions
23   interaction_prediction = compute_rho_fn(x=output_population, y=output_population)
24
25   # Generalized kernel score
26   score = 0.5 * lbda * interaction_prediction - confinement
27
28   # Our aim to maximize the score, so the loss is negative score
29   loss = -score
30
31   return loss
```

In this function, we have focused on the $x_0$-prediction and have omitted the weighting for simplicity.

## H. Extended related work

**Diffusion and GANs.** In (Xiao et al., 2022), the authors propose to replace the conditional mean estimator of diffusion models with a GAN. This allows to model bigger steps in the denoising process. Recall that, seeing diffusion models as hierarchical VAEs, the ELBO is given by

$$\log p_\theta(x) \geq -\sum_{k=1}^{N} \mathrm{KL}(p(x_{t_{k-1}}|x_{t_k})|p_\theta(x_{t_{k-1}}|x_{t_k})) + C,$$

where $C$ is a constant. In (Xiao et al., 2022), the relative entropy $\mathrm{KL}(q(x_{t-1}|x_t)|p_\theta(x_{t-1}|x_t))$ is replaced by an adversarial loss. Namely, the authors consider

$$\mathcal{L}(\theta, \phi) = \sum_{k=1}^{N} \left( \mathbb{E}_{p(x_{t_{k-1}}|x_{t_k})}[-\log(D_\phi(t, x_{t_{k-1}}, x_{t_k}))] + \mathbb{E}_{p_\theta(x_{t_{k-1}}|x_{t_k})}[-\log(1 - D_\phi(t, x_{t_{k-1}}, x_{t_k}))] \right).$$

In that case, $D_\phi$ is a *discriminator* and $p_\theta$ is a *generator*. The main differences with our setting is that in our case, we do not train a discriminator. In our setting the discriminative parameters are only given by the kernel parameters, i.e., in the case of the energy distance, the parameters are given by $\lambda$ and $\beta$ in (13) and are *fixed*. In this regards our parameterization is much more lightweight. In addition, while the transition $p_\theta(x_{t_{k-1}}|x_{t_k})$ is estimated in a similar way as in our paper, i.e.,

$$p_\theta(x_{t_{k-1}}|x_{t_k}) = \int_{\mathbb{R}^d} p_\theta(x_0|x_{t_k})p(x_{t_{k-1}}|x_0, x_{t_k})\mathrm{d}x_0,$$

the discriminator is not applied on the $x_0$ variable conditionally to $x_{t_k}$ but on the couple $(x_{t_k}, x_{t_{k-1}})$. This is similar to the differences we highlighted in Appendix D.5 when comparing joint loss and conditional loss. Note that our approach could be extended to consider kernels not on $p_{0|s}$ but instead leveraging the characterisation

$$p(x_s|x_t) = \int_{\mathbb{R}^d} p(x_u|x_t)p(x_s|x_u, x_t)\mathrm{d}x_u,$$

for $u \leq \min(s, t)$. In that case, choosing $u = 0$ is simply a special case.

**Modeling the covariance.** The importance of modeling $p_{0|t}$ in the low step regime has been highlighted in numerous papers, see (Ho et al., 2020; Nichol & Dhariwal, 2021; Bao et al., 2022a;b; Ou et al., 2024). A number of approaches aim at learning/approximating the covariance of $p_{s|t}$ or $p_{0|t}$, see (Ho et al., 2020; Nichol & Dhariwal, 2021; Bao et al., 2022a;b; Ou et al., 2024; Rozet et al., 2024). Most of the time in order to obtain a tractable approximation of this covariance they consider diagonal approximations leveraging the Hutchinson trace estimator (Hutchinson, 1989). In contrast, we do not assume a Gaussian form for $p_{0|t}$ or $p_{s|t}$. Doing so we (i) model more complex distributions than Gaussian transitions (ii) avoid the additional complexity of having to model a covariance matrix.

**Relationship to works using** MMD. Scoring rules have corresponding MMD divergences, and approaches exist that use the MMD in diffusion and particle flow models. Galashov et al. (2025) generates a sequence of distributions from a forward diffusion process. It then generates noise-dependent neural MMDs between clean and noisy data, and performs gradient flow to move particles from noise level $t$ to a lower noise level $s < t$. Compared to ours, Galashov et al. (2025) does not use a generator since it is a particle flow. Their best CIFAR-10 FID is 7.7. Aiello et al. (2024) first trains a diffusion model, and then refines/distills by coarsening the reverse timesteps, and uses MMD on CLIP features to finetune the DDM denoiser. Their CIFAR10 FID for $NFE = 10$ improves from 13.6 to 3.8, while ours is 3.19.

## I. Experimental details

### I.1. 2D experiments

We consider a target distribution $p_0$ given by a mixture of two Gaussians, $p_0 = 0.5\mathcal{N}(\mu_1, \sigma^2\mathrm{Id}) + 0.5\mathcal{N}(\mu_2, \sigma^2\mathrm{Id})$, where $\mu_1 = (3, 3)$, $\mu_2 = (-3, 3)$ and $\sigma = 0.5$. We create a dataset from this distribution by sampling 102400 points.

We train diffusion model and *distributional* diffusion models for 100k steps with batch size 128 with learning rate $1e-3$ using Adam optimizer with a cosine warmup for first 100 iterations. We use $b_1 = 0.9$, $b_2 = 0.999$, $\epsilon = 1e-8$ in Adam optimizer. On top of that, we clip the updates by their global norm (with the maximal norm being 1). We use EMA decay of $0.99$. We use the *flow matching* noise schedule (3) and we use safety epsilon $1e-2$. When training the model, we use sigmoid loss weighting as in (Kingma et al., 2021). The time is encoded via sinusoidal embedding into the dimension 2048 followed by 2 layer MLP where hidden dimension is 2048 and the output dimension is 2048, using `gelu` activation. As a backbone architecture, we use a 9-layers MLP with `gelu` activation and with hidden dimension of 64. The MLP is applied as follows. First, we apply 4 preprocessing MLP layers to the embedding of time $t$ and separately to the $x_t$. After that, we concatenate these two on the last dimension and pass through 4 MLP layers which is then followed by one output MLP layer of dimension 2. When we use the *distributional* diffusion model, the noise $\xi$ has the same dimensionality as $x_t$ and we concatenate the two along the last dimension. After passing it through the MLP, which produces a vector of dimension 4, we ignore first 2 dimension to get the output $\hat{x}_\theta(t, x_t, \xi)$ of dimension 2. We use the population size $m = 32$ (see Algorithm 1) and we use $\beta = 0.1$, $\lambda = 1$ for *distributional* model and $\beta = 2$, $\lambda = 0$ for diffusion-like variant.

For evaluation, we sample 4096 samples $X_0^i \sim p_0$ as well as samples $X_t^i | X_0^i$ from the forward process (3). For Figure 2, right, we use the MMD squared given by $D_\rho$ (10) with $\rho(x, x') = -k_{\text{rbf}}(x, x')$ (see (9b)) for $\sigma = 1$.

For each $X_t^i$, we produce 8 samples $\xi \sim \mathcal{N}(\xi, 0, \text{Id})$ and we compute standard deviation of $\hat{x}_\theta(t, x_t^i, \xi)$ over $\xi$. Then, we average the standard deviation over all the $x_t^i$. On top of that, since we know the posterior $p(X_0 | X_t)$, which is given by

$$p(x_0 | x_t) = w_1 \mathcal{N}(x_0; \nu_1, \Sigma) + w_2 \mathcal{N}(x_0; \nu_2, \Sigma),$$
$$w_k \propto \mathcal{N}(\alpha_t \mu_k, (\alpha_t^2 \sigma^2 + \sigma_t^2) \text{Id}),$$
$$\nu_k = \Sigma \left( \frac{\alpha_t}{\sigma_t^2} X_t + \frac{1}{\sigma^2} \mu_k \right),$$
$$\Sigma = \left( \frac{\alpha_t^2}{\sigma_t^2} + \frac{1}{\sigma^2} \right)^{-1} \text{Id},$$

where $\mu_k$ with $k = 1, 2$ are the means of $p_0$ and $\sigma^2$ is the variance. The values of $\alpha_t$ and $\sigma_t$ are given by the schedule (3). We produce 8 samples $X_0 \sim p(\cdot | X_t^i)$ and compute their standard deviation and then average over $X_t^i$. We report this metric in Figure 3, right. In order to produce Figure 4, we fix $x_0$ to be equal to either $\mu_1$ or to $\mu_2$. We then produce $X_t | x_0$ for each $t$ (one sample for each $t$ and $x_0$). After that, we visualize $\hat{x}_\theta(t, X_t^i, \xi)$ for $n = 4096$ samples of $\xi \sim \mathcal{N}(0, \text{Id})$.

## I.2. Image space experiments

**Architecture.** In our image space experiments we consider a $U$-net architecture (Ronneberger et al., 2015). We consider a channel of size 256 and apply the following multiplier for the channel at each level of the $U$-net, $(1, 2, 2, 2)$. At each level we consider 2 residual blocks. We apply an attention layer at the second level of the $U$-net (resolution 16 in the case of CIFAR-10 and resolution 32 in the case of CelebA and LSUN Bedrooms). We use an attention block in the bottleneck block of the $U$-net. Each residual block consists of normalizing the input, applying a $3 \times 3$ convolution block and applying a non-linearity. This is then followed by a dropout layer and a $3 \times 3$ convolution block. Finally we add the input (processed through a $1 \times 1$ convolutional block) of the residual block to this output (this is the residual connection). We use the RMSnorm for the normalization layer and GELU for the non-linearity. When using an attention layer, this is done after the convolutional residual block just described. We use a multi-head attention mechanism with 8 heads. We process the time with a sinusoidal embedding followed by a MLP layer. In the case of CelebA, we use a linear embedding layer to process the conditioning vector of shape $(40, )$ while in the case of CIFAR10 we consider an embedding matrix to process the class information. Once the time conditioning and the (optional) other conditioning have been obtained we sum them. The conditioning is used in the model by replacing the RMSnorm with an adaptive RMSnorm layer, i.e., the scale and bias of the normalization layers are obtained by passing the conditioning vector through a MLP layer. Similarly to the two dimensional setting, we concatenate $x_t$ of shape $(b, h, w, c)$ and $\xi$ of shape $(b, h, w, c)$ along the last dimension in order to get an input of shape of shape $(b, h, w, 2c)$. The output of the $U$-net is of shape of shape $(b, h, w, 2c)$ and we drop half of the channels to get a sample of shape $(b, h, w, c)$.

For latent CelebA-HQ, we follow the recipe of (Rombach et al., 2022) to train the autoencoder. We train the autoencoder with a similar architecture as the autoencoder used in LDM-4 (see (Rombach et al., 2022)). It encodes CelebA-HQ of shape $(256, 256, 3)$ into latent code of shape $(64, 64, 3)$. Then, for latent diffusion model and *distributional* latent model, we use a

similar architecture as LDM-4 except that we concatenate the latent $x_t$ and $\xi$ over the last dimension. More precisely, we consider a channel of size 256 and apply the following multiplier for the channel at each level of the $U$-net, $(1, 2, 2, 2)$. At each level we consider 2 residual blocks. We apply an attention layer at the foruth level of the $U$-net. The structure of the residual block is identical as in image space. Similarly, we use the same type of normalization layer and non-linearity. We use a multi-head attention mechanism with 8 heads.

**Training and evaluation details.** We train all the pixel-space models for $1e6$ steps with batch size 256 for CIFAR-10, batch size of 64 for CelebA, LSUN Bedrooms. The latent space model consists of two-stage training regime, where we first train latent autoencoder for $2e6$ steps with batch size 16. It is then followed by training a latent diffusion model for $2e6$ steps with batch size 16.

For all the experiments, we use the Adam optimizer with additional norm clipping (maximal norm equal to 1) with the learning rate equal to $1e - 4$ and an additional cosine warmup for 100 first iterations.

Our autoencoder on CelebA-HQ is trained with $0.1$ coefficient on adversarial loss, 100 coefficient on generator loss and 100 coefficient on codebook, from the VQ-VAE (van den Oord et al., 2017) loss. We use a $\beta$-VAE with $\beta = 1e - 6$.

For all the models, we sweep over weighting $w_t$ (see (6), (13)), where we either do not use any weighting or we use sigmoid weighting, see (Kingma et al., 2021), weighting with bias parameters $\{-2, -1, 0, 1, 2\}$. When we train classical diffusion model, we always use velocity as a prediction target, i.e., the target is $X_0 - X_1$. When we train *distributional* model, we use $X_0$ as prediction target. For *distributional* models, we additionally sweep over $\lambda \in \{0, 0.1, 0.5, 1.0\}$, $\beta \in \{0.0001, 0.001, 0.01, 0.1, 0.5, 1.0, 1.5, 1.99, 2.0\}$. We use population size $m = 4$. We also tried $m = 2$ but it led to slightly worse results (the resulting FID was 7% worse). We did not use higher population sizes since it led to higher computational and memory cost (see Appendix J for more details). When we use kernel diffusion losses, see Section 2.2 and Section 3, for (imq) kernel (9a), we sweep over $c \in \{0.01, 0.1, 0.3, 0.5, 1.0, 1.5, 2.0, 10.0, 50.0, 100.0\}$. For for (rbf) kernel (9b) and for (exp) kernel (9c), we firstly compute median $M^2 = ||x - x'||_2^2$ for (rbf) kernel and $M = ||x - x'||_2$ for (exp) kernel, on a subset of 100 batches of size 512 from the dataset. Then, for (rbf) kernel, we define $\sigma^2 = M^2\gamma$ and we sweep over $\gamma \in \{0.01, 0.1, 0.3, 0.5, 1.0, 1.5, 2.0, 10.0\}$. For (exp) kernel, we define $\sigma = M\gamma$ and we sweep over $\gamma \in \{0.01, 0.1, 0.3, 0.5, 1.0, 1.5, 2.0, 10.0\}$. Since we only use kernel diffusion losses on CIFAR-10, we only report the value of the medians in this case. We have that $M^2 = 1357.8127$ and $M = 36.9$.

In order to produce samples, we follow Algorithm 2 for *distributional* models and SDE sampler for classical diffusion models. To evaluate performance, we use FID (Heusel et al., 2017) metric. As we train the models, every 10000 steps we produce samples with 10 denoising steps for *distributional* models and with 100 denoising steps for classical diffusion model. We then evaluate the performance of the models by computing FID on a subset of 2048 datapoints from the original dataset. Then, we use this metric in order to select the best hyperparameters and the best checkpoints for each best hyperparameter. After training is done, for every combination of $\lambda$, $\beta$ (or $c$, $\sigma$, $\sigma^2$ in case of other kernels), we select the best checkpoint based on their FID on this small subset of data. The corresponding checkpoints are then used for evaluation when sampling with $10, 30, 50, 70, 90, 100$ denoising steps. To compute the final FIDs, we use 50000 samples for CIFAR-10, LSUN Bedrooms and CelebA. We use 30000 samples for latent CelebA-HQ.

### I.3. Robotics experiments

We used the Libero benchmark (Liu et al., 2024), a life-long learning benchmark that consists of 130 language-conditioned robotic manipulation tasks. There are 5 suites in the Libero benchmark, Libero-Spatial, Libero-Object, Libero-Goal, Libero-Long and Libero-90. Each suite contains 10 tasks, except Libero-90 which has 90 tasks. In our experiments we focus on Libero-Long, the most challenging suite with 10 tasks that features long-horizon tasks with diverse object interactions, and reproduce the main experiment settings on three other suites: Libero-Spatial, Libero-Object, Libero-Goal, with 10 tasks. We train language-conditioned multitask diffusion policies for the 10 tasks of Libero10 suite and evaluate the success rates of the policies in simulation. We encode the visual observations using a ResNet (He et al., 2016), and encode the language instruction using a pretrained and frozen Bert (Devlin et al., 2018) encoder. Our multitask diffusion policy does action chunking (Chi et al., 2023), and predicts and executes a chunk of 8 7-dimensional actions conditioned on the language instruction, the current visual observations and proprioceptive states. For the denoiser, we follow Aloha unleashed (Zhao et al., 2024) and use a cross-attention based transformer denoiser.

We follow the original Libero paper for training and evaluating our multitask diffusion policy. As a loss function for training the diffusion policy, we use the Mean Squared Error (MSE). Libero10 suite has 138090 steps of training data and we use a

batch size of 32 to train our policies for 250k steps (roughly 50 epochs). Evaluation is on simulated environment for 50 episodes per task following the Libero benchmark. We evaluate our policy every 25k steps (roughly 5 epochs). We take the median success rate of the policy across all tasks in the suite and repeat the experiment over 3 seeds. We report the median success rates over all tasks and best checkpoint during training. We use the same set for other three suites and train for 50 epochs and evaluate every 5 epochs.

**Energy distance diffusion model.** For the energy distance diffusion variant, we concatenated 2 noise dimensions, sampled from $\mathcal{N}(0; \mathrm{Id})$ to the action dimension of the chunks. For training we used population size $m = 16$ per data sample, and computed the loss with varying $\beta$s and $\lambda$s. For evaluation, we use a single sample and execute the predicted action chunk from the diffusion policy using a varying number of diffusion steps $(2, 16, 50)$.

## J. Computational complexity

Our method has the same computational complexity during sampling as ordinary diffusion models. However, it has an increased computational complexity during training which is detailed below.

Assume that the computational complexity of the forward pass in our diffusion model is $O(F)$ and that the dimensionality of $x$ is $D$.

**Diffusion loss case.** The loss function in (14) with $m = 1$ and $\lambda = 0$, can be thought of as a standard diffusion model loss function. It requires $O(nF)$ evaluations to compute $x_\theta(t_i, X_{t_i})$ for every element of a batch. Then, to evaluate the loss (the norm), the complexity is $O(nD)$. Therefore the total cost of computing the loss is $O(n(F + D))$. The backwards pass is proportional to the forward pass and has computational complexity $O(nF)$.

**Distribution loss case.** For $\lambda \neq 0$ and $m > 1$, we first need $O(nmF)$ function evaluations to compute $x_\theta(t_i, X_{t_i}, \xi_{i,j})$ for $i = 1, \ldots, n$ and for $\xi_{i,j}$, $j = 1, \ldots, m$. Additionally, we need $O(nmD)$ operations to compute the first (diffusion-like) terms of the loss. Then, in order to compute the interaction terms, we need $O(nm(m - 1)D)$ time. Therefore, the total cost is of computing the loss is $O(nm(F + D) + nm(m - 1)D)$. Naively, the backwards pass will take $O(nmF + nm(m-1)F) = O(nm^2 F)$ time. However, since the 2nd term uses the gradients of $x_\theta(t_i, X_{t_i}, \xi_{i,j})$, one could precompute these for all $i = 1, \ldots, n$ and $j = 1, \ldots, m$ and therefore decrease the total backwards cost to $O(nmF)$. We found that in practice, the XLA compiler in JAX performs this optimization without explicit coding, see our results below.

**Runs on real hardware**. We compared the training times on real hardware. We study the training of CIFAR-10 as we vary $m$. We report steps per second metric where a step corresponds to a full forward+backward step. As hardware, we use A100 GPU (40Gb of memory) with batch size = 16, H100 GPU (80Gb of memory) with batch size = 64 and TPUv5p (95Gb of memory) with batch size = 64 (per device with 4 devices in total). The results below indicate that steps per second decreases proportionally to $m$.

| Hardware | Diffusion | Distributional (m=2) | Distributional (m=4) | Distributional (m=8) |
|---|---|---|---|---|
| A100 (batch size =16) | 9.05 | 6.78 | 4.6 | 2.77 |
| H100 (batch size =64) | 14.3 | 7.85 | 4.15 | 2.06 |
| TPUv5 (batch size =64) | 11.2 | 8.5 | 4.3 | 2.22 |

*Table 1.* Steps per seconds on real hardware for different models.

## K. Additional experiments

### K.1. Additional 2D experiments

We trained an unconditional standard diffusion model and our distributional models on a more complex 2D distribution – checkerboard, see Galashov et al. (2025). We report MMD between sampled and target distributions with different NFEs, smaller MMD is in bold, in Table 2 Our approach outperforms standard diffusion for small NFEs.

### K.2. Additional image experiments

In this section, we provide additional results for image space experiments.

| NFEs | Diffusion | Distributional |
|------|-----------|----------------|
| 5    | 7.00e-3   | **1.57e-4**    |
| 10   | 2.10e-3   | **1.96e-4**    |
| 50   | 2.70e-4   | **1.55e-4**    |
| 100  | **1.96e-4** | 2.00e-4      |
| 1000 | **1.71e-4** | 1.83e-4      |

*Table 2.* MMD between target and sampled distribution for checkerboard dataset.



*Figure 10.* Heatmap of FIDs for conditional image generation on CIFAR-10. $x$-axis corresponds to $\lambda$, $y$-axis corresponds to $\beta$, the color corresponds to FID going from minimal (blue) to maximal (red) values. Different columns denote number of diffusion steps. Black squares correspond to the cases where performance reaches FID higher than specified threshold.

**Impact of parameters $\beta$ and $\lambda$ on FID.** We report detailed performance of *distributional* models $\hat{x}_\theta$ based on chosen $\beta$ and $\lambda$ for different number of diffusion steps. We present results for conditional image generation on CIFAR-10 in Figure 10 and on CelebA in Figure 11. The unconditional image generation for LSUN Bedrooms are given in Figure 12 and for latent CelebA-HQ in Figure 13. In practice, we noticed that performance of models for some $\lambda$ and $\beta$ led to very high FIDs. In order to better visually inspect the results, we mask the corresponding blocks if FID reaches a certain threshold. We used following thresholds: 9.8 for CIFAR-10, 20 for CelebA and LSUN Bedrooms, 300 for latent CelebA HQ. The results on CIFAR-10 and CelebA suggest that for a small number of steps, the best values of parameters are located in the bottom right corner, i.e. $\lambda$ close to 1.0 and $\beta$ small. As we increase the number of steps, better values appear in the top left corner, i.e., $\lambda$ close or equal to 0 and $\beta \approx 2$. In case of unconditional generation, we noticed generally that using the minimal possible $\beta$ led to the best results, as can be seen from Figure 12 and Figure 13, where the bottom row leads to the best performance.

**Detailed results with different kernels.** We present results on conditional image generation on CIFAR-10 with different kernels. First, in Figure 14, we show the value of the $\lambda$ for every kernel which achieves the lowest FID for the given number of diffusion steps. Overall, we observe that $\lambda$ follows a downward trend as we increase the number of diffusion steps, which is consistent with out experiments in the main paper. Furthermore, we show the corresponding value of the kernel parameter which achieves the lowest FID for given number of diffusion steps. Below, we also show the detailed figures of FID for every $\lambda$ and kernel parameter combination.

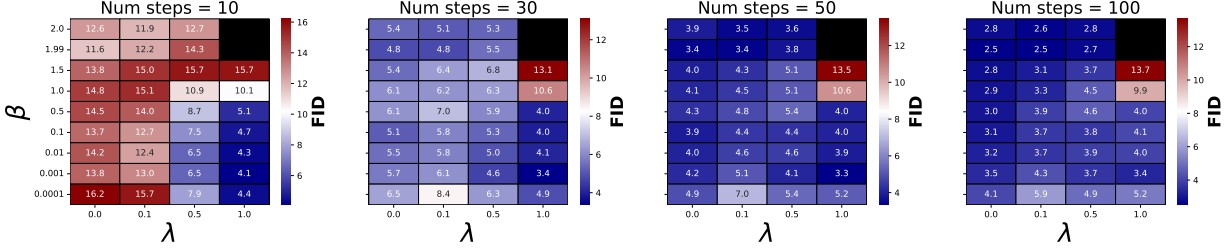

*Figure 11.* Heatmap of FIDs for conditional image generation on CelebA. $x$-axis corresponds to $\lambda$, $y$-axis corresponds to $\beta$, the color corresponds to FID going from minimal (blue) to maximal (red) values. Different columns denote number of diffusion steps. Black squares correspond to the cases where performance reaches FID higher than specified threshold.

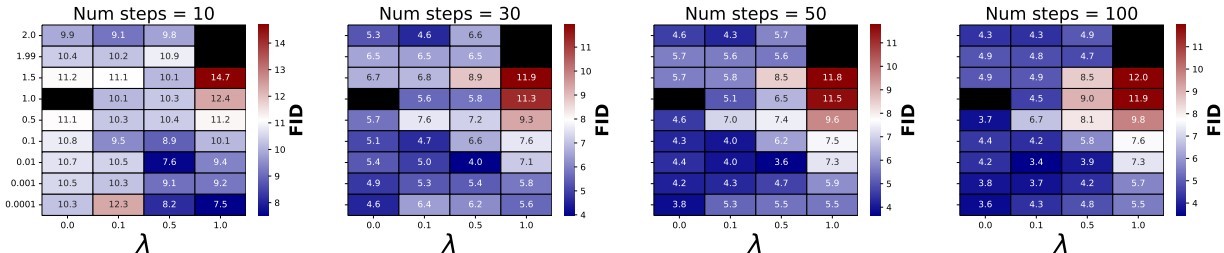

*Figure 12.* Heatmap of FIDs for unconditional image generation on LSUN Bedrooms. $x$-axis corresponds to $\lambda$, $y$-axis corresponds to $\beta$, the color corresponds to FID going from minimal (blue) to maximal (red) values. Different columns denote number of diffusion steps. Black squares correspond to the cases where performance reaches FID higher than specified threshold.

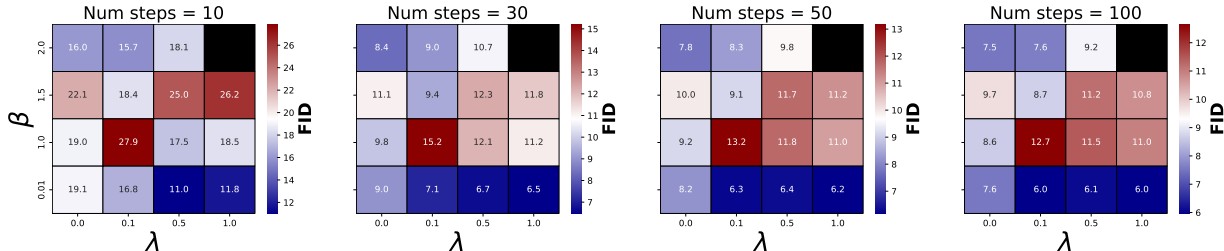

*Figure 13.* Heatmap of FIDs for unconditional image generation on Latent CelebA-HQ. $x$-axis corresponds to $\lambda$, $y$-axis corresponds to $\beta$, the color corresponds to FID going from minimal (blue) to maximal (red) values. Different columns denote number of diffusion steps. Black squares correspond to the cases where performance reaches FID higher than specified threshold.

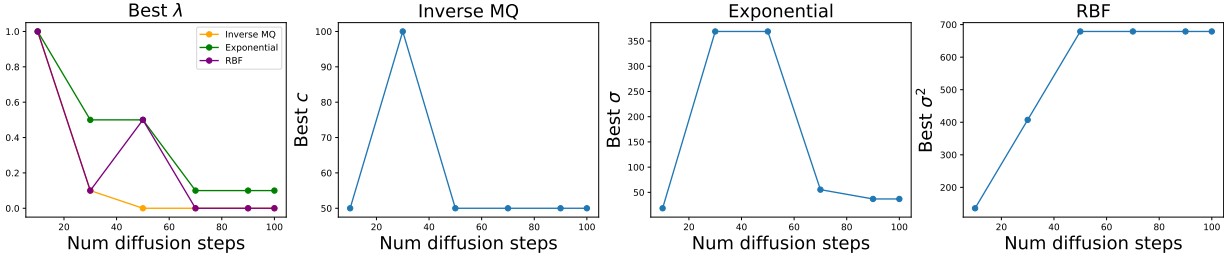

*Figure 14.* Best parameters $\lambda$ and corresponding kernel parameters for each kernel, achieving the lowest FID for the given number of diffusion steps.

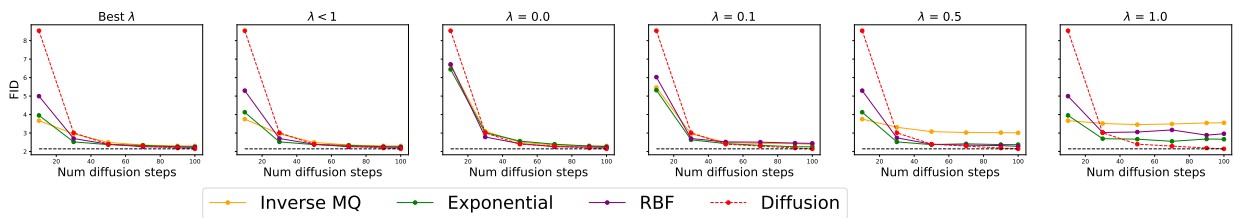

*Figure 15.* Performance of different kernels on conditional image generation on CIFAR-10, for different values of $\lambda$ (different columns). The kernel parameters are chosen to minimize FID for the given $\lambda$ condition. We also report performance of the diffusion model (red) and show in the black dashed line the performance of diffusion model with 100 steps.

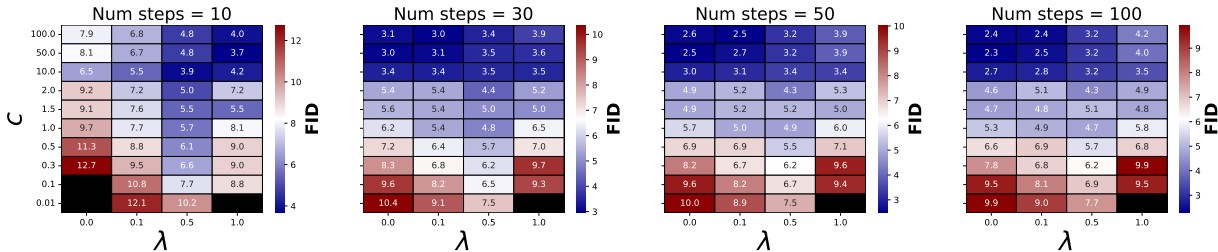

*Figure 16.* Heatmap of FIDs for conditional image generation on CIFAR-10 for Inverse MQ (9a) kernel. $x$-axis corresponds to $\lambda$, $y$-axis corresponds to $c$, the color corresponds to FID going from minimal (blue) to maximal (red) values. Different columns denote number of diffusion steps. Black squares correspond to the cases where performance reaches FID higher than specified threshold.

On top of that, we compare performance of different kernels for a fixed value of $\lambda$. We also report the performance of *energy* kernel, which is defined by $\rho(x, x') = \|x - x'\|^\beta$, see also Section 2.2. The results are given in Figure 15. We see that overall using different (to *energy*) kernels leads to similar results overall.

Finally, we present detailed results of FIDs for different kernels with different parameters $\lambda$ and corresponding kernel parameters. We report the corresponding heatmaps. Whenever the value of FID is larger than a threshold, we put black rectangles. For Inverse MQ, we use threshold 13.3. For RBF kernel, we use threshold 10. For Exponential kernel, we use threshold 10. For Inverse MQ, the results are given in Figure 16. We observe that overall the best FIDs are achieved for larger values of parameter $c$ for all three kernels we investigate. Moreover, when number of diffusion steps increases, the region with the best FIDs moves from $\lambda \approx 0$ to $\lambda \approx 1$. The results for RBF kernel are given in Figure 17. We observe that in case of small number of diffusion steps, the region of the best values is around $\lambda = 1$ and smaller values of $\sigma^2$. As we increase the number of diffusion steps, we generally observe that many values of $\lambda$ and $\sigma^2$ yield similar reasonable results. Results for Exponential kernel are given in Figure 18. When the number of steps is low, the best values are achieved for $\lambda$ close to 1. As we increase the value of $\lambda$, the best region moves closer to smaller values of $\lambda$. This is coherent with our theoretical and methodological motivation, i.e., $\lambda \approx 1$ is beneficial in the low number of diffusion steps regime, and confirms our experiments when using the energy kernel $\rho(x, x') = -\|x - x'\|^\beta$, see Figure 10.

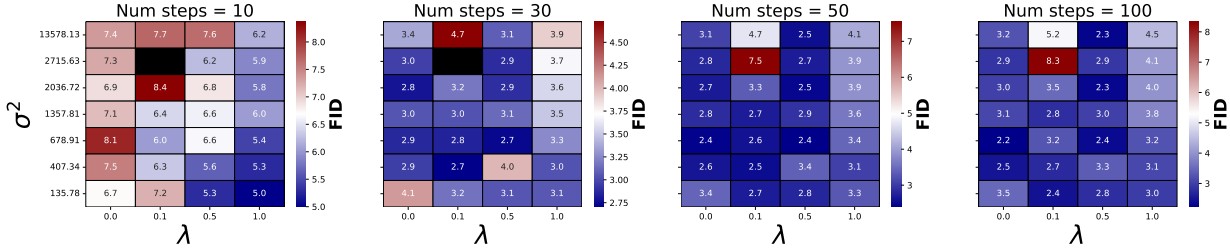

*Figure 17.* Heatmap of FIDs for conditional image generation on CIFAR-10 for RBF (9b) kernel. $x$-axis corresponds to $\lambda$, $y$-axis corresponds to $\sigma^2$, the color corresponds to FID going from minimal (blue) to maximal (red) values. Different columns denote number of diffusion steps. Black squares correspond to the cases where performance reaches FID higher than specified threshold.

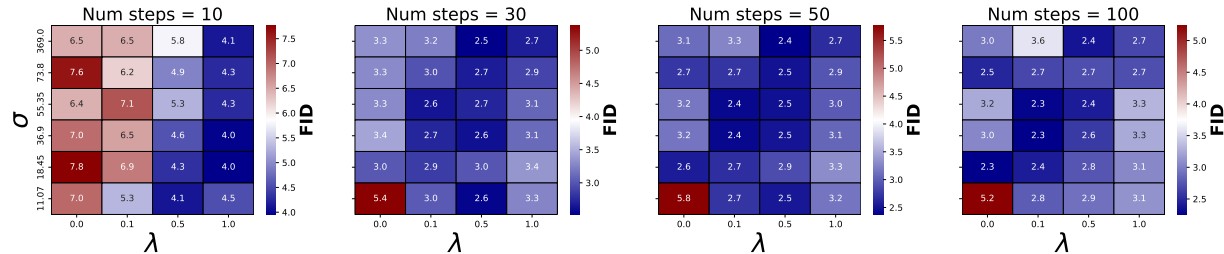

*Figure 18.* Heatmap of FIDs for conditional image generation on CIFAR-10 for Exponential (9c) kernel. $x$-axis corresponds to $\lambda$, $y$-axis corresponds to $\sigma$, the color corresponds to FID going from minimal (blue) to maximal (red) values. Different columns denote number of diffusion steps. Black squares correspond to the cases where performance reaches FID higher than specified threshold.

**ImageNet.**    We ran initial experiments on ImageNet (Russakovsky et al., 2015) with resolution $64 \times 64 \times 3$, but so far were unable to obtain satisfactory results. We will focus on scaling our method up to large datasets in the follow-up work.

### K.3. Comparisons to distillation and numerical solvers

In this section, we compare our approach with the *DPM-solver++* method Lu et al. (2022) and the state-of-the-art, multi-step distillation method, moment-matching distillation (Salimans et al., 2024).

For all the sampling methods, we sweep over the "safety" parameter $\eta$, which specifies the time interval to be $[\eta, 1-\eta]$. We considered values $\{0.1, 0.01, 0.001, 0.0001\}$. Moreover, for all the methods we also sweep over the "churn" parameter $\varepsilon$ which controls the stochasticity, see Equation (4), we tried $\{0.0, 0.25, 0.5, 0.75, 1.0\}$. We found the stochastic version of *DPM-solver++* (with a churn parameter close to 1) overall worked the best. For ordinary diffusion and our method, we use a "uniform" time schedule. For *DPM-solver++* we use the "logsnr" time schedule (see Lu et al. (2022)), which led to the best results (we also tried EDM and uniform).

For our distillation comparisons, we distill pre-trained diffusion models into student models. These student models are trained for 100,000 iterations, using a batch size of 256 for pixel-space models and 32 for latent-space models. During training, we sweep over the learning rate and Exponential Moving Average (EMA) decay.

The student model utilizes a stochastic DDIM sampler with a churn parameter, $\varepsilon$. We sweep over $\varepsilon$ values in the set $\{0.0, 0.5, 1.0\}$. We found that at sampling time, setting the churn parameter to $\varepsilon = 0$ for the distilled student yielded the best results.

For all the methods, we present the results with the best hyperparameters in the tables below (Table 3,Table 4,Table 5,Table 6).

*Table 3.* **CIFAR-10 (Conditional) results**. The metric is FID averaged across 3 random seeds. We report results with different NFEs for baseline diffusion model, DPM++, Distributional (ours) and moment matching distillation. In bold, we highlight the method with the lowest FID and in italic, we highlight the method with the second lowest FID.

| NFEs | Diffusion | DPM++ | Distributional | Distillation |
|---|---|---|---|---|
| 2 | 80.77 | 19.41 | 29.33 | **5.19** |
| 4 | 23.31 | 12.60 | *4.67* | **3.84** |
| 8 | 7.53 | *3.95* | 3.21 | **3.13** |
| 10 | 5.67 | *3.55* | 3.19 | **2.99** |
| 12 | 4.83 | *3.69* | 3.03 | **2.93** |
| 15 | 3.61 | *3.91* | 2.87 | **2.76** |

The results indicate that our approach is competitive with multistep distillation (across a varied number of steps) and outperforms *DPM-Solver++* at 8 or more Number of Function Evaluations (NFEs). The performance of *DPM-Solver++* degrades as the number of steps increases, which is an expected consequence of its numerical instability.

Our approach has two compelling advantages over multistep distillation. First, it does not rely on distillation, eliminating the need to train a large, slow teacher model. Second, it does not require specifying additional sampler hyperparameters during training.

*Table 4.* **CelebA (Conditional) results**. The metric is FID averaged across 3 random seeds. We report results with different NFEs for baseline diffusion model, DPM++, Distributional (ours) and moment matching distillation. In bold, we highlight the method with the lowest FID and in italic, we highlight the method with the second lowest FID.

| NFEs | Diffusion | DPM++ | Distributional | Distillation |
|------|-----------|-------|----------------|--------------|
| 2    | 62.85     | *58.39* | 81.79        | **7.02**     |
| 4    | 26.25     | *20.39* | 22.98        | **6.47**     |
| 8    | 13.07     | 17.28 | *4.94*         | **4.43**     |
| 10   | 11.67     | 22.48 | **3.60**       | *4.43*       |
| 12   | 10.00     | 23.23 | **3.37**       | *4.25*       |
| 15   | 8.51      | 23.53 | **3.45**       | *4.27*       |

*Table 5.* **LSUN (Unconditional) results**. The metric is FID averaged across 3 random seeds. We report results with different NFEs for baseline diffusion model, DPM++, Distributional (ours) and moment matching distillation. In bold, we highlight the method with the lowest FID and in italic, we highlight the method with the second lowest FID.

| NFEs | Diffusion | DPM++ | Distributional | Distillation |
|------|-----------|-------|----------------|--------------|
| 2    | 238.85    | *120.99* | 233.24      | **16.04**    |
| 4    | *41.57*   | 43.60 | 69.90          | **5.54**     |
| 8    | 10.82     | 12.95 | *9.79*         | **4.52**     |
| 10   | 9.08      | 17.85 | *7.14*         | **4.64**     |
| 12   | 7.04      | 16.07 | *6.12*         | **3.81**     |
| 15   | 6.51      | 23.91 | *5.78*         | **4.39**     |

*Table 6.* **Latent CelebA-HQ (Unconditional) results**. The metric is FID averaged across 3 random seeds. We report results with different NFEs for baseline diffusion model, DPM++, Distributional (ours) and moment matching distillation. In bold, we highlight the method with the lowest FID and in italic, we highlight the method with the second lowest FID.

| NFEs | Diffusion | DPM++ | Distributional | Distillation |
|------|-----------|-------|----------------|--------------|
| 2    | 101.47    | 114.44 | *67.60*       | **37.88**    |
| 4    | 53.83     | 26.05 | *23.80*        | **14.47**    |
| 8    | 15.60     | 20.70 | *9.84*         | **8.22**     |
| 10   | 11.89     | 21.84 | *8.15*         | **6.76**     |
| 12   | 9.97      | 24.61 | *7.04*         | **6.05**     |
| 15   | 8.39      | 27.31 | *6.21*         | **5.46**     |

Furthermore, combining our distributional approach with a modern distillation method is a potentially interesting topic for future research.

### K.4. Additional robotics experiments

In addition to Libero10 suite, we repeated our main experiment on three other suites in Libero benchmark: Liber-Goal, Libero-Spatial, and Libero-Object. We present the performance on these three additional suites in Figure 19. In all three suites, distributional diffusion helps the performance when fewer number of diffusion steps are used, reproducing our results on Libero10.

In the setting of Section 6.3, we only consider the case $\beta = 1$ and compared our results for several values of $\lambda \in [0, 1]$, namely $\lambda \in \{0.0, 0.1, 0.5\}$. In Figure 20, we present a similar study as Figure 9 but in the case where $\beta = 2$.

We also ran an ablation over $m$ with fixed $\beta = 1$ and $\lambda = 0.5$. The results are reported in Table 7.

### K.5. Additional samples

For CIFAR-10, samples from the *distributional* model $\hat{x}_\theta$ trained with $\beta = 0.1, \lambda = 1$ and with 10 sampling steps are shown in Figure 21, left. Samples from the one trained with $\beta = 2, \lambda = 0$ and with 100 sampling steps are shown in Figure 22, left. The samples from the diffusion model with 10 steps are shown in Figure 21, right and with 100 steps are shown in

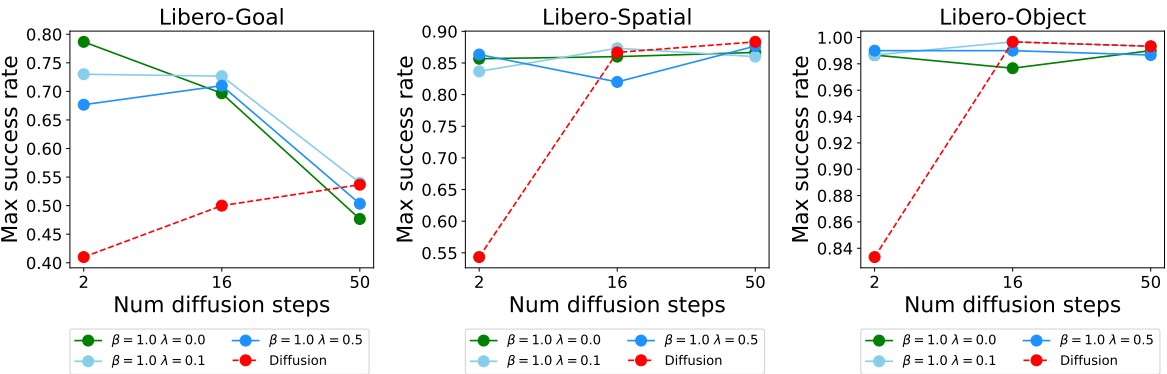

*Figure 19.* Performance in three other Libero suites - Libero-Goal, Libero-Spatial, and Libero-Object - as function of diffusion steps and $\lambda$ with same settings as Figure 9. We plot the maximum success rate from best checkpoint during training. Distributional models consistently performed the best when fewest diffusion steps are used. $\lambda > 0$ also performs better than $\lambda = 0$

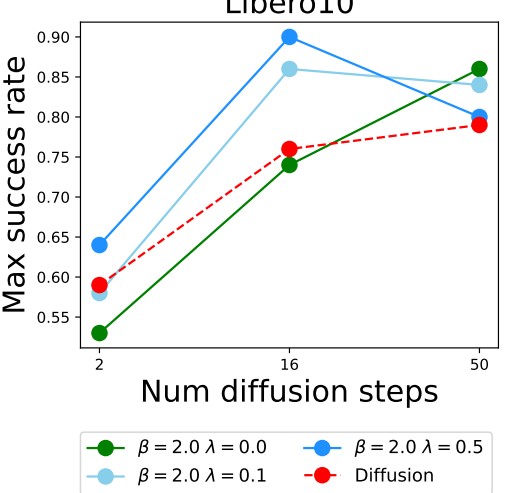

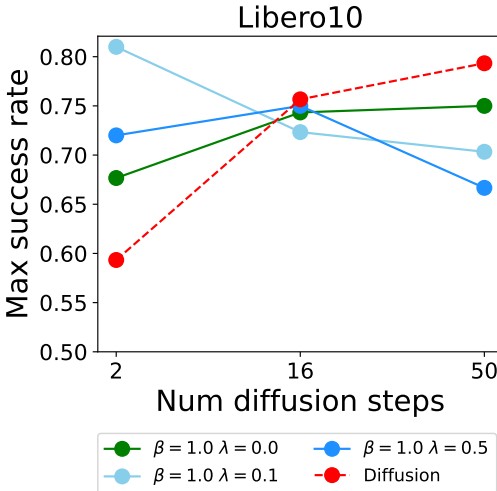

*Figure 20.* **Left**. Performance in Libero10 as function of diffusion steps and $\lambda$, same as Figure 9 but for $\beta = 2$. **Right**. we reproduce Figure 9 for ease of read.

| $m$ | Success rate |
|----|--------------|
| 2  | 0.783 |
| 4  | 0.73 |
| 8  | 0.817 |
| 16 | 0.800 |
| 32 | 0.840 |
| 64 | 0.797 |

*Table 7.* Ablation over $m$ for robotics dataset.

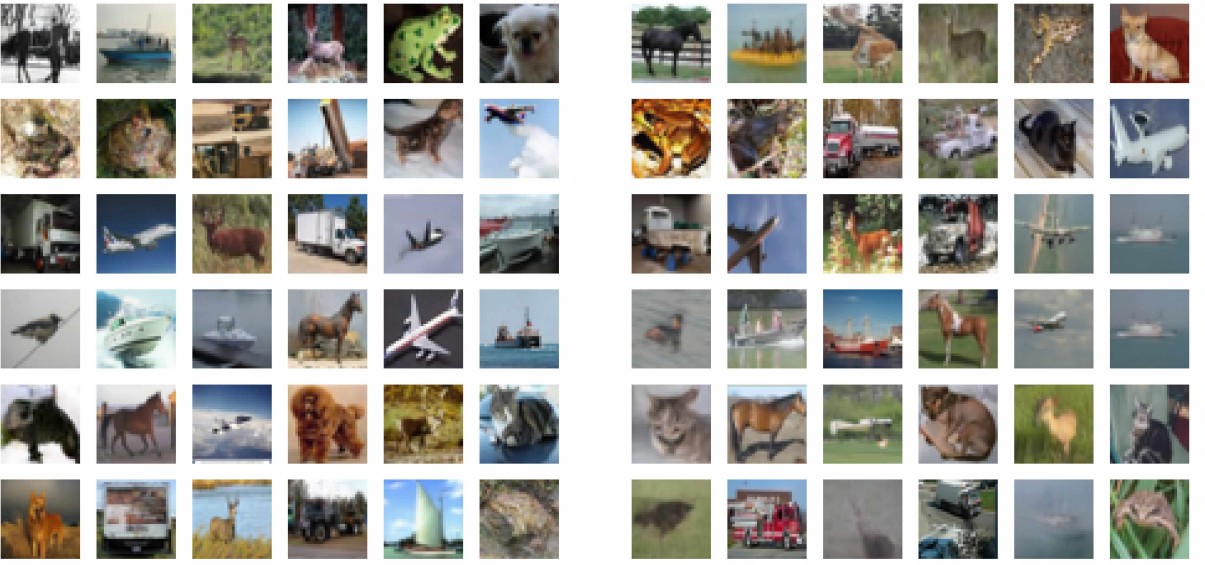

*Figure 21.* **Left**, samples from *distributional* model trained on CIFAR-10 and sampled with 10 steps. $\beta = 0.1, \lambda = 1$. **Right**, samples from diffusion model trained on CIFAR-10 and sampled with 10 steps.

Figure 22, right.

For CelebA, samples from the *distributional* model trained with $\beta = 0.001, \lambda = 1$ and with 10 sampling steps, are shown in Figure 23, left. Samples from the one trained with $\beta = 2, \lambda = 0$ and with 100 sampling steps, are shown in Figure 24, left. Samples from diffusion model with 10 steps are shown in Figure 23, right and with 100 steps are shown in Figure 24, right.

For LSUN Bedrooms, samples from the *distributional* model trained with $\beta = 0.0001, \lambda = 1$ and with 10 sampling steps, are shown in Figure 25, left. Samples from the one trained with $\beta = 0.0001, \lambda = 0$ and with 100 sampling steps, are shown in Figure 26, left. Samples from the diffusion model with 10 steps are shown in Figure 25, right and with 100 steps are shown in Figure 26, right.

For latent CelebA HQ, samples from the *distributional* model trained with $\beta = 0.01, \lambda = 0.5$ and with 10 sampling steps, are shown in Figure 27, left. The samples from the one trained with $\beta = 0.01, \lambda = 0.1$ and with 100 sampling steps, are shown in Figure 28, left. Samples from the diffusion model with 10 steps are shown in Figure 27, right and with 100 steps are shown in Figure 28, right.

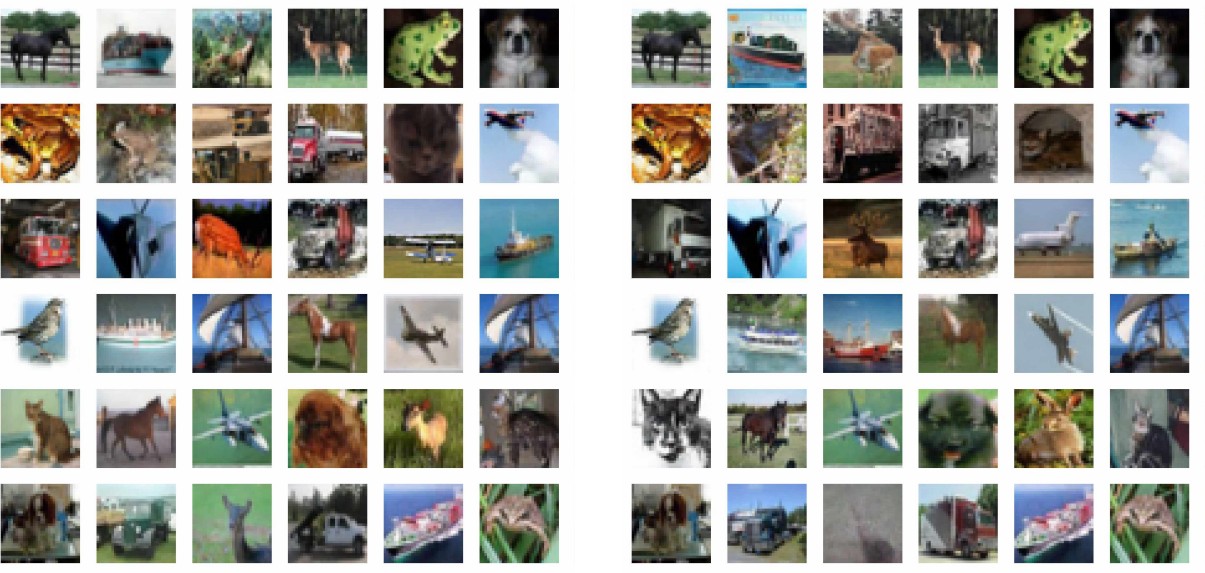

Figure 22. **Left**, samples from *distributional* model trained on CIFAR-10 and sampled with 100 steps. $\beta = 2, \lambda = 0$. **Right**, samples from diffusion model trained on CIFAR-10 and sampled with 100 steps.

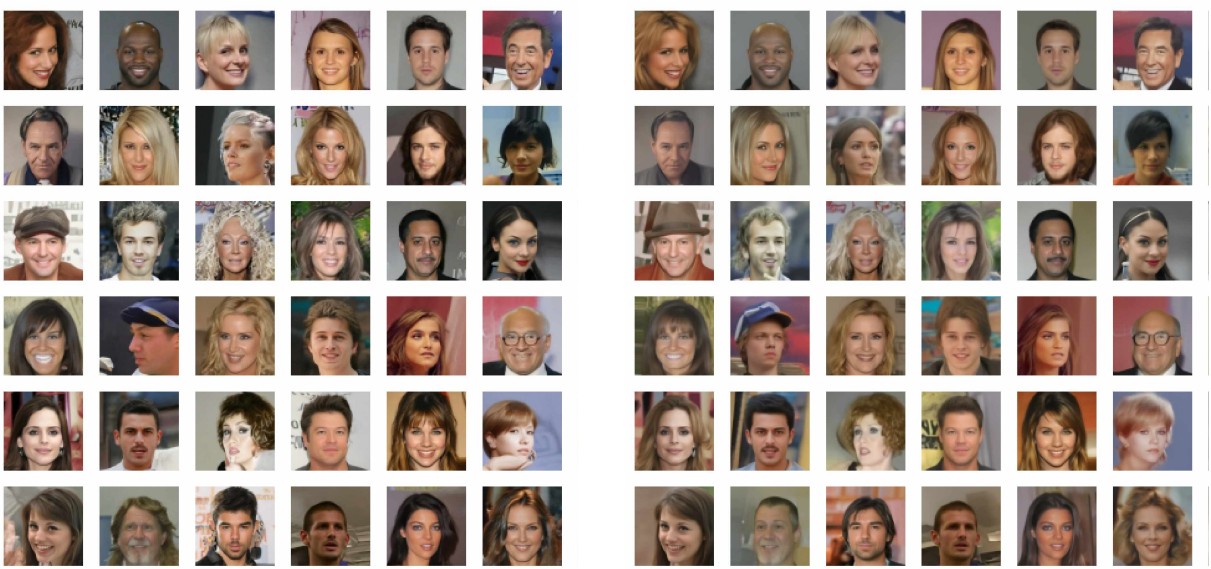

Figure 23. **Left**, samples from *distributional* model trained on CelebA and sampled with 10 steps. $\beta = 0.001, \lambda = 1$. **Right**, samples from diffusion model trained on CelebA and sampled with 10 steps.

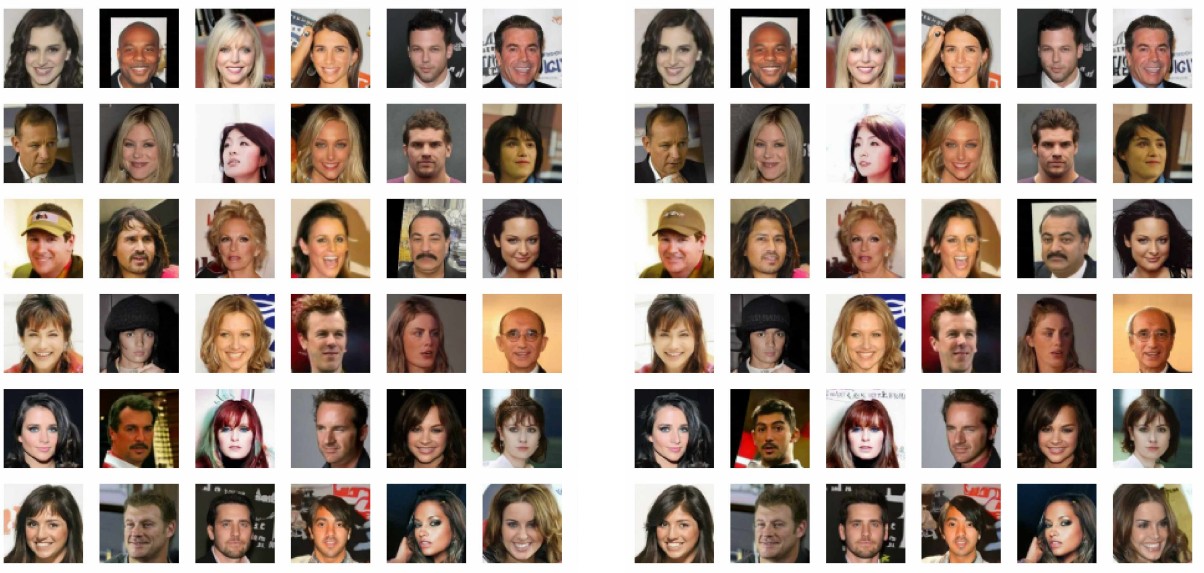

*Figure 24.* **Left**, samples from *distributional* model trained on CelebA and sampled with 100 steps. $\beta = 2, \lambda = 1$. **Right**, samples from diffusion model trained on CelebA and sampled with 100 steps.

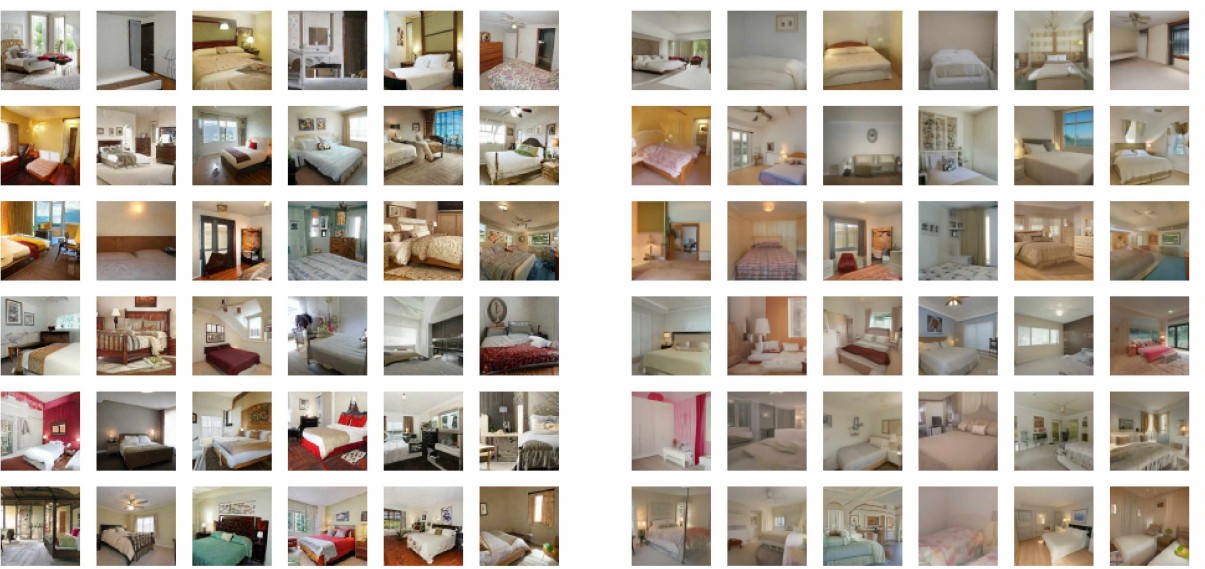

*Figure 25.* **Left**, samples from *distributional* model trained on LSUN Bedrooms and sampled with 10 steps. $\beta = 0.0001, \lambda = 1$. **Right**, samples from diffusion model trained on LSUN Bedrooms and sampled with 10 steps

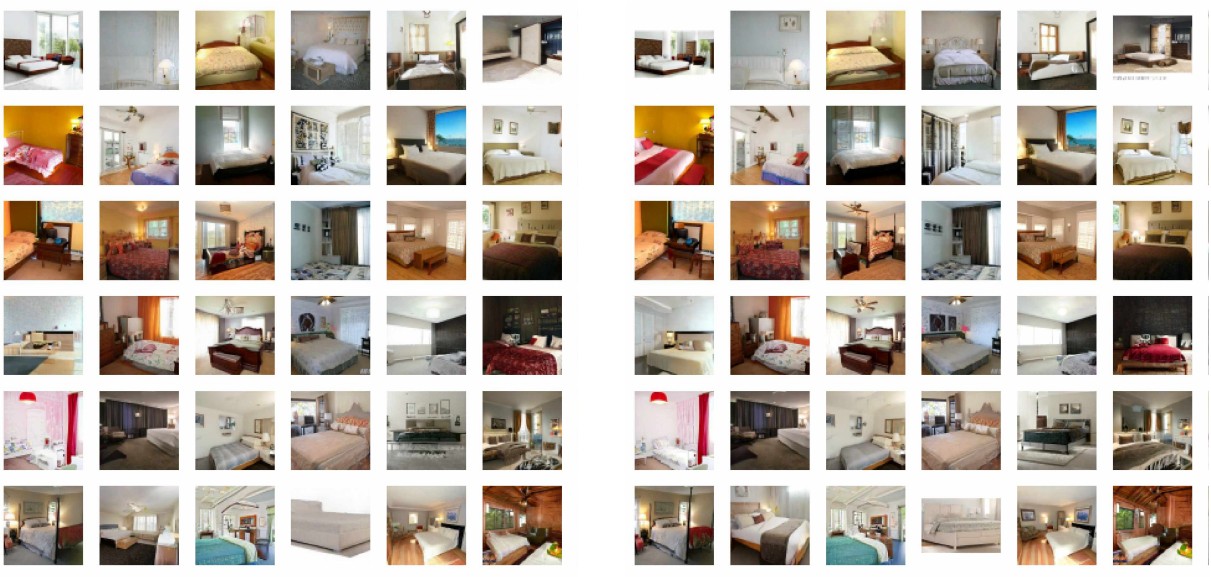

*Figure 26.* **Left**, samples from *distributional* model trained on LSUN Bedrooms and sampled with 100 steps. $\beta = 0.0001, \lambda = 0$. **Right**, samples from diffusion model trained on LSUN Bedrooms and sampled with 100 steps

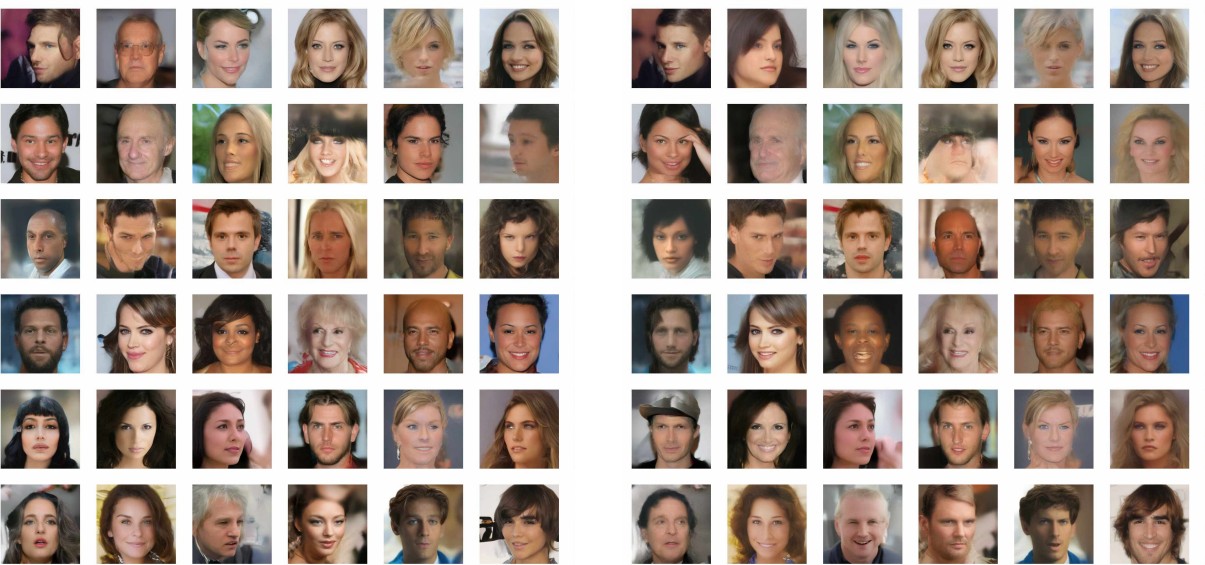

*Figure 27.* **Left**, samples from *distributional* model trained on latent CelebA HQ and sampled with 10 steps. $\beta = 0.01, \lambda = 0.5$. **Right**, samples from diffusion model trained on latent CelebA HQ and sampled with 10 steps.

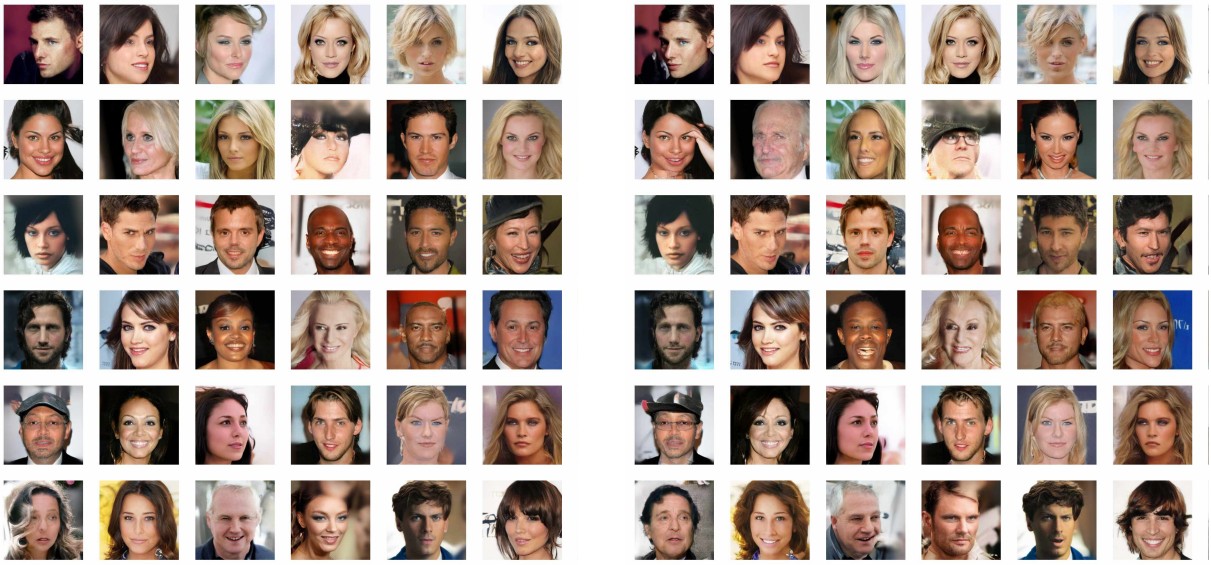

*Figure 28.* **Left**, samples from *distributional* model trained on latent CelebA HQ and sampled with 100 steps. $\beta = 0.01, \lambda = 0.1$. **Right**, samples from diffusion model trained on latent CelebA HQ and sampled with 100 steps.

