# OpenReview forum: "Distributional Diffusion Models with Scoring Rules"
_ICML.cc/2025/Conference — ICML 2025 poster_

### Official Review · Reviewer_FCZ5 · 2025-03-03

**Overall Recommendation:** 3

**Summary:**

This paper introduces a approach to accelerate diffusion models by learning the full conditional posterior distribution of clean data given noisy samples, rather than just the conditional mean, as done in standard diffusion models. The authors propose replacing the traditional regression loss (used to estimate conditional means) with a generalized energy score based on scoring rules, which allows sampling from the posterior distribution on a coarser time scale, thus reducing the number of denoising steps required.

**Claims And Evidence:**

The author claim that $p_{0|t}(x_0|x_t)$ is approximated by $\delta_{\hat{x}_\theta}(t,x_t)$, which is not true. In ddpm, $p\_{0|t}(x_0|x_t)=\mathcal{N}(x_0;\tilde{\mu}(x_t), \tilde{\beta}_t)$, where $\tilde{\mu}(x_t)=\frac{1}{\sqrt{\bar{\alpha}_t}}(x_t-\sqrt{1-\bar{\alpha_t}}\epsilon_t)$ and $\tilde{\beta}_t=\frac{(1-\bar{\alpha}_t)}{\bar{\alpha}_t}I$, $\epsilon_t = \mathbb{E}[\epsilon]$ denotes noise added in the forward process. Hence, it's strange to claim that  posterior distribution is  Dirac. I think the author should explain this thing, maybe i am wrong, but author should give the reason.

**Essential References Not Discussed:**

[1] Galashov A, de Bortoli V, Gretton A. Deep MMD gradient flow without adversarial training[J]. arXiv preprint arXiv:2405.06780, 2024.

[2] Aiello E, Valsesia D, Magli E. Fast inference in denoising diffusion models via mmd finetuning[J]. IEEE Access, 2024.

**Experimental Designs Or Analyses:**

The experimental designs are sound and validate the core hypothesis of improved performance with fewer steps. However, the reliance on a single metric (FID for images, success rate for robotics) limits the depth of the evaluation. Including diversity metrics and more complex 2D distributions would strengthen the analysis.

**Methods And Evaluation Criteria:**

The proposed method replaces the regression loss in standard diffusion models with a generalized energy score (Equation 12) or kernel score, training a generative network to approximate the full posterior. The evaluation criteria include:

 - 2D Experiments: The authors use a mixture of Gaussians as the target distribution and measure performance via squared Maximum Mean Discrepancy (MMD) with an RBF kernel.
 - Image Generation: Evaluated on CIFAR-10, CelebA, LSUN Bedrooms, and CelebA-HQ (latent space) using the FID.
 - Robotics: Evaluated on the Libero-Long benchmark, measuring success rates for multi-task robotic manipulation.

**Other Comments Or Suggestions:**

Additional Experiments: Including experiments on larger model (e.g., sdxl, sd3.5) or more diverse robotics tasks would strengthen the empirical validation.

**Other Strengths And Weaknesses:**

Strengths:

- The use of scoring rules to learn the full posterior in diffusion models is a novel and elegant solution to the problem of slow sampling, avoiding the complexity of GANs or distillation.

 - The method’s ability to reduce the number of denoising steps while maintaining quality has practical implications for real-time applications (e.g., robotics).

 - The paper is well-written, with a clear structure and detailed explanations of the method and experiments.

Weaknesses:

 - The author do not discuss the relationship between this paper and [1, 2].

 - The author claim that $p_{0|t}(x_0|x_t)$ is approximated by $\delta_{\hat{x}_\theta}(t,x_t)$ may be wrong.

 - The proposed method is similar to Maximum Mean Discrepancy (MMD), but use a different story.

[1] Galashov A, de Bortoli V, Gretton A. Deep MMD gradient flow without adversarial training[J]. arXiv preprint arXiv:2405.06780, 2024.

[2] Aiello E, Valsesia D, Magli E. Fast inference in denoising diffusion models via mmd finetuning[J]. IEEE Access, 2024.
- The experiments focus heavily on FID and success rates, neglecting metrics like sample diversity or robustness, which are crucial for generative models.

**Questions For Authors:**

- In the empirical energy diffusion loss (Equation 14), you use m=8 (2D experiments) and m=16 (robotics). How sensitive is the performance to the choice of m, and what motivated these specific values? A larger m might better approximate the posterior but increase computational cost—how does this trade-off affect your method?

 - Proposition 4.3 shows that IMQ and RBF kernels are diffusion-compatible, but the exponential kernel (kexp) is not. However, you note that kexp still results in desirable properties (Section 4.3). Can you elaborate on these properties and why non-diffusion-compatible kernels might still be useful? This could clarify the practical implications of kernel choice.

 - Perhaps the author needs to explain that if $\Sigma_{s,t} = \sigma_s^2 \left( 1 - \left( \varepsilon^2 r_{1,1}(s,t) + (1 - \varepsilon^2) \right)^2 \right) \text{Id},$ is defined as such, then why is learning still necessary? Wouldn't it be possible to directly use this $\Sigma_{s,t}$? (In the algorithm, the author also directly uses this $\Sigma_{s,t}$, with only the mean being modified.)
 If, as I mentioned earlier, the transition probability distribution no longer follows a Gaussian distribution due to cross-step state superposition, then what should $\Sigma_{s,t}$ look like? Is the learned result from the method proposed by the author correct?

**Relation To Broader Scientific Literature:**

They leveraged the idea of diffusion distillation to fit the posterior distribution across different step sizes. Due to the accumulation of multiple states, an excessively large step size can lead to significant approximation errors in the posterior estimation, causing it to deviate from a Gaussian distribution. In this case, the predicted \(\epsilon\) can no longer serve as the score, and the variance becomes uncertain in its evolution, requiring a neural network for estimation.

**Theoretical Claims:**

The paper makes several theoretical claims, primarily in Section 4. The theoretical claims are generally sound for the specific cases analyzed (Gaussian distributions, specific kernels)

---

> ### Author Rebuttal · Authors · 2025-03-31
>
> We thank the reviewer FCZ5 for their feedback on our work. Please see our response below
>
> |$p_{0|t}(x_0|x_t)$,Dirac…not true
>
> Using $p_{0|t}(x_0|x_t)\approx\delta_{\hat{x}_\theta(t,x_t)}$ in $p(x_s|x_t)$ (see eq.5) leads to
>
> $p(x_s|x_t)\approx p(x_s|x_0=\hat{x}_\theta(t,x_t),x_t),$
>
> where $p(x_s|x_0,x_t)$ is Gaussian (see eq.4). Sampling from these approximations of $p(x_{t_k}|x_{t_{k+1}})$ for some $t_N=1>t_{N-1}>....>t_1>t_0=0$ gives exactly the DDIM algorithm (see DDIM paper [1],eq.10)
>
> |Other metrics,diversity
>
> FID [2] already quantifies the diversity of the sampled images as it is Wasserstein-2 distance between Gaussian approximations to the distributions in the space of Inception V3 coding layer features, and it takes into account the covariances of these distributions. See [3] regarding image metrics for evaluation (i.e., “[the FID score] unlike IS [..] is able to detect intra-class mode dropping”)
>
> A good metric for evaluating diffusion policy in robotics is an open challenge and so far success rates during independent evaluation is the only measure that has proven useful for robotics. Systematic evaluations of various metrics for policies yielded no reliable metrics [4]
>
> |more 2D
>
> We trained unconditional standard diffusion and our distributional models on a more complex 2D distribution – checkerboard, see [5]. We report MMD between sampled and target distributions with different NFEs, smaller MMD is in **bold**. Our approach outperforms standard diffusion for small NFEs. We will add these results to the paper
>
> NFEs|Diffusion|Distributional
> -|-|-
> 5|7.00e-3|**1.57e-4**
> 10|2.10e-3|**1.96e-4**
> 50|2.70e-4|**1.55e-4**
> 100|**1.96e-4**|2.00e-4
> 1000|**1.71e-4**|1.83e-4
>
> |[5-6]
>
> Thank you for pointing out these references. We’ll discuss as follows:
>
> “Scoring rules have corresponding MMD divergences, and approaches exist that use the MMD in diffusion and particle flow models. [5] generates a sequence of distributions from a forward diffusion process. It then generates noise-dependent neural MMDs between clean and noisy data, and performs gradient flow to move particles from noise level $t$ to a lower noise level $s<t$. Compared to ours, [5] does not use a generator since it is a particle flow. Their best CIFAR10 FID is 7.7. [6] first trains a diffusion model, and then refines/distills by coarsening the reverse timesteps, and uses MMD on CLIP features to finetune the DDM denoiser. Their CIFAR10 FID for NFE=10 improves from 13.6 to 3.8, while ours is 3.19.
>
> |m sensitivity
>
> Please see our response to reviewer r8Za about m and computational complexity. For image experiments $m=2,4$ already worked well. We will add an ablation over $m$ for CIFAR10 in the revised version of the paper. For robotics, we ran an ablation over $m$ ($\beta=1,\lambda=0.5$), see results below
>
> m|Success rate
> -|-
> 2|0.783
> 4|0.73
> 8|0.817
> 16|0.800
> 32|0.840
> 64|0.797
>
> |IMQ,RBF,kexp
>
> Proper scoring rules define divergences. For kernel scores, this is the squared MMD. The energy distance kernel is appealing – its loss (13) is similar to standard diffusion loss, and other kernels can recover this loss in certain limits. While this recovery is good for interpretability, any characteristic kernel (including the $kexp$) gives a valid scoring rule. A benefit of (13) is its numerical stability without exponentiation
>
> |additional experiments
>
> Please see our response to reviewer waLv for comparisons with DPM-solver++ and SOTA multi-step distillation.
>
> We started an experiment on ImageNet (64x64), but due to time constraints, will only add results after rebuttal. As for robotics, we studied 10 most challenging tasks from Libero benchmark, which is already a diverse set of language-conditioned robotics tasks. We started an experiment on all tasks from Libero and will report results after rebuttal
>
> |$\Sigma_{s,t}$...why learning necessary?
>
> We need to approximately sample from $p(x_s|x_t)=\int p(x_s|x_0,x_t)p(x_0|x_t)dx_0$ where $p(x_s|x_0,x_t)=N(x_s;\mu_{s,t}(x_0),\Sigma_{s,t})$ is fixed. We obtain a better approximation by learning $p(x_0|x_t)$ with scoring rules instead of Dirac mass as denoiser. Minimizing loss (13) recovers the true $p(x_0|x_t)$, if $p^\theta(x_0|x_t)$ is rich enough. The resulting $p(x_s|x_t)$ is indeed non-Gaussian and its form is fully problem dependent.
>
> We hope that our explanations and updated experiments have addressed your concerns. If so, we would be grateful if you would consider raising your score.
>
> [1]Song J et al.,Denoising Diffusion Implicit Models,ICLR2021
>
> [2]Heusel M et al.,GANs Trained by a Two Time-Scale Update Rule Converge to a Local Nash Equilibrium,NeurIPS2017
>
> [3]Borji A,Pros and Cons of GAN Evaluation Measures,2018
>
> [4]Hussenot L, et al.,Hyperparameter Selection for Imitation Learning,2021
>
> [5]Galashov A et al.,Deep MMD gradient flow without adversarial training,ICLR2025
>
> [6]Aiello E et al.,Fast inference in denoising diffusion models via MMD finetuning,IEEE2024
>
> [7]Arbel M et al.,MMD Gradient Flow,NeurIPS2019

---

### Official Review · Reviewer_r8Za · 2025-03-10

**Overall Recommendation:** 4

**Summary:**

The paper proposes to learn the conditional distribution $p(x_0 | x_t)$ instead of learning the mean of this conditional distribution. To do this, the authors propose the use of scoring rules. In practice, this resorts to balancing two losses: the standard denoising loss, and a diversity loss on the learned conditional distribution $p(x_0 | x_t)$. Theoretical justifications of the proposed method are provided in the Gaussian case. Also, some theoretical comparison is done between two configurations of the loss (conditional vs joint distribution learning). Empirical evidence demonstrates that the proposed method improves sampling quality of diffusion models, mostly in the few-step setting.

**Claims And Evidence:**

Claims made in the submission are supported by clear and convincing evidence. The main claim is that learning conditional distribution $p(x_0 | x_t)$ allows to sample with a coarser discretization grid. This is validated empirically on image benchmarks with FID metric, and on robotic dataset.

**Essential References Not Discussed:**

No.

**Experimental Designs Or Analyses:**

I checked the design of all experiments, and there is no strong issue. The main weakness is the lack of comparison with other methods aiming high performance of diffusion models with low number of sampling steps.

**Methods And Evaluation Criteria:**

Method: The method is sound, elegant, and simple to implement.

Evaluation criteria: Evaluations criteria is well done. It consists in: theoretical proofs in a simplified setting (Gaussian target distribution), experiments on toy datasets with squared MMD metric, image datasets with FID, and robotic datasets with max success rate on a robotic manipulation task.
Something that is missing is a comparison with other methods that improve the sampling quality when using few number of steps. In the current approach, the authors only compare to vanilla DDIM. However, many approaches have been proposed to reduce the neural function evaluations (NFEs) required for achieving high sampling quality. For example, it would be interesting to compare to works such as DPM-solver (Lu et al.).

**Other Comments Or Suggestions:**

No.

**Other Strengths And Weaknesses:**

Weakness 1: Although this paper improves sampling quality at low number of sampling steps, it lacks comparisons with other methods that share the same purpose. Many different methods exist and share the same goal, e.g. improved samplers for diffusion models. How does this compare to some of these other methods?

Weakness 2: The proposed method leads to a computational overhead during training (generating m points for each training point). I did not see any study on the training time, but this would be important to understand the practical feasibility of the method.

**Questions For Authors:**

Could this framework be extended to score-based models with ODE sampling?

**Relation To Broader Scientific Literature:**

This paper proposes a novel view to improve the sampling quality of diffusion models with low number of steps. In my knowledge, the approach of learning $p(x_0 | x_t)$ with scoring rules is novel and original. Other approaches to learn $p(x_0 | x_t)$ made use of GANs and are discussed by the authors.

**Theoretical Claims:**

I checked the proof of Proposition 4.1 (proof of the loss in the Gaussian setting) and it is correct. I did not check the other proofs in detail. Also note that there is no theoretical proofs of the main claim, which is that the proposed method allows to reduce the number of sampling steps compared to a standard diffusion model.

---

> ### Author Rebuttal · Authors · 2025-03-31
>
> We thank reviewer r8Za for their overall positive assessment of our paper. Below, we address your comments.
>
> | Weakness 1
>
> As suggested, we have now compared our method to the DPM-solver++ [1]. We have also compared it to a SOTA (multistep) distillation method [2]. Please see our response to the reviewer waLv above, where we added these results.
>
> Our results show that our approach is competitive with multistep distillation (we compared different amounts of steps) and outperforms DPM-solver++ over 8 NFEs (whose performance degrades as expected as the number of steps increases, due to numerical instability).
>  Two compelling benefits of our approach over multistep distillation are that it does not rely on distillation, so there is no need to train a large (slow) teacher model; and that it does not require specifying additional sampler hyperparameters during training. It is certainly possible for our distributional approach to be combined with a modern distillation method. This is an interesting topic for future research.
>
> | Weakness 2
>
> We thank you for bringing up this point regarding the training time complexity. Indeed, as you suggested, our paper will benefit from adding a complexity analysis of our method. We will add the following section to the appendix of our paper.
>
> ## Computational complexity
> Our method has the same computational complexity during sampling as ordinary diffusion models, and only has an increased computational complexity during training which is detailed below.
>
> Assume that the computational complexity of the forward pass in our diffusion model is $O(F)$ and that the dimensionality of $x$ is $D$.
>
> **Diffusion loss case.** The loss function in eq. (14) with $m=1$ and $\lambda=0$, can be thought of as a standard diffusion model loss function. It requires $O(nF)$ evaluations to compute $x_{\theta}(t_i,X_{t_i})$ for every element of a batch. Then, to evaluate the loss (the norm), the complexity is $O(nD)$. Therefore the total cost of computing the loss is $O(n (F+D))$. The backwards pass is proportional to the forward pass and has computational complexity $O(nF)$.
>
> **Distribution loss case.** For $\lambda \neq 0$ and $m>1$, we first need $O(nmF)$ function evaluations to compute $x_{\theta}(t_i, X_{t_i}, \xi_{i,j})$ for $i=1,\ldots,n$ and for $\xi_{i,j}$, $j=1,\ldots,m$. After that, we need $O(nmD)$ operations to compute the first (diffusion-like) terms of the loss. Then, in order to compute the interaction terms, we need $O(nm(m-1)D)$ time. Therefore, the total cost is of computing the loss $O(nm(F+D) + nm(m-1)D)$. Naively, backwards pass will take $O(nmF + nm(m-1)F) = O(nm^2F)$ time. However, since the 2nd term uses the gradients of $x_{\theta}(t_i, X_{t_i}, \xi_{i,j})$, one could precompute these for all $i=1,\ldots,n$ and $j=1,\ldots,m$ and therefore decrease the total backwards cost to $O(nmF)$. We found that in practice, the XLA compiler in jax performs this optimization without explicit coding (see below).
>
> **Runs on real hardware.** We compared the training times on real hardware. We study the training of CIFAR-10 and we vary $m$. We report **steps per second** metric where a step corresponds to a full forward+backward step. As hardware, we use A100 GPU (40 Gb of memory) with batch size = 16, H100 GPU (80Gb of memory) with batch size = 64 and TPUv5p (with 95Gb of memory) with batch size = 64 (per device with 4 devices in total). The results below indicate that **steps per second** decreases proportionally to $m$.
>
> Hardware| Diffusion|Distributional (m=2)|Distributional (m=4)| Distributional (m=8)
> -|-|-|-|-
> A100 (bs=16)|  9.05 | 6.78 | 4.6 | 2.77
> H100 (bs=64) |  14.3  | 7.85 | 4.15 | 2.06
> TPUv5 (bs=64)|  11.2   | 8.5 | 4.3  | 2.22
>
> | Could this framework be extended to score-based models with ODE sampling?
>
> If the churn parameter $\epsilon$ is taken to be zero in the DDIM equations, one recovers a deterministic sampler which can be thought of as the discretization of an ODE. This ODE is a special case of the standard probability flow ODE; see Section 4.3 of [3] for a discussion. In particular, the DDIM update coincides with the exponential integrator for the probability flow ODE.
>
> We emphasize that our method can also be extended directly to sample a transport map between any two distributions $p_0$ and $p_1$ and does not require one of the distributions to be Gaussians. This is a framework for which flow matching/stochastic interpolants are typically used.  We will add a section in our paper to detail this extension.
>
> We hope that our updated experiments have addressed your concerns. If so, we would be grateful if you could consider increasing your score.
>
> **References**:
>
> [1] DPM-Solver++: Fast Solver for Guided Sampling of Diffusion Probabilistic Models, Cheng Lu et al., 2022
>
> [2] Multistep Distillation of Diffusion Models via Moment Matching, Tim Salimans et al., 2024.
>
> [3] Denoising diffusion implicit models, Jiaming Song et al., 2020.

---

### Official Review · Reviewer_waLv · 2025-03-12

**Overall Recommendation:** 4

**Summary:**

This paper proposes Distributional Diffusion Models (DDM), an acceleration method of diffusion model. Specifically, the authors follow the  Denoising Diffusion Implicit Models (DDIM) and replace the regression loss with a loss based on scoring rules to accomplish sample generation. This method learns the posterior distribution of clean data samples given their noisy versions, instead of only the mean of this distribution.  Additionally, the authors provide some theoretical analysis of the generalized scoring rules and validate DDM on image and robot trajectory generation.

**Claims And Evidence:**

yes

**Essential References Not Discussed:**

In DDM, the authors take DDIM as a starting point to introduce research on scoring rules. The main contribution is to replace the regression loss with a loss based on scoring rules and to provide the theoretical analysis. No significant related work appears to have been omitted.

**Experimental Designs Or Analyses:**

Yes, I have checked the experimental designs and analyses, including 2D experiments, image experiments and robotics experiments.

**Methods And Evaluation Criteria:**

The proposed method and evaluation criteria in this paper are both reasonable for diffusion acceleration.

**Other Comments Or Suggestions:**

Double quotation marks are wrong in Section 7 (line 419).

**Other Strengths And Weaknesses:**

Strengths:

1. The paper provides an acceleration method of diffusion models, learns the posterior distribution of clean data samples given their noisy versions, instead of only the mean of this distribution.

2. The paper replace the regression loss with a loss based on scoring rules, and provides the theoretical analysis.

3. The authors conduct so many experiments on image and robot trajectory generation.

4. The theoretical analysis is detailed.

Weaknesses:

1. In the experimental section, this study only trains a baseline unconditional diffusion model by optimizing (6), but it lacks comparisons with other acceleration methods. As mentioned in the introduction, improved numerical integrators[1], distillation techniques[2], and parallel simulation methods[3], among others, are commonly employed strategies. Comparisons with these methods would make the experiment more comprehensive.

[1] DPM-Solver-v3: Improved Diffusion ODE Solver with Empirical Model Statistics

[2] Multistep Distillation of Diffusion Models via Moment Matching

[3] Parallel Sampling of Diffusion Models

It would be better to compare with other baselines.

**Questions For Authors:**

I have no questions for the authors.

**Relation To Broader Scientific Literature:**

1. This paper  provides a unique research perspective by learning the posterior distribution of clean data samples given their noisy counterparts, rather than only the mean of this distribution.

2. Diffusion models are combined with traditional scoring rules to improve the loss function, enabling the model to learn the full distribution rather than the mean.

3. This paper provides the gaussian analysis, joint or conditional scoring rules analysis, and from kernel scores to diffusion losses analysis in the theoretical analysis section, which collectively explain why the use of scoring rules can improve the performance of diffusion models.

**Theoretical Claims:**

Yes, I have checked the gaussian analysis, joint or conditional scoring rules analysis, and from kernel scores to diffusion losses analysis in the theoretical analysis section. I did not find any apparent issues.

---

> ### Author Rebuttal · Authors · 2025-03-31
>
> We would like to thank reviewer waLv for their comments and positive feedback.
>
> As requested, we re-implemented and have added comparisons to a popular numerical solver DPM-solver++ [1] (as suggested by reviewer r8Za) and to the recommended distillation method [2]. We did implement the parallel sampling of diffusion models but haven't had enough time to sweep over hyperparameters to obtain competitive results. We will add this comparison as well as the comparisons below, in the final version of the paper.
>
> **Experimental details**: For all the sampling methods, we sweep over the “safety” parameter $\eta$, which specifies the time interval to be “[$\eta$,$1-\eta$]”. We consider values $[1e-1,1e-2,1e-3,1e-4]$. Moreover, for all the methods we sweep over the “churn” parameter which controls the stochasticity, we tried $[0.0,0.25,0.5,0.75,1.0]$. We found the stochastic version of DPM-solver++ (with a churn parameter close to $1$) overall worked the best. For ordinary diffusion and our method, we use a “uniform” time schedule. For DPM-solver++ we use the “logsnr” time schedule (see [1]), which led to the best results (we also tried EDM and uniform).
>
> For distillation comparisons, we distill pretrained diffusion models into a student model with a given number of student sampling steps. We train a distilled model for 100k iterations with batch size 256 for pixel-space and with batch size 32 for latent space models. We sweep over EMA decay and learning rate. For the student, we use a stochastic DDIM sampler with a churn parameter $\epsilon$ and we sweep over it with considered values $[0,0.5,1.0]$. At sampling time, we found that using a different churn parameter $\epsilon=0$ for the distilled student, led to the best results.
>
> For all the methods, we present the results with the best hyperparameters.
>
> **Results**: The results are presented in the table below. They indicate that our approach is competitive with multistep distillation (we compared different amounts of steps) and outperforms DPM-solver++ over 8 NFEs (whose performance degrades as expected as the number of steps increases due to numerical instability).
>  Two compelling benefits of our approach over multistep distillation are that it does not rely on distillation, so there is no need to train a large (slow) teacher model; and additionally it does not require specifying additional sampler hyperparameters during training. It is certainly possible for our distributional approach to be combined with a modern distillation method. This is an interesting topic for future research.
>
> The tables below contain FID scores (averaged across 3 random seeds) for different number of function evaluations (NFEs) and for different methods. In **bold**, we highlight the method with the lowest FID and in *italic*, we highlight the method with the second lowest FID.
>
> ## Dataset name = CIFAR-10 (Conditional)
> |NFEs| Diffusion | DPM++ | Distributional | Distillation |
> |------|-----------|-------|----------------|--------------|
> | 2|80.77|*19.41*|29.33|**5.19**|
> | 4|23.31|12.60|*4.67*|**3.84**|
> | 8|7.53|3.95|*3.21*|**3.13**|
> | 10|5.67|3.55|*3.19*|**2.99**|
> | 12|4.83|3.69|*3.03*|**2.93**|
> | 15|3.61|3.91|*2.87*|**2.76**|
>
> ## Dataset name = CelebA (Conditional)
> |NFEs| Diffusion | DPM++ | Distributional | Distillation |
> |------|-----------|-------|----------------|--------------|
> | 2|62.85|*58.39*|81.79|**7.02**|
> | 4|26.25|*20.39*|22.98|**6.47**|
> | 8|13.07|17.28|*4.94*|**4.43**|
> | 10|11.67|22.48|**3.60**|*4.43*|
> | 12|10.00|23.23|**3.37**|*4.25*|
> | 15|8.51|23.53|**3.45**|*4.27*|
>
> ## Dataset name = LSUN (Unconditional)
> |NFEs| Diffusion | DPM++ | Distributional | Distillation |
> |------|-----------|-------|----------------|--------------|
> | 2|238.85|*120.99*|233.24|**16.04**|
> | 4|*41.57*|43.60|69.90|**5.54**|
> | 8|10.82|12.95|*9.79*|**4.52**|
> | 10|9.08|17.85|*7.14*|**4.64**|
> | 12|7.04|16.07|*6.12*|**3.81**|
> | 15|6.51|23.91|*5.78*|**4.39**|
>
> ## Dataset name = Latent CelebA-HQ (Unconditional)
> |NFEs| Diffusion | DPM++ | Distributional | Distillation |
> |------|-----------|-------|----------------|--------------|
> | 2|101.47|114.44|*67.60*|**37.88**|
> | 4|53.83|26.05|*23.80*|**14.47**|
> | 8|15.60|20.70|*9.84*|**8.22**|
> | 10|11.89|21.84|*8.15*|**6.76**|
> | 12|9.97|24.61|*7.04*|**6.05**|
> | 15|8.39|27.31|*6.21*|**5.46**|
>
> We hope that our updated experiments have addressed your concerns. If so, we would be grateful if you could consider increasing your score.
>
> **References**:
>
> [1] DPM-Solver++: Fast Solver for Guided Sampling of Diffusion Probabilistic Models, Cheng Lu, Yuhao Zhou, Fan Bao, Jianfei Chen, Chongxuan Li, Jun Zhu, 2022
>
> [2] Multistep Distillation of Diffusion Models via Moment Matching, Tim Salimans et al., 2024.

---

### Decision · Program_Chairs · 2025-05-01

**Decision:**

Accept (poster)

**Comment:**

The manuscript proposes, in the context of diffusion models, learning the posterior distribution of clean samples given their noisy counterparts using a scoring rule–based loss. The reviewers found that the manuscript provides a novel view on improving the sampling quality of diffusion models and that the theoretical contributions are sound, elegant and analytically detailed. The reviewers had concerns regarding limitations in the empirical evaluation, particularly the lack of comparison to other methods and baselines, analysis of the computation overhead during training, and relation to other literature. However, the authors addressed all these concerns during rebuttal, and all reviewers recommend accepting the paper. Therefore, I also recommend accepting the paper. Since the rebuttal contains many additional details and experience, it is crucial that the authors incorporate all promised changes into the final version of the paper.